



# Factors controlling coccolithophore biogeography in the Southern Ocean

Cara Nissen[1], Meike Vogt[1], Matthias Münnich[1], Nicolas Gruber[1], and F. Alexander Haumann[1]

[1]Institute for Biogeochemistry and Pollutant Dynamics, ETH Zürich, Universitätstrasse 16, 8092 Zürich, Switzerland

**Correspondence:** C. Nissen (cara.nissen@usys.ethz.ch)

**Abstract.** The biogeography of Southern Ocean phytoplankton controls not only the local biogeochemistry, but also the export of macronutrients to lower latitudes and depth. Of particular relevance is the interaction between coccolithophores and diatoms, with the former being prevalent along the "Great Calcite Belt" (40-60°S), while diatoms tend to dominate the regions south of 60°S. To address the factors controlling coccolithophore distribution and the competition between them and diatoms, we use a regional high-resolution model (ROMS-BEC) for the Southern Ocean (24-78°S) that has been extended to include an explicit representation of coccolithophores. We assess the relative importance of bottom-up (temperature, nutrients, light) and top-down (grazing by zooplankton) factors in controlling Southern Ocean coccolithophore biogeography over the course of the growing season. In our simulations, coccolithophores are an important member of the Southern Ocean phytoplankton community, contributing 15% to annually integrated net primary productivity south of 30°S. Coccolithophore biomass is highest north of 50°S in late austral summer, when light levels are high and diatoms become silicate limited. Furthermore, we find top-down factors to be a major control on the relative abundance of diatoms and coccolithophores in the Southern Ocean. Consequently, when assessing potential future changes in Southern Ocean coccolithophore abundance, both abiotic (temperature, light, nutrients, pH) and biotic factors (interaction with diatoms and zooplankton) need to be considered.

## 1 Introduction

The ocean is changing at an unprecedented rate as a consequence of increasing anthropogenic $CO_2$ emissions and related climate change. Changes in density stratification and nutrient supply, as well as acidification lead to changes in phytoplankton community composition and consequently ecosystem structure and function. Some of these changes are already observable today (e.g. Soppa et al., 2016; Winter et al., 2013) and may have cascading effects on global biogeochemical cycles and oceanic carbon uptake (Laufkötter et al., 2016; Freeman and Lovenduski, 2015; Cermeño et al., 2008). Changes in Southern Ocean (SO) biogeography are especially critical due to the importance of the SO in fueling primary production at lower latitudes through the lateral export of nutrients (Sarmiento et al., 2004) and in taking up anthropogenic $CO_2$ (Frölicher et al., 2015). For the carbon cycle, the ratio of calcifying and silicifying phytoplankton is crucial due to the counteracting effects of calcification and photosynthesis on seawater $pCO_2$, which ultimately controls $CO_2$ exchange with the atmosphere, and the differing ballasting effect of calcite and silicate shells for organic carbon export.



Calcifying coccolithophores and silicifying diatoms are globally ubiquitous phytoplankton functional groups (O'Brien et al., 2013; Leblanc et al., 2012). Diatoms are a major contributor to global annual net primary production (40% of NPP, Sarthou et al., 2005) and phytoplankton biomass (≈6-70%, Buitenhuis et al., 2013b). In comparison, coccolithophores contribute less to global NPP (0.4-17%, O'Brien, 2015; Jin et al., 2006; Moore et al., 2004; Gregg and Casey, 2007a) and biomass (≈0.04-

6%, Buitenhuis et al., 2013b). However, coccolithophores are the major phytoplanktonic calcifier (Iglesias-Rodríguez et al., 2002), thereby significantly impacting the global carbon cycle. Diatoms dominate the phytoplankton community in the SO (e.g. Trull et al., 2018; Swan et al., 2016; Wright et al., 2010), but coccolithophores have received increasing attention in recent years. Satellite imagery of particulate inorganic carbon (PIC, a proxy for coccolithophore abundance) revealed the "Great Calcite Belt" (GCB, Balch et al., 2011), an annually reoccurring circumpolar band of elevated PIC concentrations between

40°S and 60°S. In-situ observations confirmed coccolithophore abundances of up to $2.4 \cdot 10^3$ cells ml$^{-1}$ in the Atlantic sector, up to $3.8 \cdot 10^2$ cells ml$^{-1}$ in the Indian sector (Balch et al., 2016) and up to $5.4 \cdot 10^2$ cells ml$^{-1}$ in the Pacific sector of the SO (Cubillos et al., 2007) with *Emiliania huxleyi* being the dominant species (Balch et al., 2016; Saavedra-Pellitero et al., 2014). However, the contribution of coccolithophores to total SO phytoplankton biomass and NPP has not yet been assessed. Locally, elevated coccolithophore abundance in the GCB has been found to turn surface waters into a source of $CO_2$ for the atmosphere

(Balch et al., 2016), emphasizing the necessity to understand the controls on their abundance in the SO in the context of the carbon cycle and climate change. While coccolithophores have been observed to have moved polewards in recent decades (Rivero-Calle et al., 2015; Winter et al., 2013; Beaugrand et al., 2012), their response to the combined effects of future warming and ocean acidification is still subject to debate (Schlüter et al., 2014; Beaugrand et al., 2012; Beaufort et al., 2011; Iglesias-Rodríguez et al., 2008; Riebesell et al., 2000). As their response will also crucially depend on future phytoplankton community

composition and predator-prey interactions (Dutkiewicz et al., 2015), it is essential to assess the controls on their abundance in today's climate.

Coccolithophore biomass is controlled by a combination of bottom-up (physical/biogeochemical environment) and top-down factors (predator-prey interactions), but the relative importance of the two has not yet been assessed for coccolithophores in the SO. Bottom-up factors directly impact phytoplankton growth, and diatoms and coccolithophores are traditionally discriminated

based on their differing requirements for nutrients, turbulence and light. Based on this, Margalef's mandala predicts a seasonal succession from diatoms to coccolithophores as light levels increase and nutrient levels decline (Margalef, 1978). In-situ studies assessing SO coccolithophore biogeography have found coccolithophores under various environmental conditions (e.g. Trull et al., 2018; Charalampopoulou et al., 2016; Balch et al., 2016; Saavedra-Pellitero et al., 2014; Hinz et al., 2012), thus suggesting a wide ecological niche, but all of the mentioned studies have almost exclusively focussed on bottom-up controls.

However, phytoplankton growth rates do not necessarily covary with biomass accumulation rates. Using satellite data from the North Atlantic, Behrenfeld (2014) stresses the importance of simultaneously considering bottom-up and top-down factors when assessing seasonal phytoplankton biomass dynamics and succession of different phytoplankton types, owing to the spatially and temporally varying relative importance of the physical/biogeochemical and the biological environment. Other studies have shown zooplankton grazing to control total phytoplankton biomass (Le Quéré et al., 2016), phytoplankton community

composition (Scotia Weddell Sea, Granéli et al., 1993) and ecosystem structure (Smetacek et al., 2004; De Baar, 2005) in the

SO, suggesting that top-down control might also be an important driver for the relative abundance of coccolithophores and diatoms.

While none of the SO in-situ studies directly assessed interactions of diatoms and coccolithophores over the course of the year, some in-situ studies infer a diatom-coccolithophore succession from depleted silicate coinciding with iron levels high

enough to sustain elevated coccolithophore abundance (highFe-lowSi niche, Balch et al., 2016, 2014; Painter et al., 2010). In contrast to this, recent in-situ and satellite studies find coccolithophores and diatoms to coexist rather than succeed each other throughout the growth season in the North Atlantic (Daniels et al., 2015) and the global open ocean (Hopkins et al., 2015). In fact, large areas of the GCB have been identified as "coexistence" areas (Hopkins et al., 2015), thereby putting into question the succession pattern predicted by Margalef's mandala (Margalef, 1978) and results of in-situ studies for the SO (Balch et al.,

2016, 2014; Painter et al., 2010). This highlights the necessity to better understand the drivers and seasonal dynamics of the relative importance of coccolithophores and diatoms in the SO before assessing potential future changes.

In this study, we use a regional high-resolution model for the SO to simultaneously assess the relative importance of bottom-up versus top-down factors in controlling SO coccolithophore biogeography over a complete annual cycle. In particular, we assess the role of diatoms in constraining high coccolithophore abundance and the importance of zooplankton grazing for the

relative importance of coccolithophores and diatoms in the GCB area.

## 2 Methods

### 2.1 Model description: ROMS-BEC with explicit coccolithophores

We use a regional, circumpolar SO setup of the UCLA-ETH version of the Regional Ocean Modeling System (ROMS, Shchepetkin and McWilliams, 2005; Haumann, 2016) with a latitudinal range from $\approx$24°S-78°S and an open northern boundary. The

primitive equations are solved on a curvilinear grid: The model setup has 64 topography-following vertical levels, its horizontal resolution for this study is $\frac{1}{4}^{\circ}$ (5.4-25.4 km) and the time step is 1600 seconds.

Coupled to this is an extended version of the biogeochemical model BEC (Moore et al., 2013), that we modified to include an explicit parametrization of coccolithophores, as well as an updated formulation for sedimentary iron fluxes to allow for temporal and spatial variability of these fluxes (Dale et al., 2015). BEC resolves the cycling of carbon, nitrogen, phosphorus, silicate and

iron by simulating a total of 30 tracers. Besides explicit coccolithophores, it includes three phytoplankton (diatoms, $N_2$-fixing diazotrophs, and a mixed small phytoplankton class (SP)) and one zooplankton functional types (PFT). Phytoplankton C/N/P stoichiometry in photosynthesis is fixed close to Redfield ratios (117:16:1, Anderson and Sarmiento, 1994), but the ratios of C/Fe, Si/C and Chl/C vary according to surrounding nutrient levels. Detrital matter is split into a non-sinking and a sinking pool, with ballasting of the latter by atmospheric dust, biogenic silica or calcium carbonate (Armstrong et al., 2002). Dissolved

inorganic carbon (DIC) and alkalinity are included to complete the cycling of carbon in the model.

The phytoplankton PFTs differ with respect to their maximum growth rate ($\mu_{max}$), temperature ($Q_{10}$) and light ($\alpha_{PI}$) sensitivities, half-saturation constants for nutrient uptake (k), as well as grazing preferences by zooplankton ($\gamma_{max}$, Table 1). The SO coccolithophore community appears to mainly consist of the ubiquitous *Emiliania huxleyi* (mainly the lightly calcified mor-




**Table 1.** Most relevant BEC parameters for this study as used in the reference run (see section 2.2) for the four phytoplankton PFTs coccolithophores (C), diatoms (D), small phytoplankton (SP), and diazotrophs (N). Z=zooplankton, P=phytoplankton, PI=photosynthesis-irradiance.

| Parameter | Unit | Description | C | D | SP | N |
|---|---|---|---|---|---|---|
| $\mu_{max}$ | d$^{-1}$ | max. growth rate at 30° C | 3.8 | 4.6 | 3.6 | 0.9 |
| $Q_{10}$ | | temperature sensitivity | 1.45 | 1.55 | 1.5 | 1.5 |
| $k_{NO3}$ | mmol m$^{-3}$ | half-saturation constant for NO$_3$ | 0.3 | 0.5 | 0.1 | 1.0 |
| $k_{NH4}$ | mmol m$^{-3}$ | half-saturation constant for NH$_4$ | 0.03 | 0.05 | 0.01 | 0.15 |
| $k_{PO4}$ | mmol m$^{-3}$ | half-saturation constant for PO$_4$ | 0.03 | 0.05 | 0.01 | 0.02 |
| $k_{DOP}$ | mmol m$^{-3}$ | half-saturation constant for DOP | 0.3 | 0.9 | 0.26 | 0.09 |
| $k_{Fe}$ | nmol m$^{-3}$ | half-saturation constant for Fe | 0.10 | 0.12 | 0.08 | 0.08 |
| $k_{SiO3}$ | mmol m$^{-3}$ | half-saturation constant for SiO$_3$ | - | 1.0 | - | - |
| $\alpha_{PI}$ | $\frac{\text{mmol C m}^2}{\text{mg Chl W s}}$ | initial slope of PI-curve | 0.4 | 0.44 | 0.44 | 0.38 |
| $\gamma_{max}$ | d$^{-1}$ | max. growth rate of Z grazing on P | 4.4 | 3.8 | 4.4 | 2.0 |
| $z_{grz}$ | mmol m$^{-3}$ | half-saturation constant for ingestion | 1.05 | 1.0 | 1.05 | 1.2 |

photype B/C, see e.g. Saavedra-Pellitero et al., 2014; Krumhardt et al., 2017) and parameter values used for coccolithophores here are based on available data of this species in the literature, both from in-situ and laboratory studies (Daniels et al., 2014; Heinle, 2013; Buitenhuis et al., 2008; Zondervan, 2007; Nielsen, 1997; Le Quéré et al., 2016, and references therein). Based on the available information, parameter values for coccolithophores lie between those of diatoms and SP (Table 1). Due to their smaller size, coccolithophores have a higher nutrient affinity (smaller half-saturation constants, Eppley et al. (1969)) and a smaller maximum growth rate than diatoms (Buitenhuis et al., 2008). Coccolithophores grow well at high light intensities and at a range of different temperatures, but have been shown to be light-inhibited at low light levels (<1 W m$^{-2}$, Zondervan, 2007) and to reduce their growth at low temperatures (<6°C, Buitenhuis et al., 2008). For this study, we use a constant calcite-to-organic matter (CaCO$_3$:C$_{org}$) production ratio for coccolithophores of 0.2 (SO *Emiliania huxleyi* B/C, Müller et al., 2015). Previous work has shown this ratio to vary from 0.1-0.3 across environmental conditions for the SO morphotype of *Emiliania huxleyi* (Krumhardt et al., 2017), and we assess the sensitivity of integrated annual calcification estimates to this ratio in section 4.2.

In BEC, phytoplankton are grazed by a single zooplankton PFT, comprising characteristics of both micro- and macrozooplankton (Moore et al., 2002; Sailley et al., 2013). The single zooplankton PFT grazes on all phytoplankton PFTs using a Holling-Type II ingestion function (Holling, 1959). Microzooplankton exert the biggest grazing pressure on coccolithophores, possibly mainly through non-selective grazing for species like *Emiliania huxleyi* (Monteiro et al., 2016). In BEC, we assign the same maximum zooplankton growth rate ($\gamma_{max}$, Table 1) for feeding on SP and coccolithophores, thereby assuming that only differences in their absolute biomass concentrations leads to differences in grazing pressure, not the absence/presence of a coccosphere. In contrast, diatoms are mainly grazed by larger, slower-growing macrozooplankton (lower $\gamma_{max}$, Table 1). A full





description of the model equations regarding phytoplankton growth and loss terms can be found in section 3 and in appendix B.

## 2.2 Model setup & baseline simulation

At the surface, ROMS-BEC is forced by daily fluxes of momentum, heat and freshwater from ERA-Interim (Dee et al., 2011).
These fluxes are obtained by first calculating monthly climatological fluxes from 1979-2014 and then adding daily anomalies of the year 2003 to account for higher-frequency variability. The surface freshwater flux is corrected for river runoff, sea ice formation and melting (Haumann, 2016), and dust deposition (Mahowald et al., 2009) is scaled by the monthly climatological sea ice cover.

At the open northern boundary, the model is forced with monthly climatological fields for all tracers. Current velocities are
taken from SODA (Simple Ocean Data Assimilation, version 1.4.2, Carton and Giese, 2008), temperature and salinity from WOA (World Ocean Atlas 2013, 0.25° horizontal resolution, Locarnini et al., 2013; Zweng et al., 2013). For BEC, WOA data are used for macronutrients (1° horizontal resolution, Garcia et al., 2013b) and oxygen (1° horizontal resolution, Garcia et al., 2013a), GLODAP data for DIC and alkalinity (Global Ocean Data Analysis Project, Key et al., 2004). Dissolved iron, ammonium and dissolved organic carbon, nitrogen, phosphorus and iron fields are from climatological model output from the
global model CESM-BEC (Yang et al., 2017). Phytoplankton chlorophyll biomass fields are taken from a climatological surface chlorophyll field (NASA-OBPG, 2014b) using a constant partitioning of the different phytoplankton PFTs to total chlorophyll everywhere at the boundary (SP: 90%, diatoms: 4.5%, coccolithophores: 4.5%, diazotrophs: 1%) and then extrapolating to depth according to Morel and Berthon (1989). Phytoplankton carbon biomass fields are then derived using a constant carbon-to-chlorophyll ratio of three for diatoms and five for all other PFTs. To minimize model drift in the physical parameters, sea
surface temperature (Reynolds et al., 2007) and salinity (Good et al., 2013) fields are restored wherever sea ice is absent, with a restoring time scale of 45 days for salinity and a spatially and temporally varying sensitivity of the surface heat flux to sea surface temperatures (Haumann, 2016). No restoring is applied to the biogeochemical tracers.

The model is first spun up from rest for velocity in a physics-only setup for 30 years and subsequently for another 10 years in the coupled ROMS-BEC setup. All tracers are initialized using the same data sources for initial fields as used for the lateral
boundary forcing. The reference simulation analyzed in this study is run for 10 years after the coupled ROMS-BEC spinup, of which only a daily climatology of the last 5 years is analyzed. To capture 5 full seasonal cycles at high southern latitudes, we calculate the climatology from 1 July of year 5 until 30 June of year 10 of the simulation. Ultimately, we focus the analysis in this study on the area south of 30°S to minimize potential effects of the open northern boundary on biomass distributions.

## 2.3 Sensitivity simulations

We perform a set of sensitivity simulations to assess the sensitivity of SO coccolithophore biogeography to choices of model parameters, parametrizations, and biases in the physical fields (Table 2). At first, we consecutively set the different coccolithophore parameters to the corresponding diatom value (run 1-9). For all simulations, we then quantify the sensitivity as a change of each PFT's annual mean surface biomass, focusing particularly on coccolithophores in section 4.7.





**Table 2.** Overview of sensitivity simulations. 1-9: Sensitivity to chosen parameter values of coccolithophores. See Table 1 for parameter values of coccolithophores in reference run. 10-11: Sensitivity to the chosen grazing formulation. 12-13: Sensitivity to biases in temperature and mixed layer depth. C=coccolithophores, D=diatoms.

| | Run Name | Description |
|---|---|---|
| 1 | GROWTH | Set $\mu_{\max}^{C}$ to $\mu_{\max}^{D}$ |
| 2 | ALPHA$_{PI}$ | Set $\alpha_{PI}^{C}$ to $\alpha_{PI}^{D}$ |
| 3 | Q10 | Set $Q_{10}^{C}$ to $Q_{10}^{D}$ |
| 4 | GRAZING | Set $\gamma_{\max}^{C}$ and $z_{grz}^{C}$ to $\gamma_{\max}^{D}$ and $z_{grz}^{D}$ |
| 5 | IRON | Set $k_{Fe}^{C}$ to $k_{Fe}^{D}$ |
| 6 | SILICATE | Limit coccolithophore growth by silicate by using $k_{SiO3}^{D}$ |
| 7 | NITRATE | Set $k_{NO3}^{C}$ and $k_{NH4}^{C}$ to $k_{NO3}^{D}$ and $k_{NH4}^{D}$ |
| 8 | PHOSPHATE | Set $k_{PO4}^{C}$ and $k_{DOP}^{C}$ to $k_{PO4}^{D}$ and $k_{DOP}^{D}$ |
| 9 | NUTRIENTS | Set all $k_{Nutrient}^{C}$ to $k_{Nutrient}^{D}$ |
| 10 | HOLLINGIII | Use $\frac{P'^{i} \cdot P'^{i}}{z_{grz}^{i} \cdot z_{grz}^{i} + P'^{i} \cdot P'^{i}}$ instead of $\frac{P'^{i}}{z_{grz}^{i} + P'^{i}}$ in Eq. 5 |
| 11 | ACTIVE_SWITCHING | Use $\frac{P'^{i}}{\sum_{j=1}^{4} P'^{j}}$ in Eq. 5 |
| 12 | TEMP | Reduce temperature in BEC subroutine by 1°C everywhere |
| 13 | MLD | Reduce incoming PAR in BEC subroutine by -20% everywhere |

Second, we assess the sensitivity of the results to the chosen grazing formulation by performing two additional simulations: We first replace the Holling type II ingestion term (Eq. 5) by a Holling type III term (run 10, Holling, 1959). Thereby, the grazing pressure is decreased on prey in low concentrations. We then add an active prey switching term to the original Holling type II formulation in Eq. 5 so that the grazing pressure is linearly proportional to each phytoplankton's relative importance for total phytoplankton biomass (run 11). Similarly to the simulation using a Holling type III ingestion term, we expect the less abundant PFTs to profit most, as relatively, more of the total grazing pressure acts on the most abundant PFT (Vallina et al., 2014).

Ultimately, we performed two additional sensitivity simulations (run 12 & 13 in Table 2) to assess the effect of biases in the physical fields (temperature and mixed layer depth) on coccolithophore biogeography. To do this, we reduce temperatures by 1°C (run 12) and the incoming PAR field by 20% (to counteract bias in MLD, run 13) everywhere for the biological subroutine only.

All sensitivity runs start from the common spin up described in section 2.2 and only differ in their respective settings within BEC (Table 2). As for the control run, each simulation is run for 10 years of which the average over the last 5 years is analyzed.



## 3  Analysis framework: Factors controlling phytoplankton growth & loss

To disentangle the effect of the different controlling factors, relative growth and grazing ratios are computed as introduced by Hashioka et al. (2013) and outlined in the following. In BEC, phytoplankton biomass $P^i$ ($i \in \{C, D, SP, N\}$) is the balance of growth ($\mu^i$) and loss terms (grazing by zooplankton $\gamma_g^i$, non-grazing mortality $\gamma_m^i$ and aggregation $\gamma_a^i$, see Appendix B for a full description of the model equations regarding phytoplankton growth and loss terms):

$$\frac{dP^i}{dt} = \mu^i \cdot P^i - \gamma^i(P^i) \cdot P^i \tag{1}$$

$$= \mu^i \cdot P^i - \gamma_g^i(P^i) \cdot P^i - \gamma_m^i \cdot P^i - \gamma_a^i(P^i) \cdot P^i \tag{2}$$

with the specific phytoplankton growth $\mu^i$ being dependent on the maximum growth rate $\mu_{max}^i$ (Table 1), temperature ($f^i(T)$, Eq. B5), nutrient availability ($g^i(N)$, Eq. B8; nitrate, ammonium, phosphorus and iron for all PFTs, silicate for diatoms only) and light levels ($h^i(I)$, Eq. B9; following the growth model by Geider et al. (1998)) :

$$\mu^i = \mu_{max}^i \cdot f^i(T) \cdot g^i(N) \cdot h^i(I) \tag{3}$$

The relative growth ratio $\mu_{rel}^{ij}$ between two phytoplankton types $i$ and $j$, e.g. diatoms and coccolithophores, can then be defined as (Hashioka et al., 2013):

$$\mu_{rel}^{DC} = \log \frac{\mu^D}{\mu^C}$$
$$= \log \frac{\mu_{max}^D}{\mu_{max}^C} + \log \frac{f^D(T)}{f^C(T)} + \log \frac{g^D(N)}{g^C(N)} + \log \frac{h^D(I)}{h^C(I)} \tag{4}$$

If $\mu_{rel}^{DC}$ is negative, the specific growth rate of coccolithophores is larger than that of diatoms and bottom up factors promote the dominance of coccolithophores over diatoms (and vice versa). Based on chosen parameter values for coccolithophores and diatoms in ROMS-BEC (see section 2.1 and Table 1), coccolithophores grow better than diatoms when nutrient concentrations are low and irradiance is high (towards the end of the growth season). Simultaneously, coccolithophores are limited less by the ambient temperature than diatoms. Since the coccolithophores' maximum growth rate is lower than that of diatoms (Table 1), ideal environmental conditions are required for coccolithophores to overcome this disadvantage and to develop a higher specific growth rate than diatoms. Whether the resulting $\mu_{rel}^{DC}$ is positive or negative at any given location and point of time will therefore depend on the complex interplay of the physical and biogeochemical environment at every location.

The specific grazing rate $\gamma_g^i$ of the generic zooplankton on the respective phytoplankton $i$ is described by

$$\gamma_g^i = \gamma_{max}^i \cdot f^Z(T) \cdot Z \cdot \frac{P'^i}{z_{grz}^i + P'^i} \tag{5}$$

with $Z$ being zooplankton biomass, $f^Z(T)$ the temperature scaling function (Eq. B13), $\gamma_{max}^i$ the maximum growth rate of zooplankton when feeding on phytoplankton $i$ (Table 1), $z_{grz}^i$ the respective half-saturation coefficient for ingestion (Table 1) and $P'^i$ the phytoplankton biomass, which was corrected for a loss threshold below which no losses occur (Eq. B11).



Ultimately, the relative grazing ratio $\gamma_{\mathrm{g_{rel}}}^{\mathrm{ij}}$ of phytoplankton $i$ and $j$, e.g. diatoms and coccolithophores, is defined as (Hashioka et al., 2013):

$$\gamma_{\mathrm{g,rel}}^{\mathrm{DC}} = \log \frac{\gamma_{\mathrm{g}}^{\mathrm{C}}/P^{\mathrm{C}}}{\gamma_{\mathrm{g}}^{\mathrm{D}}/P^{\mathrm{D}}} \tag{6}$$

If $\gamma_{\mathrm{g,rel}}^{\mathrm{DC}}$ is negative, the specific grazing rate on diatoms is larger than that on coccolithophores and grazing promotes the dominance of coccolithophores over diatoms (and vice versa). While the maximum grazing rate is larger on coccolithophores

than on diatoms (see section 2.1 and Table 1), the interplay with biomass concentrations at any given location and point of time will decide whether $\gamma_{\mathrm{g,rel}}^{\mathrm{DC}}$ is positive or negative, i.e. whether the strength and direction of the grazing pressure favors coccolithophores or diatoms.

To assess differences in biomass accumulation rate between different PFTs, we compute clearance rates $c^{\mathrm{i}}$ of phytoplankton $i$ as the ratio of the specific grazing rate and the respective phytoplankton's biomass $P^{\mathrm{i}}$:

$$c^{\mathrm{i}} = \frac{\gamma_{\mathrm{g}}^{\mathrm{i}}}{P^{\mathrm{i}}} \tag{7}$$

The higher the clearance rate, the more difficult it is for a phytoplankton $i$ to accumulate biomass.

In contrast to Hashioka et al. (2013), who analyzed the relative growth/grazing ratio at the time of the annual maximum total chlorophyll concentration, we analyze them as a function of time to assess temporal variability in the controls on phytoplankton competition. We particularly focus on the interplay between coccolithophores and diatoms, as maximum coccolithophore abundance in the SO may be facilitated by declining diatom abundance (indicated by depleted silicate levels, see e.g. Balch

et al., 2014).

## 4 Results

### 4.1 Model evaluation

Phytoplankton growth directly responds to the physical and biogeochemical environment (Eq. 3), which is why systematic biases in the underlying bottom-up factors have to be assessed to understand biases in simulated phytoplankton biogeography

and phenology. The data sets used for the model evaluation are presented in Table A1, a more detailed description is found in the supplementary material.

In ROMS-BEC, SST is on average 0.9°C/0.2°C too high and the ML is 1m/5m too shallow in austral summer south/north of 60°S, respectively (Fig. S1), leading to an overestimation of phytoplankton growth (Fig. S1-S3). Macronutrients in ROMS-BEC are generally too low at the surface compared to WOA data (especially south of 60°S, Fig. S1 & S2), caused either by

too much uptake by phytoplankton, too little supply from below, or both.

Total SO summer surface chlorophyll in ROMS-BEC reproduces the general south-north gradient as detected by remote sensing (Fig. 1a & b), with highest values above 10 mg chl m$^{-3}$ in our model in areas close to the Antarctic continent and lower concentrations of around 0.1 mg chl m$^{-3}$ north of 40°S. However, integrated over 30-90°S, ROMS-BEC overestimates





**Table 3.** Comparison of ROMS-BEC based phytoplankton biomass, production, calcification and export estimates with available observations (given in parentheses). See Table A1 for data sources.

| | | ROMS-BEC (Data) | | |
| --- | --- | --- | --- | --- |
| | | **30-90°S** | **40-60°S** | **60-90°S** |
| Surface chlorophyll biomass | total, annual mean [Gg chl] | 39.95 (34.52) | 15.72 (17.14) | 19.67 (9.49) |
| Coccolithophore carbon biomass | 0-200m, annual mean [Pg C] | 0.012 (global: 0.001-0.03) | 0.0055 | 0.0007 |
| Diatom carbon biomass | 0-200m, annual mean [Pg C] | 0.071 (global: 0.013-0.75) | 0.0375 | 0.0248 |
| NPP | Pg C yr$^{-1}$ | 15.7 (12.1-12.5) | 8.09 (5.8-6.2) | 2.6 (0.68-1.7) |
| | Coccolithophores [%] | 14.7 | 10.9 | 0.8 |
| | Diatoms [%] | 64.6 | 75.8 | 88.2 |
| | SP [%] | 19.7 | 13.2 | 11.0 |
| Calcification | Pg C yr$^{-1}$ | 0.46 (0.79) | 0.18 (0.45) | 0.004 (0.15) |
| POC export at 100m | Pg C yr$^{-1}$ | 2.88 (2.3-2.96) | 1.64 (1.18-1.98) | 0.53 (0.21-0.24) |
| PIC export at 100m | Pg C yr$^{-1}$ | 0.30 (0.52) | 0.11 (0.28) | 0.002 (0.10) |
| PIC:POC export ratio at 100m | - | 0.10 | 0.07 | 0.005 |

annual mean satellite derived surface chlorophyll biomass estimates by 15.7% (40 Gg chl in ROMS-BEC compared to 34.5 Gg chl in satellite product, Table 3 and Fig. S2) and satellite derived NPP by 25.6-29.8% (15.7 compared to 12.1-12.5 Pg C yr$^{-1}$, Table 3 and Fig. S2 & S3). This overestimation is mainly driven by the area south of 60°S (NPP and surface chlorophyll are overestimated by a factor 3 and 2, respectively), while between 40-60°S, surface chlorophyll biomass is in fact underestimated by 8% (Table 3 and Fig. S2).

The overestimation of phytoplankton production can at least partly be attributed to biases in SST and MLD promoting phytoplankton growth (see also discussion section 5.4). However, data coverage south of 60°S, an area almost completely covered by sea ice every year, is low (Holte et al., 2017, their Fig. 1), impeding the assessment of model performance and the attribution of the production/biomass bias to underlying physcial fields in this area. Additionally, satellite derived surface chlorophyll and NPP fields are known to be associated with significant errors in high latitudes due to low sun elevation, clouds or sea ice cover, complicating model assessment (Gregg and Casey, 2007b). In addition to the underlying physical and biogeochemical fields, phytoplankton biomass is also controlled by loss rates (Eq. 2). Since production is overestimated between 40-60°S in ROMS-BEC compared to satellite derived estimates, the concurrent underestimation of surface chlorophyll biomass hints towards overestimated phytoplankton losses for this area (see also discussion section 5.4).

## 4.2 Quantifying the importance of SO coccolithophores for biogeochemical cycles

In ROMS-BEC, the annual mean SO coccolithophore carbon biomass within the top 200 m is 0.012 Pg C (Table 3), which is within the globally estimated range based on in-situ observations (0.001-0.03 Pg C, see O'Brien et al., 2013). Total NPP in





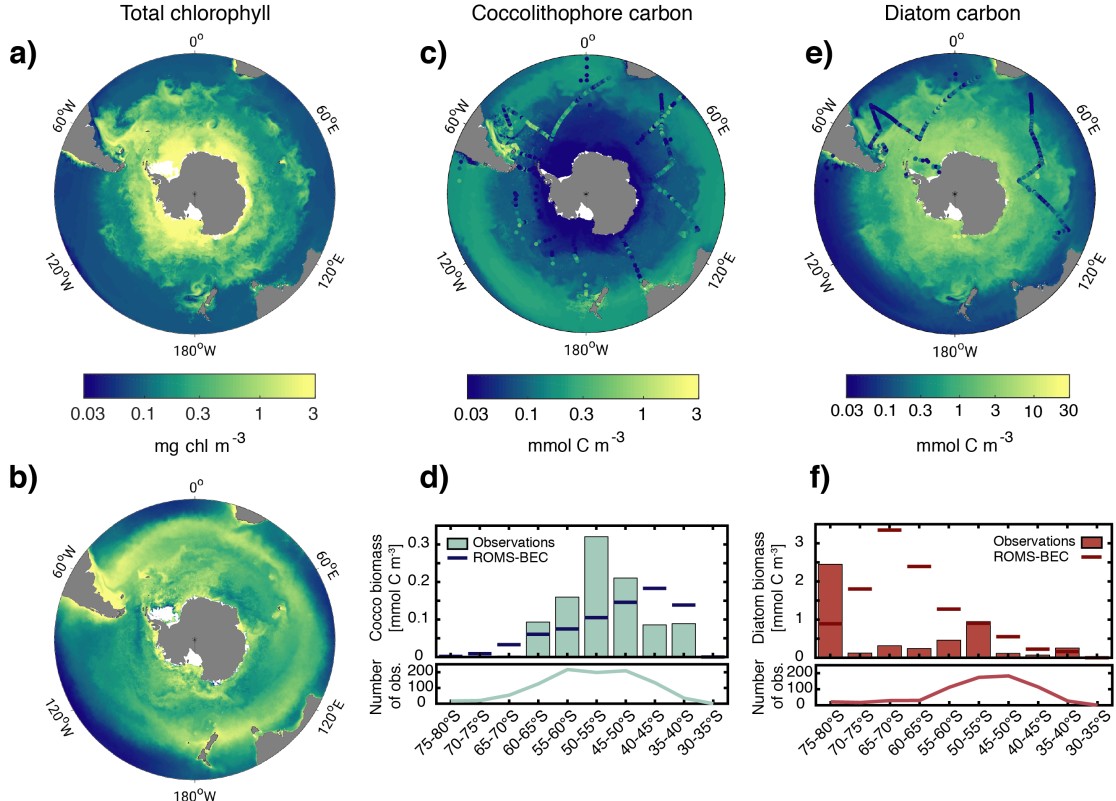

**Figure 1.** Biomass distributions for December-March (DJFM). Total surface chlorophyll [mg chl m$^{-3}$] in a) ROMS-BEC and b) MODIS-Aqua climatology (NASA-OBPG, 2014a), using the chlorophyll algorithm by Johnson et al. (2013). c) & e) Mean top 50 m c) coccolithophore and e) diatom carbon biomass [mmol C m$^{-3}$] in ROMS-BEC. Coccolithophore and diatom biomass observations from the top 50 m are indicated by colored dots in c) & e), respectively. d) & f) Mean top 50 m zonally averaged d) coccolithophore and f) diatom carbon biomass [mmol C m$^{-3}$], binned into 5° latitudinal intervals for ROMS-BEC (line) and observations (bars). The lower panels show the number of observations used to obtain the bars in the respective upper panels. Note that a)-b) are on the same scale, while the scales in panels c)-f) are different. For more details on the biomass validation, see Table A1 and the supplementary material.

ROMS-BEC south of 30°S is 15.7 Pg C yr$^{-1}$ with diatoms contributing 64.6%, small phytoplankton 19.7%, coccolithophores 14.7% and diazotrophs 1%. Compared to previous global estimates, annual coccolithophore NPP south of 30°S alone (2.3 Pg C yr-1) accounts for 3.5-4.5% of total global NPP (58±7 Pg C yr$^{-1}$, Buitenhuis et al., 2013a). Modeled integrated calcification amounts up to 0.46 Pg C yr$^{-1}$ south of 30°S (using a CaCO$_3$:C$_{org}$ production ratio of 0.2 for coccolithophores). Applying the full experimental range of CaCO$_3$:C$_{org}$ production ratios of SO *Emiliania huxleyi* (0.1-0.3, Krumhardt et al., 2017), and accounting for the relative error associated with the satellite calcification estimate (18.75% based on global data, Balch et al., 2007), the model estimate (0.23-0.69 Pg C yr$^{-1}$) falls within the range estimated from satellite observations (0.64-0.94 Pg C yr$^{-1}$, obtained using Eq. 1 in Balch et al. (2007) with satellite sea surface temperature, chlorophyll and PIC concentrations





from NASA-OBPG (2014c,a,d), see section S1 in supplementary material). Compared to global satellite derived estimates, the simulated calcification estimate south of 30°S accounts for 19% (8.1-35.4%) of global calcification.

The ratio of particulate inorganic (calcite) to organic carbon exported to depth (PIC:POC ratio, typically reported at depths of ≈100m) is important for the long-term fate of atmospheric $CO_2$. In ROMS-BEC, PIC and POC export south of 30°S are 0.3 Pg C yr$^{-1}$ and 2.88 Pg C yr$^{-1}$, respectively. Accounting for the uncertainty in the $CaCO_3:C_{org}$ production ratio of coccolithophores (Krumhardt et al., 2017), the average PIC:POC export ratio is 0.1 (0.05-0.16), which is in the same range as previously estimated for the global mean export ratio (0.06±0.03, Sarmiento et al., 2002). PIC:POC export ratios in ROMS-BEC are highest on the Patagonian Shelf (0.06-0.19 for the annual mean, 0.10-0.31 for summer mean only, not shown) where coccolithophore biomass is highest (see section 4.3), consistent with the elevated PIC:POC export ratios reported for this area by Balch et al. (2016, up to 0.33 in January).

### 4.3 Phytoplankton biogeography and community composition in the SO

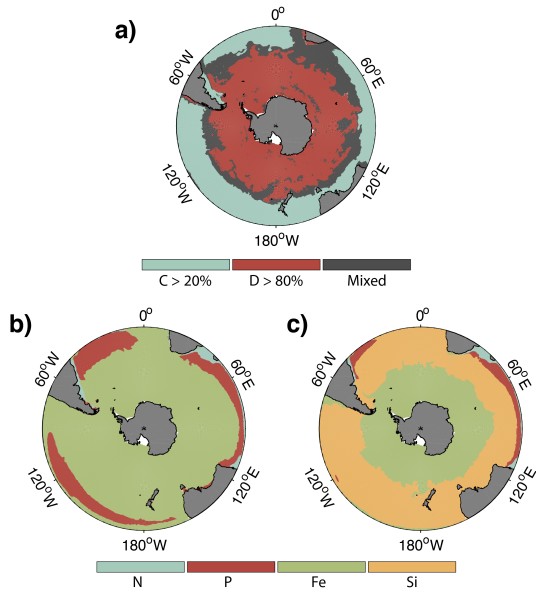

**Figure 2.** a) Spatial distribution of phytoplankton communities in ROMS-BEC: Diatom-dominated phytoplankton community vs. mixed communities with significant contributions of coccolithophores and small phytoplankton. Communities in which neither coccolithophores (C) contribute >20% (blue) nor diatoms (D) >80% (red) to total annual NPP are classified as mixed communities (grey). b)-c) Annual mean most limiting nutrient for b) coccolithophore and c) diatom growth rates at the surface. For small phytoplankton, the nutrient limitation pattern south of 40°S is generally the same as for coccolithophores (not shown).

Summer biomass distributions of coccolithophores and diatoms show distinct geographical patterns in the top 50 m of the water column in ROMS-BEC (Fig. 1c & e). Coccolithophore biomass is highest in a broad circumpolar band between 35-60°S with maximum concentrations of 2.8 mmol C m$^{-3}$ on the Patagonian Shelf and a rapid decline south of 60°S (Fig.





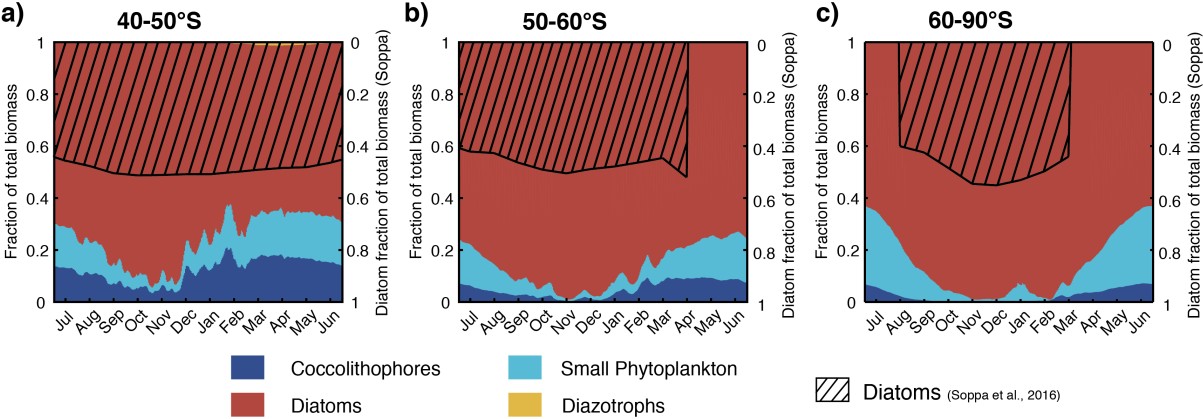

**Figure 3.** Relative contribution of the four phytoplankton PFTs to total surface chlorophyll biomass [mg chl m$^{-3}$] for a) 40-50°S, b) 50-60°S and c) south of 60°S. Shaded areas (right axis) depict the contribution of diatoms to total surface chlorophyll derived from monthly climatological MODIS-Aqua chlorophyll (Johnson et al., 2013) using the algorithm by Soppa et al. (2016). For months without shading, no satellite data were available.

1c & d). This pattern is broadly confirmed by observations: The latitudinal range of elevated coccolithophore biomass in the model agrees well with the observed location of the GCB (Balch et al., 2011), an area of elevated PIC levels between 40-60°S which has frequently been linked to high coccolithophore abundance (Trull et al., 2018; Balch et al., 2016; Saavedra-Pellitero et al., 2014; Poulton et al., 2013; Hinz et al., 2012). Maximum coccolithophore abundances in the upper 50 m of

the water column of up to ≈2500 cells ml$^{-1}$ (2.7 mmol C m$^{-3}$) have been reported for the Patagonian Shelf (Fig. 1c; Balch et al., 2016, biomass conversion following O'Brien et al., 2013). However, we find a systematic overestimation of simulated coccolithophore biomass north of ≈40°S and substantial scatter in the model-observation agreement (Fig. 1d & S4). The latter is expected when a model climatology is compared to in-situ observations, with an uncertainty of up to 400% due to the biomass conversion (see section S1).

In contrast to coccolithophores, diatoms biomass is highest south of 60°S with maximum concentrations of 17.2 mmol C m$^{-3}$ at 75°S in ROMS-BEC (top 50 m mean), and rapidly declines north of 60°S (Fig. 1e & f). Satellite derived diatom chlorophyll generally confirms this south-north gradient in diatom biomass (Soppa et al., 2014). Maximum summer in-situ biomass in the upper 50 m of the water column increases from 2.7 mmol C m$^{-3}$ north of 40°S to 13.6 mmol C m$^{-3}$ south of 60°S (Fig. 1e). Both in-situ observations (Fig. 1f) and satellite derived diatom chlorophyll (Soppa et al., 2014, comparison

not shown) suggest an overestimation of surface diatom biomass in ROMS-BEC south of 60°S during austral summer. Acknowledging the substantial uncertainty of the observational estimates (165% for the carbon biomass in Fig. 1f, on average at least 20% for satellite derived chlorophyll estimates in Soppa et al. (2014)), this overestimation might also be due to biases in the underlying physics (see section 4.1) or missing ecosystem complexity, due to either missing zooplankton trophic levels potentially contributing to the high total chlorophyll bias in the high latitudes in ROMS-BEC (Le Quéré et al., 2016) or missing

phytoplankton competitors. Locally, *Phaeocystis antarctica* can be as important as diatoms for total phytoplankton biomass





in SO waters (MAREDAT biomass data base: Vogt et al., 2012; Leblanc et al., 2012), and this PFT is not included in our simulations. Which proportion of the simulated diatom biomass bias has to be attributed to the missing competitor especially at the beginning of the growth season, will be part of future work.

CHEMTAX data (based on HPLC data) support the simulated gradient from a clearly diatom dominated community south
of 60°S to a more mixed community north thereof with a south-north increase of the coccolithophore contribution (Fig. 2a & Fig. 3) for the Western Atlantic sector of the SO (≈0% south of 60°S, up to 70% at around 40°S in fall, Swan et al., 2016) and for the Eastern Indian sector (<4% south of 60°S up to ≈18% at 40°S in summer, Takao et al., 2014). In available HPLC data for the SO, diatoms make up between 70-90% of the total summer phytoplankton chlorophyll biomass south of 60°S (Swan et al., 2016; Takao et al., 2014). In ROMS-BEC, the summer phytoplankton community south of 60°S is often almost solely
composed of diatoms (Fig. 2a & Fig. 3c).

In summary, ROMS-BEC reproduces spatial patterns of SO phytoplankton biomass and community composition. Summer coccolithophore biomass is highest north of 50°S, an area coinciding with the observed GCB, in which several PFTs coexist in our simulation, whereas diatom biomass peaks south of 60°S where they dominate the community. Observational data reveal phytoplankton community composition to not only vary in space, but also in time. In the following, we will assess
phytoplankton succession and the seasonal variability of phytoplankton community composition in the SO.

### 4.4 Bloom characteristics & seasonal succession

Generally, with increasing latitude, coccolithophore blooms in ROMS-BEC start and peak later (Fig. 4a & b) and the bloom amplitude decreases (Fig. 4c). Between 40-50°S, where their maximum in absolute biomass is located (up to 2.8 mmol C m$^{-3}$, Fig. 1c), coccolithophore blooms in ROMS-BEC start in week 20 (November) and peak in week 24 (December, at about 0.05
mg chl m$^{-3}$, Fig. 4a). Peak coccolithophore biomass thereby precedes the maximum contribution of coccolithophores (21%) to total surface phytoplankton biomass in early February (Fig. 3a). Between 50-60°S, coccolithophore blooms start in week 29 (January). Coccolithophores contribute up to 10% to total phytoplankton biomass in late February in our model (Fig. 3b), coinciding with peak absolute biomass of 0.015 mg chl m$^{-3}$ in week 35 (Fig. 4b).

As for coccolithophores, the diatom bloom onset and peak times are later at higher latitudes (Fig. 4a & b). However, in
contrast to coccolithophore blooms, the diatom bloom peak increases with latitude (Fig. 4c). Diatom blooms start in week 8 and 7 (August) and peak in week 18 and 20 (November, at 0.65 and 1.5 mg chl m$^{-3}$) between 40-50°S and 50-60°S, respectively (Fig. 4a & b). Thereby, diatom blooms precede coccolithophore blooms in ROMS-BEC. In our model, diatoms dominate total phytoplankton biomass everywhere south of 40°S (Fig. 2a) and diatoms therefore dominate total chlorophyll bloom dynamics.
Overall, the timing of blooms of coccolithophores agrees well with observations, but blooms of diatoms tend to start and peak too early and at too high chlorophyll concentrations in ROMS-BEC when compared to satellite estimates (especially south of 60°S, not shown). More specifically, PIC imagery (a proxy for coccolithophore abundance) suggests annual peak concentrations for December and January for 40-50°S and 50-60°S (NASA-OBPG, 2014c), comparing well with the simulated peaks in December and February in ROMS-BEC. Soppa et al. (2016) find diatom biomass to peak around mid-December (40-

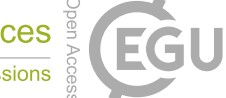

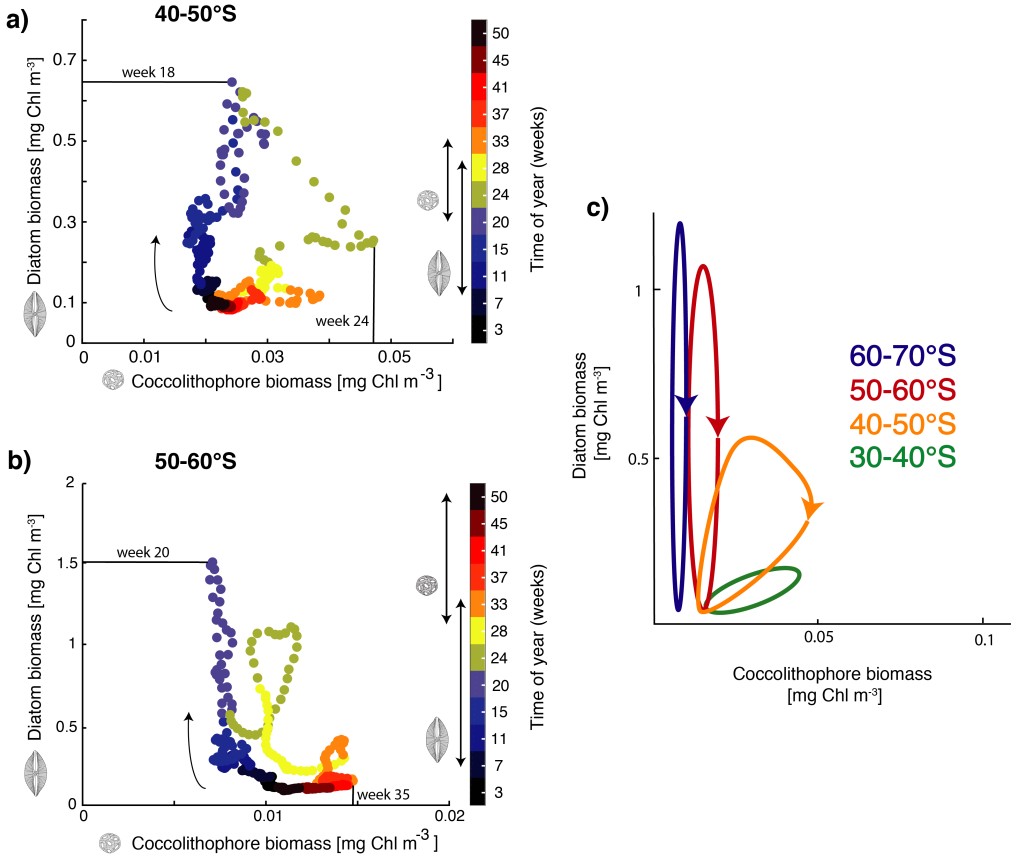

**Figure 4.** Phase diagram of daily surface diatom and coccolithophore chlorophyll biomass [mg chl m$^{-3}$] for a) 40-50°S and b) 50-60°S. The colors indicate the time of the year (given in weeks) and the arrow indicates the course of time. Bloom start, bloom end, and bloom duration are marked with arrows on the colorbar showing time evolution from July-June for diatoms and coccolithophores, and bloom peak is drawn directly into the phase diagram. c) Sketch of diatom and coccolithophore chlorophyll biomass evolution [mg chl m$^{-3}$] for the different latitudinal bands. Lowest biomass in bottom left, arrows indicates temporal evolution. For details on the definition of the bloom metrics, see the supplementary material.

60°S) and between mid-January and mid-February south of 60°S, about 1-2 months later than in our simulation. Additionally, while the simulated peak diatom chlorophyll biomass is close to the value suggested by Soppa et al. (2016) for 40-60°S (0.27 vs. 0.25 mg chl m$^{-3}$), the simulated peak diatom chlorophyll biomass is fivefold higher south of 60°S (not shown).

Despite these discrepancies, the simulated succession pattern of diatoms and coccolithophores agrees with that suggested for
5   the GCB. In-situ studies for the GCB area have inferred the succession of diatoms by coccolithophores from depleted silicate levels coinciding with high coccolithophore abundance between 40°S-65°S, especially for the Patagonian Shelf (Balch et al., 2016, 2014; Painter et al., 2010), supporting the seasonal dynamics simulated by ROMS-BEC. In the following sections, we assess the controlling factors of the simulated spatial and temporal variability, with a particular focus on the biogeography of



coccolithophores and their interplay with diatoms. For this, we restrict the discussion to the latitudinal bands between 40-50°S and 50-60°S, where coccolithophore biomass is highest (see section 4.3).

## 4.5 Bottom-up controls on coccolithophore biogeography

Phytoplankton growth rates in BEC are determined as a function of the a maximum growth rate and surrounding environmental
conditions with respect to temperature, nutrient and light levels (Eq. 3). Here, we use the relative growth ratio of diatoms versus coccolithophores as defined in Eq. 4 (Hashioka et al., 2013) in order to disentangle the effect of individual bottom-up factors on diatom-coccolithophore competition and their relative contribution to total surface phytoplankton biomass.

In ROMS-BEC, the latitudinal band between 40-50°S is the area with the highest coccolithophore biomass in austral summer (see Fig. 1d and 4). The relative growth ratio of diatoms vs. coccolithophores between 40-50°S (solid black line in Fig. 5a)
is negative from the end of September until the end of April ($\mu^{\mathrm{Cocco}} > \mu^{\mathrm{Diatoms}}$). For the summer months (December-March, DJFM), the specific growth rate of coccolithophores is on average 10% larger than that of diatoms (Fig. 6a, shaded dark grey bar, calculated from non-log transformed ratios), favoring the buildup of coccolithophore relative to diatom biomass. In comparison, the relative growth ratio of diatoms vs. coccolithophores between 50-60°S (solid black line in Fig. 5b) is negative for a shorter time period (December until mid-February). The specific growth rate of coccolithophores is only 2% larger than
that of diatoms in summer (Fig. 6b, shaded dark grey bar), making it harder for coccolithophores to build up biomass relative to diatoms between 50-60°S as compared to 40-50°S.

The relative growth ratio can be separated into the contribution of the maximum growth rate $\mu_{\max}$, temperature, nutrients and light, which all affect phytoplankton growth (Eq. 4, colored areas in Fig. 5a & b and Fig. 6). The 21% larger $\mu_{\max}$ of diatoms compared to that of coccolithophores favors diatom relative to coccolithophore growth all year round in the whole
model domain (Table 1, green area in both Fig. 5a & b is positive). Differences in the temperature limitation of diatoms and coccolithophores arise from differences in $Q_{10}$ of each PFT (Eq. B5), with coccolithophores being less temperature limited than diatoms (Table 1, red area in Fig. 5a is negative). This leads to a DJFM mean growth advantage of 11% and 15% of coccolithophores relative to diatoms for 40-50°S and 50-60°S respectively (Fig. 6, shaded red bars).

Due to their lower half-saturation constants for nutrient uptake (Table 1), coccolithophores are less nutrient limited than
diatoms, resulting in the negative blue areas in Fig. 5a & b (19% and 7% less nutrient limited for DJFM between 40-50°S and 50-60°S respectively, see Fig. 6, shaded blue bars). For the summer months, amongst all environmental factors, this is the biggest difference simulated between the two latitudinal bands (compare shaded colored bars between Fig. 6a & b). The spatial pattern of the most limiting nutrient for coccolithophore and diatom growth in ROMS-BEC (Fig. 2b & c respectively) provides the explanation for this: Between 50-60°S, iron is the most limiting nutrient for both PFTs, but silicate is the most
limiting nutrient for diatom growth between 40-50°S. While coccolithophores remain iron limited, silicate limitation of diatoms increases the difference in nutrient limitation between coccolithophores and diatoms, thus explaining the greater advantage for coccolithophores between 40-50°S as compared to 50-60°S.

In ROMS-BEC, differences in light limitation between two PFTs are controlled by differences in the sensitivity to increases of PAR at low irradiances ($\alpha_{\mathrm{PI}}$) and differences in photoacclimation, i.e. the ability of each PFT to adjust its chlorophyll-to-





**Figure 5.** (a)-(b) Relative growth ratio (solid black line) and relative grazing ratio (dashed black line) of diatoms vs. coccolithophores. Colored areas are contributions of the maximum growth rate $\mu_{\max}$ (green), nutrient limitation (blue), light limitation (yellow) and temperature sensitivity (red) to the relative growth ratio. See section 3 for definition of metrics. (c)-(d) Surface carbon biomass evolution [mmol C m$^{-3}$], (e)-(f) specific growth rates ([d$^{-1}$], Eq. 3), and (g)-(h) clearance rates ([d$^{-1}$], Eq. 7). For (c)-(h), coccolithophores (C) are shown in blue, diatoms (D) in red, and small phytoplankton (SP) in green. For all metrics, left panels are for 40-50°S, those on the right for 50-60°S.





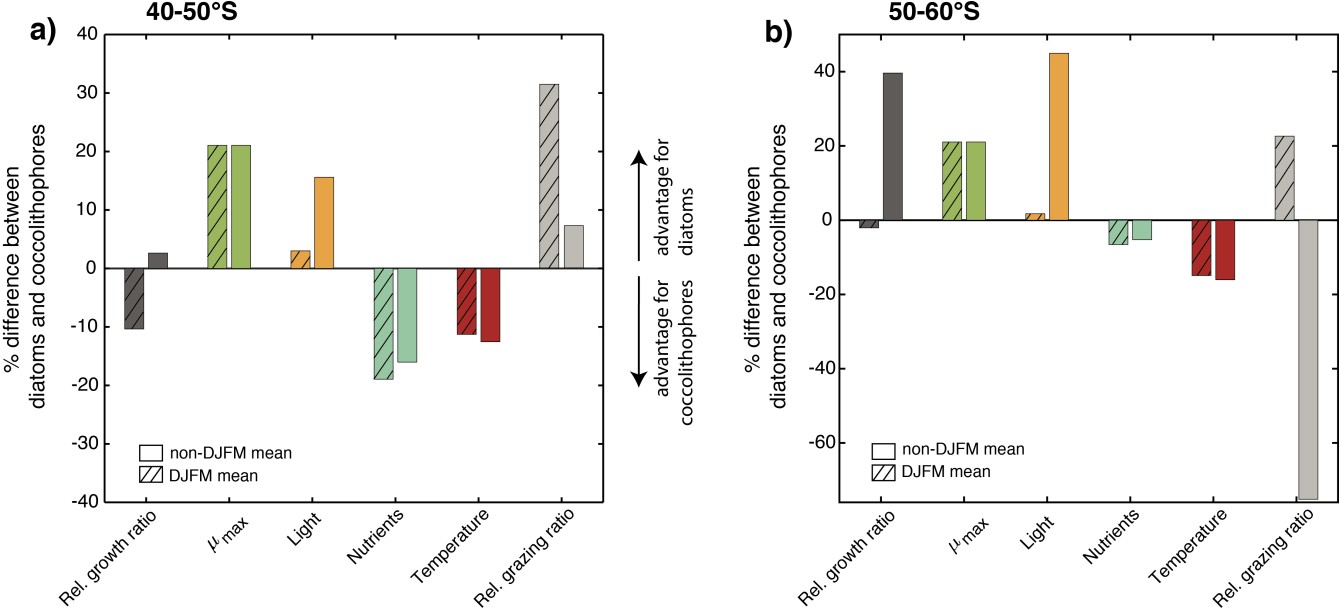

**Figure 6.** Percent difference in growth rate (dark grey), growth-limiting factors (maximum growth rate $\mu_{\mathrm{max}}$ in green, nutrient limitation in blue, light limitation in yellow and temperature sensitivity in red) and grazing rate (light grey) of diatoms and coccolithophores for a) 40-50°S and b) 50-60°S. Respective left bar shows the December-March average (DJFM) calculated from the non-log transformed ratios, the shaded right bars show the average for all other months (non-DJFM). Full seasonal cycle is shown in Fig. 5a & b.

carbon ratio to surrounding light, nutrient and temperature conditions (Eq. B9, Geider et al., 1998). Coccolithophores have a lower $\alpha_{\mathrm{PI}}$ (Table 1), a generally lower chlorophyll-to-carbon ratio (not shown) and are less nutrient limited than diatoms (blue areas in Fig. 5a & b), all together resulting in a stronger light limitation of coccolithophores compared to diatoms. In austral summer, coccolithophores are on average 2% and 3% more light limited than diatoms between 40-50°S and 50-60°S respectively (Fig. 6, shaded yellow bars). While light appears to affect the relative importance of coccolithophores and diatoms almost equally during the summer months for the two latitudinal bands, unsurprisingly, the model simulates pronounced differences between the two latitudinal bands throughout the rest of the year (16% and 45% for non-DJFM mean between 40-50°S and 50-60°S respectively, Fig. 6). Compared to 40-50°S, light levels are generally lower between 50-60°S, resulting in a larger growth advantage for diatoms partly due to their higher $\alpha_{\mathrm{PI}}$ and mainly due to their higher chlorophyll-to-carbon ratio (see also Fig. S5), allowing them to use low irradiances more efficiently.

Coccolithophores and diatoms together contribute on average 87% and 95% to total DJFM mean surface phytoplankton biomass between 40-50°S and 50-60°S, respectively (Fig. 3). They are thus not only competing for resources between each other, but with SP as well. SP biomass largely covaries with coccolithophore biomass between 40-50°S (Fig. 5c), but coccolithophores outcompete SP in summer due to their higher maximum growth rate (Table 1) and growth advantages with respect to temperature, outweighing disadvantages with respect to light and nutrients (Fig. S6A & S7A). Between 50-60°S, SP biomass is





higher than coccolithophore biomass for most of the year (Fig. 5d). Similarly to the diatom-coccolithophore interplay, coccolithophores have a growth advantage relative to SP for a smaller time period (mid-November until April as compared to August until mid-May, Fig. S6), while it is slightly bigger in amplitude in summer for this latitudinal band as compared to 40-50°S (6% as compared to 4%, Fig. S7B).

In summary, coccolithophores have an advantage in specific growth relative to diatoms in austral summer both between 40-50°S and 50-60°S. Comparing the two latitudinal bands, this advantage is higher for 40-50°S, explaining the greater importance of coccolithophores for total phytoplankton biomass in this band as compared to 50-60°S (Fig. 3). Comparing all environmental factors and the two latitudinal bands, nutrient conditions control the difference in total relative growth ratio between 40-50°S and 50-60°S in summer, while differences in light limitation drive differences between the summer months and the rest of the

year (DJFM vs. non-DJFM, Fig. 6). However, both for 40-50°S and 50-60°S, despite the higher specific growth rate for part of the year, coccolithophores never outcompete diatoms in terms of absolute biomass (Fig. 5c & d). We calculated whether the length of the growing season is long enough for coccolithophores to outcompete diatoms, given their biomass ratio at the end of November, as well as the DJFM growth advantage of 2%/10% (40-50°S and 50-60°S respectively, Fig. 6) for coccolithophores, assuming no difference in loss rates between the two PFTs. We found that for 50-60°S, the growth advantage of 2% is not large

enough to result in a dominance of coccolithophores over diatoms at the end of the growth season, given the 45 times higher diatom biomass at the end of November, in agreement with the simulated biomass evolution (Fig 5d). For 40-50°S, however, our calculations show that despite the 13 times higher biomass of diatoms at the end of November (Fig. 5c), coccolithophores should outcompete diatoms at the end of March with a 10% higher specific growth rate if loss rates are the same for both PFTs. While the analysis of bottom-up factors can explain the increase in the relative importance of coccolithophores in late

austral summer as well as the higher relative importance of coccolithophores between 40-50°S than 50-60°S, it fails to explain the magnitude of absolute biomass concentrations for 40-50°S. Absolute biomass of each PFT (and therefore the relative importance of coccolithophores and diatoms) is not only controlled by bottom-up factors described in this section, but by top-down factors as well (Eq. 2). In the following, we will assess the importance of grazing by zooplankton in ROMS-BEC for the relative importance of coccolithophores between 40-60°S.

**4.6   Top-down controls on coccolithophore biogeography**

In ROMS-BEC, the relative grazing ratio (see Eq. 6 and Hashioka et al., 2013) of diatoms vs. coccolithophores between 40-50°S (dashed black line in Fig. 5a) is positive from mid-September until the end of April ($\gamma_g^C/P^C > \gamma_g^D/P^D$). For the summer months (DJFM), the specific grazing rate on coccolithophores is on average 31% larger than that on diatoms (Fig. 6a, shaded light grey bar), favoring the buildup of diatom relative to coccolithophore biomass. In comparison, the relative grazing ratio of

diatoms vs. coccolithophores between 50-60°S (dashed black line in Fig. 5b) is positive for a shorter time period (November until end of March) and the specific grazing rate on coccolithophores is 23% larger than that of diatoms in summer (Fig. 6b, shaded light grey bar).

    Overall, differences in specific grazing rates between coccolithophores and diatoms are of similar magnitude as difference in specific growth rates (same scale for solid and dashed lines in Fig. 5a & b), implying that top-down factors are as important





as bottom-up factors in controlling the relative importance of coccolithophores and diatoms between 40-60°S. In fact, during the summer months, differences in specific grazing rates are three and 11 times larger than differences in specific growth rates for 40-50°S and 50-60°S, respectively (Fig. 6), implying that top-down factors outweigh bottom-up factors for this time of the year.

5     Periods of positive relative grazing ratios almost exactly overlap with periods of negative relative growth ratios for both 40-50°S and 50-60°S (compare solid and dashed black line in Fig. 5a & b). During these times, coccolithophores experience a larger per biomass grazing pressure than diatoms, making it harder for them to use their advantage in specific growth rate and to build up biomass relative to diatoms. More specifically, what matters for phytoplankton biomass accumulation rates is the clearance rate (Eq. 7). Due to the higher $\gamma_{\max}$ associated with grazing on coccolithophores as compared to diatoms (Table 1),

10 clearance rates for coccolithophores are higher than those for diatoms for both 40-50°S and 50-60°S in summer (Fig. 5g & h), resulting in slower biomass accumulation rates for coccolithophores.

    In summary, in ROMS-BEC, lower clearance rates make diatoms more successful than coccolithophores in accumulating and sustaining higher biomass concentrations, resulting from a higher per biomass grazing pressure on coccolithophores as compared to that on diatoms between 40-60°S. Our findings show that in ROMS-BEC, top-down factors are as important

15 as bottom-up factors in controlling the relative importance of coccolithophores and diatoms between 40-60°S, as bottom-up factors alone cannot explain the simulated biomass distributions.

## 4.7   Parameter sensitivity simulations

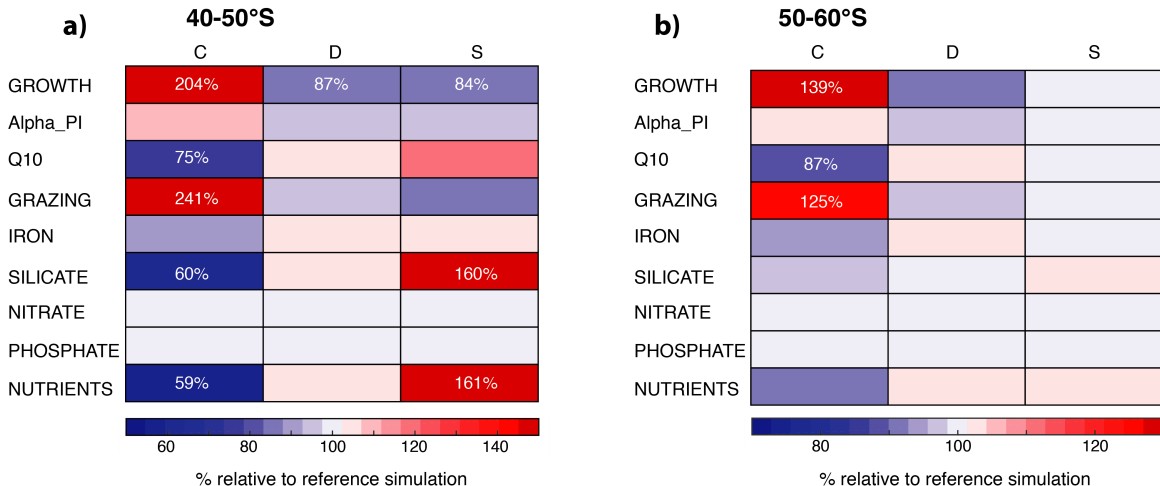

**Figure 7.** Relative change in annual mean surface chlorophyll biomass of coccolithophores (C), diatoms (D), and small phytoplankton (SP) for a) 40-50°S and b) 50-60°S for simulations assessing coccolithophore parameter sensitivities (see Table 2). Numbers of relative change are printed if change is larger than ±10%.





We assessed the sensitivity of the simulated coccolithophore biogeography by performing a set of sensitivity simulations (runs 1-9 in Table 2). Annual mean surface coccolithophore biomass between 40-60°S shows strongest increases for GROWTH (doubling and 33% increase as compared to reference simulation for 40-50°S and 50-60°S, Fig. 7) and GRAZING (2.5 times and 25% increase), supporting our finding from sections 4.5 and 4.6 that top-down and bottom-up controls are equally important

in controlling SO coccolithophore biogeography. Coccolithophore biomass decreases by 25% and 13% for 40-50°S and 50-60°S, respectively (with changes <10% in diatom and SP biomass), when making coccolithophore growth more temperature limited (Q10, Fig. 7). With respect to nutrient sensitivities, only the simulation SILICATE leads to significant changes in annual mean coccolithophore biomass for 40-50°S (decrease of 40%, which is compensated by an increase in SP biomass of 60%). Between 50-60°S, none of the simulations assessing nutrient sensitivities (runs 5-9) results in significant biomass changes (Fig.

7), confirming the minor importance of the nutrient affinity of coccolithophores and diatoms for their relative importance in this area (blue bars in Fig. 6b). Lastly, coccolithophore biogeography shows little sensitivity to the chosen $\alpha_{\mathrm{PI}}$. This confirms the result from section 4.5, namely that differences between coccolithophores and diatoms in light limitation are not driven by differences in $\alpha_{\mathrm{PI}}$ (Fig. S5). In summary, we conclude that the simulated coccolithophore biogeography is especially sensitive to the chosen maximum growth and grazing rate ($\mu_{\mathrm{max}}$ and $\gamma_{\mathrm{max}}$, Table 1), while it appears insensitive to $\alpha_{\mathrm{PI}}$ and all nutrient

half-saturation constants, except with respect to silicate limitation of diatoms.

## 5    Discussion

### 5.1    Biogeochemical implications of SO coccolithophore biogeography

In ROMS-BEC, coccolithophores are a non-negligible part of the SO phytoplankton community, contributing 14.7% to total annual NPP south of 30°S. Comparing our model results to previously published global estimates, SO coccolithophores

contribute 3.5-4.5% to global NPP ($58\pm7$ Pg C yr$^{-1}$, Buitenhuis et al., 2013a). Our model results suggest that globally, the contribution of coccolithophores to NPP is larger than previously estimated (<2%, Jin et al., 2006; 0.4%, O'Brien 2015). The modeled coccolithophore biomass between 30-40°S, an area contributing >50% to coccolithophore production and biomass south of 30°S in our model (Table 3), is likely an overestimate (Fig. 1d). At the same time, south of 40°S, coccolithophore biomass is underestimated in the model compared to in-situ observations (Fig. 1d), at least partly balancing the overestimation

in the north of the domain. Overall, the scarcity of the in-situ data, as well as their high uncertainty of up to 400% (resulting from the biomass conversion from cell counts (O'Brien et al., 2013)) have to be acknowledged, making it difficult to evaluate our model estimate. In addition, simulated coccolithophore biomass and production are prone to uncertainty arising from the chosen parameters, and integrated coccolithophore production south of 30°S varies from 1.8-3.8 Pg C yr$^{-1}$ (2.8-7.5% of global NPP) in our parameter sensitivity simulations (runs 1-8, except run 6, Table 2).

Contributing only a few percent to global NPP, coccolithophores appear to be of minor importance for global oceanic organic carbon fixation. However, coccolithophores impact the carbon cycle more significantly by inorganic carbon production (calcification). Our results suggest that SO coccolithophore calcification contributes ≈19% to global coccolithophore calcification derived from remote sensing imagery (8.1-35.4% if accounting for uncertainty in CaCO$_3$:C$_{\mathrm{org}}$ production ratio of SO



*Emiliania huxleyi*, Krumhardt et al. (2017)). Between 40-60°S (GCB area, area of highest coccolithophore biomass concentrations in both model and observations), the model simulates 7.5% (3.8-10.8%) of global calcification, which is lower than the satellite derived estimate of 18.8% (15.2-22.3%). However, in BEC, we model the rather lightly calcified SO *Emiliania huxleyi* B/C morphotype (Krumhardt et al., 2017). While *Emiliania huxleyi* in general, and this morphotype in particular, have

been shown to dominate the coccolithophore community in the SO (Saavedra-Pellitero et al., 2014; Balch et al., 2016; Smith et al., 2017), other species such as the more heavily calcified *Emiliania huxleyi* morphotype A or *C. leptoporus* might locally contribute overproportionally to total calcification, contributing to the underestimation of modeled calcification. *C. leptoporus* has been found to locally dominate the coccolithophore community (67.6% of the community at a station in the Pacific sector, Saavedra-Pellitero et al., 2014) and has a generally higher $CaCO_3:C_{org}$ production ratio than *Emiliania huxleyi* B/C (0.4-3.2

Krumhardt et al., 2017). Keeping this uncertainty in mind, we can conclude from our simulation that coccolithophores in the GCB are likely at least as important as the surface area they cover (10.9% of global ocean area, 40-60°S), making them an important contributor to the global carbon cycle, despite their relatively small contribution to global NPP.

In the context of carbon sequestration, the PIC:POC export ratio is crucial. In ROMS-BEC, we find that the PIC:POC export ratio is higher where and when coccolithophores are important (30-60°S, Table 3, especially on the Patagonian Shelf,

not shown), in agreement with in-situ observations by Balch et al. (2016). A higher PIC:POC export ratio possibly enables more $CO_2$ uptake due to the ballasting effect of calcite for downward transport of organic carbon. At the same time, calcification directly increases seawater $pCO_2$, counteracting the ballasting effect. Balch et al. (2016) found that the abundance of coccolithophores in the GCB is high enough to temporarily and locally reverse the sign of the air-sea $CO_2$ flux from a sink to neutral or even a source, inhibiting further $CO_2$ uptake. The net sign of the combined effect of ballasting and the direct

calcification effect on air-sea $CO_2$ exchange remains to be quantified for the GCB as a whole in future research. Nevertheless, the relative importance of coccolithophores in ROMS-BEC implies that it is crucial to estimate potential future change in the relative importance of coccolithophores and/or $CaCO_3:C_{org}$ production ratio of coccolithophores for estimating future oceanic carbon cycling in this area in general, and oceanic $CO_2$ uptake in particular.

## 5.2 Succession vs. coexistence: Decoupling of maximum specific growth rate and maximum biomass levels by

**zooplankton grazing in ROMS-BEC**

In ROMS-BEC, coccolithophore blooms start and peak later than those of diatoms between 40-60°S (Fig. 4), in agreement with Margalef's mandala predicting the succession of these phytoplankton functional types as a result of changing environmental conditions over time (Margalef, 1978). At the same time, the specific growth rate of coccolithophores in ROMS-BEC is higher than that of diatoms for much of the year (40-50°S) and most of austral summer (50-60°S), respectively (5e & f), implying

that the relative importance of coccolithophores and diatoms and the timing of their peak biomass are not purely controlled by environmental conditions. In fact, phytoplankton specific growth rates are not largest when the respective biomass level is at its maximum in our model (compare Fig. 5c & d with Fig. 5e & f), implying a decoupling between simulated environmental conditions and biomass peaks. This suggests that top-down processes, such as grazing by zooplankton, are non-negligible in defining patterns of phytoplankton succession and coexistence.



Several metrics have been applied in the past to assess the question of coexistence vs. succession of two phytoplankton PFTs in general, or of diatoms and coccolithophores in particular. Traditionally, studies have looked at absolute biomass concentrations only, and defined coexistence/succession based on a temporal separation in biomass peaks. For example, Hopkins et al. (2015) defined succession of diatoms and coccolithophores whenever peaks of total chlorophyll and PIC were more than 16 days apart and identified most of 40-60°S as a coexistence area. Instead, Barber and Hiscock (2006) analyzed specific growth rates rather than absolute biomass concentrations. Based on JGOFS data from the equatorial Pacific, their study suggests that all phytoplankton profit equally from improved environmental conditions, and that differences in timing of the biomass peaks can also result simply from differences in the relative abundance at the beginning of the growth season and varying grazing pressures. In agreement with this, Daniels et al. (2015) found coccolithophores to grow simultaneously with an observed diatom bloom in the North Atlantic, instead of simply succeeding it.

In agreement with Barber and Hiscock (2006) and Daniels et al. (2015), all phytoplankton respond with an increase in their specific growth rate to improving environmental conditions in spring in ROMS-BEC (Fig. 5e & f), while biomass peaks of e.g. diatoms and coccolithophores are clearly separated in time because grazing by zooplankton is crucial in controlling biomass evolution in our simulation (see section 4.6). Since maximum specific growth rates, i.e. ideal environmental conditions, do not imply concurrent maximum biomass concentrations in our simulation, the timing of maximum biomass concentrations similarly does not imply ideal growth conditions at that time. This has implications for both in-situ and remote sensing based studies: Typically, in-situ studies relate high phytoplankton abundance to local environmental conditions to infer ideal growth conditions. Our results suggest that environmental conditions at the time of maximum abundance do not necessarily represent ideal growth conditions and that a decoupling of specific growth rate and biomass levels as a result of e.g. top-down controls result in an identification of succession of phytoplankton types in terms of biomass peaks that is not purely bottom-up driven. Simply comparing peak biomass levels of two PFTs, as typically done in remote sensing studies assessing phytoplankton seasonality (e.g. Hopkins et al., 2015), might similarly result in a misleading picture of ecosystem dynamics and patterns of succession and coexistence. Therefore, assessing remote sensing data with a metric focusing on the relative increase in biomass during the "pre-peak" period rather than just the biomass peak itself might reveal different patterns of coexistence and succession between 40-60°S, possibly revealing areas of a decoupling between maximum biomass and maximum growth rate. This might reconcile the different metrics and methods used to assess phytoplankton seasonality and give a more comprehensive picture of the interplay of bottom-up and top-down controls.

## 5.3 Drivers of coccolithophore biogeography

In ROMS-BEC, absolute biomass concentrations over the course of the year as well as the relative importance of coccolithophores and diatoms between 40-60°S are controlled by the spatial and temporal variability in silicate and light availability, as well as the higher per biomass grazing pressure on coccolithophores than on diatoms (Fig. 8). A number of in-situ studies found an anticorrelation between *Emiliania huxleyi* abundance in the SO and local silicate concentrations (Smith et al., 2017; Balch et al., 2014; Mohan et al., 2008; Hinz et al., 2012). In addition, Balch et al. (2016) found *Emiliania huxleyi* to be positively correlated with in-situ iron levels, concluding that this species occupies the highFe-lowSi niche. This is in agreement





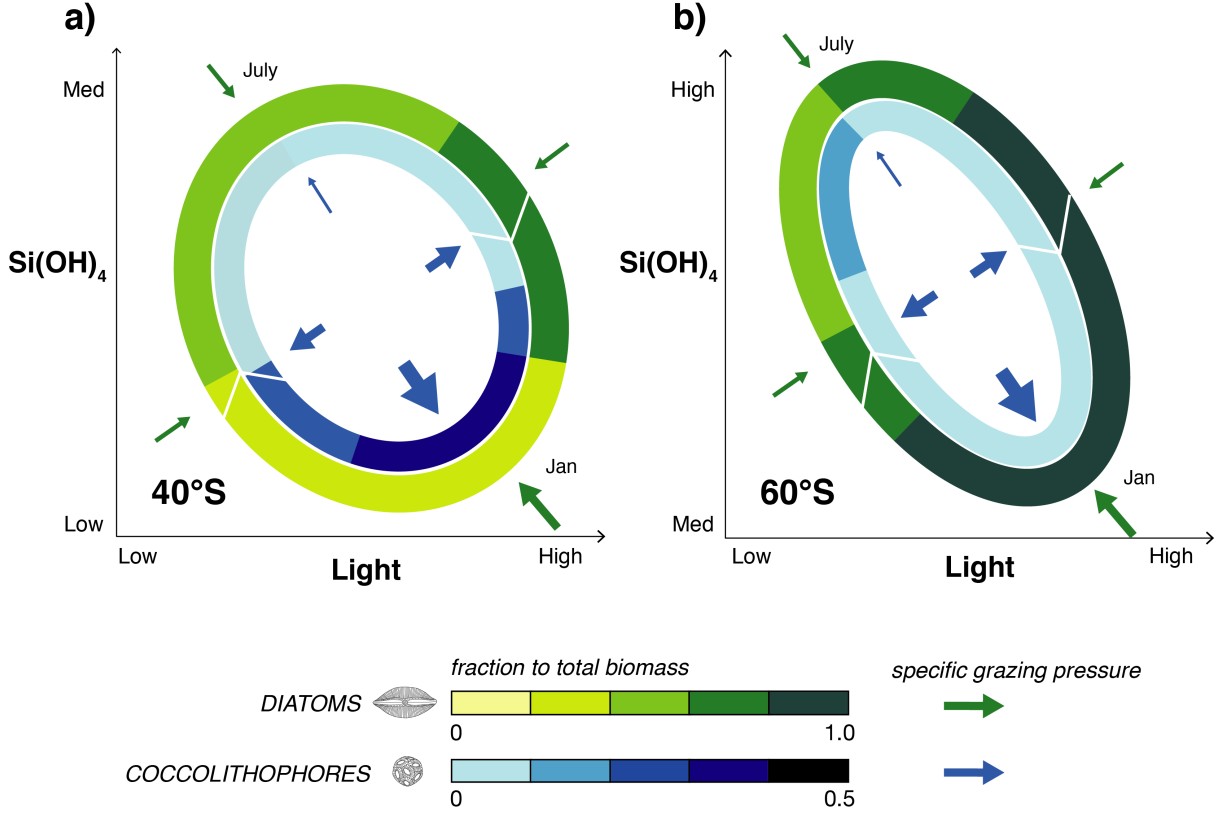

**Figure 8.** Sketch summarizing the results from ROMS-BEC: Relative importance of coccolithophores (inner circle) and diatoms (outer circle) for total phytoplankton biomass over time in light-silicate space for a) 40°S and b) 60°S. Note the different scales for coccolithophores and diatoms. Arrows in the sketch indicate the course of time (white) and the strength of the specific grazing pressure on coccolithophores (blue) and diatoms (green).

with our model results, where coccolithophores are most important where (40-50°S) and when (late austral summer) diatoms become silicate limited, but iron levels are still high enough to sustain coccolithophore growth. In contrast to most other phytoplankton, laboratory experiments have shown coccolithophore growth not to be inhibited at high light levels (photoinhibition, Zondervan, 2007), and high light levels have therefore often been considered a prerequisite for elevated coccolithophore abun-

5 dance (Charalampopoulou et al., 2016; Balch et al., 2014; Poulton et al., 2013; Balch, 2004). In our model, we do not consider the effects of photoinhibition for any of the phytoplankton PFTs. In BEC, differences in summer light levels between 40-50°S and 50-60°S cannot explain why relatively, coccolithophores are more important between 40-50°S than 50-60°S (1% difference of shaded yellow bar in 6a & b) and differences in the seasonal amplitude of light levels between the two latitudinal bands appear more important than latitudinal differences in summer alone. If photoinhibitory effects were included in our model,

10 we expect coccolithophores to increase in relative importance in the whole model domain, especially towards the end of the growth season, when light levels are highest.



Besides bottom-up factors, we find grazing by zooplankton to be key in explaining the seasonal evolution of the phytoplankton community structure between 40-60°S in our model. BEC includes a single zooplankton PFT, comprising characteristics of both micro- and macrozooplankton (by assuming microzooplankton feeding on SP and coccolithophores to grow faster than macrozooplankton feeding on diatoms, compare $\gamma_{max}$ in Table 1, Moore et al. (2002); Sailley et al. (2013)), thereby emulating

two trophic levels within the zooplankton compartment without explicitly modeling them. However, Sailley et al. (2013) found the coupling between each phytoplankton PFT and the single zooplankton PFT to be strong in BEC, meaning that any increase in phytoplankton biomass leads to a concurrent and immediate increase in zooplankton biomass until saturation is reached. This tight coupling prevents any phytoplankton PFT from escaping grazing pressure and making use of favorable growth conditions, as seen for coccolithophores between 40-60°S. Additional explicit zooplankton PFTs and an explicit representation of trophic

cascades in the zooplankton compartment might decouple phytoplankton and grazer biomass in both space and time, fostering the importance of coccolithophores relative to diatoms between 40-60°S and possibly altering total phytoplankton biomass (Le Quéré et al., 2016). The tight coupling between phytoplankton and the single zooplankton in BEC suggests a likely overestimation of the importance of top-down control in controlling the relative importance of coccolithophores in the SO, as compared to models with more zooplankton complexity. Zooplankton grazing has been shown to influence SO phytoplankton biomass (Le

Quéré et al., 2016; Painter et al., 2010; Garcia et al., 2008) and community composition (e.g. Smetacek et al., 2004; Granéli et al., 1993; De Baar, 2005), but its possible role for SO coccolithophore biogeography has not yet been addressed. Selective grazing by microzooplankton has been found to be important for the development of coccolithophore blooms in other parts of the ocean in observational (North Sea: Holligan et al. (1993), Devon coast: Fileman et al. (2002), northern North Sea: Archer et al. (2001)) and modeling studies (Bering Sea Shelf: Merico et al. (2004)). However, recent in-situ studies addressing controls

on coccolithophore biogeography in the SO (e.g. Balch et al., 2016; Charalampopoulou et al., 2016; Saavedra-Pellitero et al., 2014; Hinz et al., 2012) have exclusively focused on bottom-up controls by correlating high coccolithophore abundance with concurrent environmental conditions. Based on our findings, future SO in-situ studies should consider both bottom-up and top-down factors when assessing coccolithophore biogeography in space and time.

## 5.4   Limitations and caveats

Our findings may be impacted by several limitations regarding ecosystem complexity, chosen parametrizations and parameters in BEC, model setup and performance, as well as the analysis framework. Ecosystem models do not only vary in the number of zooplankton PFTs, but also in the chosen grazing formulation (Sailley et al., 2013), e.g. in their functional response regarding the ingestion of prey (e.g. Holling Type II vs. Holling Type III, Holling, 1959) or in the prey preferences of each predator (variable or fixed). It has been shown previously in global models that the choice of the grazing formulation impacts phytoplankton

biogeography and diversity (e.g. Prowe et al., 2012; Vallina et al., 2014). In ROMS-BEC, we found the effect of both a Holling Type III and active prey switching on our results to be similar (run 10 & 11 in Table 2): Overall, the chosen grazing functional response impacts the simulated phytoplankton biogeography and seasonality, but to a lesser extent the relative importance of each PFT for total NPP and the ranking of controlling factors of coccolithophore biogeography (Fig. S8). Using Holling Type III instead of Holling Type II (same as for active prey switching instead of fixed prey preferences) leads to increased coexis-





tence in the phytoplankton community by reducing grazing pressure on less abundant PFTs. For both sensitivity simulations, both coccolithophores and SP increase in importance relative to diatoms (contribution to total NPP south of 30°S: 31%/35.2% SP, 22%/18.3% coccolithophores and 44.3%/44.5% diatoms with Holling Type III and active prey switching respectively, as compared to 19.7%, 14.7% and 64.6% in the reference run). The chosen grazing formulation in ROMS-BEC appears to quan-

titatively impact our results, but does not qualitatively change the importance of top-down factors. This finding is in agreement with previous modeling studies, which despite using different ecosystem complexity and grazing formulations, came to the conclusion that top-down control is of vital importance for phytoplankton biogeography and diversity (Sailley et al., 2013; Vallina et al., 2014; Prowe et al., 2012).

Phytoplankton biogeography is not only affected by choices regarding ecosystem complexity and parameters, but by biases

in the underlying physical and biogeochemical fields as well. In summary, both the temperature and ML bias have little effect on phytoplankton biogeography, and both phytoplankton community composition and the relative importance of the controls for coccolithophore biogeography change only slightly compared to the reference simulation (run 12 & 13 in Table 2, Fig. S8 & S9; contribution to total NPP south of 30°S: 19.4%/18.1% SP, 14.9%/13.5% coccolithophores and 64.7%/67.2% diatoms for TEMP and MLD, respectively, as compared to 19.7%, 14.7% and 64.6% in the reference run). In addition, neither the bias in

temperature nor in MLD can explain the overestimation of NPP and total surface chlorophyll at latitudes >60°S (not shown). We conclude that biases in the physical fields do not significantly impact our results. However, the positive bias of NPP/total surface chlorophyll remains unexplained in ROMS-BEC at this point. A previous modeling study by Le Quéré et al. (2016) has shown missing complexity in the zooplankton compartment to be a possible explanation for simulated positive phytoplankton biomass biases in the high latitude SO, and the role of multiple trophic levels needs to be explored in ROMS-BEC.

In this study, we only present results for latitudinal averages even though coccolithophore biomass and their relative importance for total phytoplankton biomass varies across basins (see Fig. 1, as well as Balch et al. (2016)). Additionally, we only address differences in grazing pressure between two phytoplankton PFTs in this study. Aggregation losses and non-grazing mortality (see Eq. 2) contribute <10% to total phytoplankton loss between 40-60°S on average (not shown), suggesting them to be of minor importance in controlling the relative importance of coccolithophores and diatoms in this area. Ultimately, coc-

colithophore growth and calcification in BEC are currently not dependent on ambient $CO_2$ concentrations. However, both the study by Trull et al. (2018) and the review by Krumhardt et al. (2017) suggest carbonate chemistry to be of minor importance in controlling the relative importance of coccolithophores in the SO at present, as both specific growth rates and $CaCO_3$:$C_{org}$ production ratios of SO coccolithophores appear rather insensitive to variations in ambient $CO_2$ (Krumhardt et al., 2017). Concurrently, the $CaCO_3$:$C_{org}$ production ratio has been shown to depend on surrounding temperature, light and nutrient levels.

However, for SO coccolithophores, data are scarce and the resulting functional dependencies remain unclear (Krumhardt et al., 2017). We thus cannot estimate the effect of a varying $CaCO_3$:$C_{org}$ production ratio on our results.





## 6    Conclusions

This modeling study is the first to comprehensively assess the importance of both bottom-up and top-down factors in controlling the relative importance of coccolithophores and diatoms in the SO over a complete annual cycle. We find that coccolithophores contribute 15% to total annual NPP south of 30°S in ROMS-BEC, making them an important member of the SO phytoplankton

community. Based on our results, SO coccolithophores alone contribute 4% to global NPP. We therefore recommend the inclusion of an explicit coccolithophore PFT in global ecosystem models, and the development of existing implementations (Le Quéré et al., 2016; Kvale et al., 2014; Gregg and Casey, 2007a), to more adequately simulate both tropical and subpolar coccolithophore populations, and to better constrain their contribution to global NPP.

In our model, coccolithophore biomass is high when diatoms are most limited by silicate and when light levels are highest,

i.e., north of 50°S and towards the end of the growing season. Yet the coccolithophore biomass never exceeds that of the diatoms. This is a consequence of top-down control, i.e., the fact that the coccolithophores are subject to a much larger biomass-specific grazing pressure than the diatoms. Consequently, both abiotic and biotic interactions have to be considered over the course of the growing season to assess controls on coccolithophore biogeography, both experimentally and in modeling studies.

Coccolithophores impact biogeochemical cycles, and especially organic matter cycling, carbon sequestration and oceanic

carbon uptake both via photosynthesis and calcification, leading to cascading effects on the global carbon cycle and hence climate. Thus, it is crucial to assess more quantitatively the contribution of this crucial phytoplankton group to changes in these processes in the past, present and future ocean.

*Data availability.*  Model data are available upon email request to the first author (cara.nissen@usys.ethz.ch) and on the public repository located at ftp://data.up.ethz.ch/SO_d025/CN_CoccoBiogeography/.

**Appendix A:  Data for model evaluation**



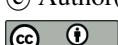

**Table A1.** Data sets used for model evaluation. Please see section S1 in the supplementary material for a more detailed description of the data used to evaluate simulated phytoplankton biogeography, community structure and phenology.

| Variable | Source |
|---|---|
| Mixed layer depth (MLD) | Monthly climatology from Argo float data (Holte et al., 2017) |
| Sea surface temperature (SST) | Optimum Interpolation SST, version 2: monthly climatology from 1981-2014 (Reynolds et al., 2007) |
| Nitrate, phosphate, silicate | Monthly climatology from World Ocean Atlas 2013 (Garcia et al., 2013b) |
| Surface total Chlorophyll | Monthly climatology from MODIS Aqua (NASA-OBPG, 2014a), SO algorithm (Johnson et al., 2013) |
| Net Primary Productivity (NPP) | Monthly climatology from from 2002-2016 from MODIS Aqua VGPM |
| | (Behrenfeld and Falkowski, 1997; O'Malley, 2016) |
| | Annually integrated NPP climatology from 2002-2016 from Buitenhuis et al. (2013a) |
| Particulate Organic Carbon (POC) export | Monthly output from a biogeochemical inverse model (Schlitzer, 2004) |
| | and a data-assimilated model (DeVries and Weber, 2017) |
| Particulate Inorganic Carbon (PIC) export | Monthly output from standard simulation in Jin et al. (2006) |
| Coccolithophore Biomass | MAREDAT (O'Brien et al., 2013; Buitenhuis et al., 2013b), additional data from |
| | Balch et al. (2016), Saavedra-Pellitero et al. (2014), |
| | Tyrrell and Charalampopoulou (2009), Gravalosa et al. (2008), Cubillos et al. (2007) |
| Diatom Biomass | MAREDAT (Leblanc et al., 2012; Buitenhuis et al., 2013b), additional data from Balch et al. (2016) |
| Coccolithophore Calcification | Monthly surface chlorophyll, SST, and particulate inorganic carbon (PIC) climatologies |
| | from MODIS Aqua (NASA-OBPG, 2014a, c, d), Eq. 1 from Balch et al. (2007) |
| HPLC | Monthly CHEMTAX climatology based on high performance liquid tomography (HPLC) data (Swan et al., 2016) |

## Appendix B: BEC equations: Phytoplankton growth & loss

Changes over time of phytoplankton biomass $P$ [mmol C m$^{-3}$] of phytoplankton $i$ ($i \in \{C, D, SP, N\}$) are controlled by growth and loss terms:

$$\frac{\mathrm{dP^i}}{\mathrm{dt}} = \mathrm{Growth} - \mathrm{Loss} \tag{B1}$$

$$= \mu^i \cdot P^i - \gamma^i(P^i) \cdot P^i \tag{B2}$$

$$= \mu^i \cdot P^i - \gamma_g^i(P^i) \cdot P^i - \gamma_m^i \cdot P^i - \gamma_a^i(P^i) \cdot P^i \tag{B3}$$

with $\gamma_g$ denoting loss by zooplankton grazing, $\gamma_m$ loss by non-grazing mortality and $\gamma_a$ loss by aggregation.

## 5 B1 Phytoplankton growth

The specific growth rate $\mu^i$ [day$^{-1}$] of phytoplankton $i$ is determined by the maximum growth rate $\mu_{\max}^i$ (see Table 1) which is modified by environmental conditions with respect to temperature (T), nutrients (N) and irradiance (I), following:

$$\mu^i = \mu_{\max}^i \cdot f^i(T) \cdot g^i(N) \cdot h^i(I) \tag{B4}$$

The temperature function $f(T)$ is an exponential function, being <1 for temperatures below T$_{\mathrm{ref}}$=30°C, modified by the constant Q$_{10}$ specific to every phytoplankton $i$ (see Table 1) describing the growth rate increase for every temperature increase





of 10°C:

$$f^i(T) = Q_{10}^i \cdot \exp\left(\frac{T - T_{ref}}{10°C}\right) \tag{B5}$$

The limitation by surrounding nutrients $L^i(N)$ is first calculated separately for each nutrient (nitrogen, phosphorus, iron for all phytoplankton, silicate for diatoms only) following a Michaelis-Menten function (see Table 1 for half-saturation constants $k_N^i$ for the respective nutrient and phytoplankton $i$). For iron (Fe) and silicate (SiO3), the limitation factor is calculated following:

$$L^i(N) = \frac{N}{N + k_N^i} \tag{B6}$$

For nitrogen and phosphorus, the limitation factor is calculated as the combined limitation by nutrient $N$ and $M$ (nitrate (NO3) and ammonium (NH4) for nitrogen, phosphate (PO4) and dissolved organic phosphorus (DOP) for phosphorus) following:

$$L^i(N, M) = \frac{N}{k_N^i + N + M \cdot (k_N^i/k_M^i)} + \frac{M}{k_M^i + M + N \cdot (k_M^i/k_N^i)} \tag{B7}$$

Then, only the most limiting nutrient is used to limit the phytoplankton growth rate:

$$g^i(N) = \min(L^i(NO3, NH4), L^i(PO4, DOP), L^i(Fe), L^i(SiO3)) \tag{B8}$$

The light limitation function $h^i(I)$ accounts for photoacclimation effects by including the chlorophyll-to-carbon ratio $\theta^i$, as well as the nutrient and temperature limitation of the respective phytoplankton $i$:

$$h^i(I) = 1 - \exp\left(-1 \cdot \frac{\alpha_{PI}^i \cdot \theta^i \cdot I}{\mu_{max}^i \cdot g^i(N) \cdot f^i(T)}\right) \tag{B9}$$

In ROMS-BEC as used in this study, coccolithophore growth is linearly reduced at temperatures <6°C following:

$$\mu^C = \mu^C \cdot \frac{\max(T + 2°C), 0)}{8°C} \tag{B10}$$

Additionally, coccolithophore growth is set to zero at PAR levels <1 W m$^{-2}$ (Zondervan, 2007).

Calcification by coccolithophores is proportional to photosynthetic growth of coccolithophores with a constant CaCO$_3$:C$_{org}$ production ratio of 0.2.

Diazotroph growth is zero at temperatures <14°C.

## B2 Phytoplankton loss

In ROMS-BEC, loss rates of phytoplankton biomass are computed using a corrected phytoplankton biomass P$'^i$, to limit phytoplankton loss rates at low biomass:

$$P'^i = \max(P^i - c_{loss}^i, 0) \tag{B11}$$




In this equation, $c_{loss}^i$ is the threshold of phytoplankton biomass $P^i$ below which no losses occur ($c_{loss}^N$=0.022 mmol C m$^{-3}$ and $c_{loss}^{C,D,SP}$=0.04 mmol C m$^{-3}$).

The grazing rate $\gamma_g^i$ [mmol C m$^{-3}$ day$^{-1}$] of the generic zooplankton $Z$ [mmol C m$^{-3}$] on the respective phytoplankton $i$ [mmol C m$^{-3}$] is described by

$$\gamma_g^i = \gamma_{max}^i \cdot f^Z(T) \cdot Z \cdot \frac{P'^i}{z_{grz}^i + P'^i} \tag{B12}$$

with

$$f^Z(T) = 1.5 \cdot \exp(\frac{T - T_{ref}}{10°C}) \tag{B13}$$

The non-grazing mortality rate $\gamma_m^i$ [mmol C m$^{-3}$ day$^{-1}$] of phytoplankton $i$ [mmol C m$^{-3}$] is the product of a maximum mortality rate $m_0^i$ [day$^{-1}$] scaled by the temperature function $f^i(T)$ with the modified phytoplankton biomass $P'^i$:

$$\gamma_m^i = m_0^i \cdot f^i(T) \cdot P'^i \tag{B14}$$

with $m_0^i$ being 0.15 day$^{-1}$ for diazotrophs and 0.12 day$^{-1}$ for all other phytoplankton.

Aggregation losses are assumed only to occur for diatoms, small phytoplankton and coccolithophores. The aggregation rate $\gamma_a^i$ [mmol C m$^{-3}$ day$^{-1}$] of phytoplankton $i$ [mmol C m$^{-3}$] is described by:

$$\gamma_a^i = \min(r_{a,max}^i \cdot P'^i, 0.001 \cdot P'^i \cdot P'^i) \tag{B15}$$
$$\gamma_a^i = \max(r_{a,min}^i \cdot P'^i, \gamma_a^i) \tag{B16}$$

with $r_{a,min}^i$ being 0.01 day$^{-1}$ for small phytoplankton and coccolithophores and 0.02 day$^{-1}$ for diatoms, and with $r_{a,max}^i$ being 0.9 day$^{-1}$ for all three phytoplankton.

*Author contributions.* NG, MV, and CN conceived the study. CN, AH and MM set up the model simulations. CN performed the analysis. CN and MV wrote the paper. All authors contributed to the interpretation of the results and to the paper. NG, MV, and MM supervised this study.

*Competing interests.* The authors declare that they have no conflict of interest.

*Acknowledgements.* We would like thank all the scientists who contributed phytoplankton and zooplankton cell count data to the MARE-DAT initiative, as well as William Balch, Helen Smith, Mariem Saavedra-Pellitero, Gustaaf Hallegraeff, José-Abel Flores, and Alex Poulton for making additional coccolithophore or diatom cell count data available for this study. Furthermore, we would like to thank Martin Frischknecht, Elisa Lovecchio, and Domitille Louchard for valuable feedback, and Damian Loher for technical support. This research was



financially supported by the Swiss Federal Institute of Technology Zürich (ETH Zürich) and the Swiss National Science Foundation (project SOGate, grant no. 200021_153452). The simulations were performed at the HPC cluster of ETH Zürich, Euler, which is located in the Swiss Supercomputing Center (CSCS) in Lugano and operated by ETH ITS Scientific IT Services in Zürich. Model output is available upon request to the corresponding author, Cara Nissen (cara.nissen@usys.ethz.ch).





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
