# Peer review of "Factors controlling coccolithophore biogeography in the Southern Ocean"

_Biogeosciences, 2018_

## Referee Comment (RC1) · Anonymous Referee #1 · 5 Jun 2018

GENERAL COMMENTS

The modelling study of Nissen et al. provides an important and interesting examination of the biotic and abiotic population and biogeographical drivers of coccolithophores, in the context of other phytoplankton groups, in the Southern Ocean. The paper is well written and clear to follow in all aspects, with the results neatly summarised in the main figures and text. There is also an appropriate level of appreciation of the limits of the model output and field data (though a few omissions which should be addressed, see comments below). I have only minor comments to make:

1. Non-grazing mortality – It is not explicitly discussed in the paper as to what the authors consider this to be. Viral lysis is seen as a major mortality pathway for coccolithophore (bloom) communities and so is this what the authors mean by this terminol-

ogy? How is it parameterised and does it fairly represent viral mortality or (e.g.) programmed cell death? Not representing (or discussing) such a major mortality pathway seems like a limitation of the study, but a necessary limitation due to the uncertainties around viral mortality dynamics and its role in the Southern Ocean. The authors should include viral mortality in their discussion over model limitations, as well as directions for future field observations.

2. Importance of bottom-up and top-down controls – The conclusion that both types of controls need to be considered when examining phytoplankton (and coccolithophore) population dynamics and biogeography is very important point to be made. However, the statement is not limited to the Southern Ocean and is relevant across the full biogeographical range of coccolithophores.

3. Coccolithophores/Emiliania huxleyi – Do the authors consider they have parameterised their model to describe the whole coccolithophore community, or rather that they are limited to E. huxleyi dynamics in the Southern Ocean? For this region it is relatively simple as E. huxleyi dominates (to almost monospecific levels depending on latitude). Within the authors recognised limitations, discussion of this point should be considered, especially if there are aspirations to expand such modelling efforts to low-latitude highly-diverse coccolithophore communities. Related to this point, the 400% overestimation of coccolithophore biomass (Pg 19, Lns 25-26) applies to the whole coccolithophore diversity, and in diverse communities would indeed lead to significant issues, however in the E. huxleyi dominated Southern Ocean such issues are far less extreme. There are also numerous estimates of E. huxleyi cell biomass (and even B/C biomass) which are in agreement (and don't vary by 400%).

SPECIFIC COMMENTS

Pg 1, Ln 16: Please specify 'Ocean Acidification' rather than just 'acidification'.

Pg 1, Ln 22: It is not just the ratio of calcifying to silicifying phytoplankton that is crucial to consider, it is the ratio of calcifying to non-calcifying (organic only) phytoplankton.

Pg 2, Lns 4-5: It should be recognised that all these references are model based estimates rather than field estimates, and also take varying ways to parameterise coccolithophore production. See also pg 19, ln 21 – here it should also be recognised that these low estimates of coccolithophore NPP are derived from model studies with diverse parameterisations of coccolithophore calcification.

Pg 2, Lns 10-11: Cell densities of 2.4 x 103 cells mL-1 have to be for the Patagonian Shelf bloom and are really (really) high whilst cell densities elsewhere in the Atlantic sector of the SO are much (much) lower. The authors should make it clear that these high numbers are from bloom waters.

Pg 3, Ln 14: Please make clear that zooplankton grazing includes both micro- and macro-zooplankton (rather than just the latter).

Pg 4, Lns 6-7: 'Coccolithophores grow well at high light intensities and at a range of different temperatures, but have been shown to be light-inhibited at low light levels' – does this statement fit coccolithophores as a group or just E. huxleyi?

Pg 5, Ln 19: What is the justification (reference) for using such extremely low carbon to chlorophyll ratios (3 to 5)? These lead to extremely chlorophyll-rich phytoplankton cells whereas ratios are typically 10 to 20 times higher. Are these based on Southern Ocean studies?

Figure 2: Colours seem to have changed on panel (a) – blue looks olive green and grey looks to be light green?

Pg 14, Ln 4: extra 'a' in this sentence.

Pg 20, Ln 30: A key statement – 'coccolithophores appear to be of minor importance for global oceanic organic carbon fixation'. Many in situ studies agree with such small contributions to phytoplankton biomass or primary production in the Southern Ocean (including those already cited in the paper: Smith et al., 2017; Charalampopoulou et al., 2016; Poulton et al., 2013; Hinz et al., 2012).

Pg 24, Lns 22: 'Based on our findings, future SO in-situ studies should consider both bottom-up and top-down factors when assessing coccolithophore biogeography in space and time'. This statement should not be limited to just the Southern Ocean.

Pg 25, Lns 19 and 22-23: As well as multiple trophic levels (and trophic cascades), what about non-grazing mortality (i.e. viral mortality?). This is not discussed anywhere in the paper and the omission of viral driven population dynamics needs to be addressed in the limitations.

---

## Referee Comment (RC2) · S. M. Vallina (Referee) · 8 Jul 2018

ïż£Title: Factors controlling coccolithophore biogeography in the Southern Ocean

C. Nissen et al.

Submitted to BGD

I. General comments:

This work is a modelling exercise seeking to understand the potential mechanisms behind the simulated patterns of spatial and temporal variability for some major phytoplankton taxa over the Southern Ocean region (south of 30S). Particularly the authors main focus is on coccolithophores (coccos) and diatoms, with an emphasis on understanding coccos dynamics. The goal is to disentangle the several environmental factors (nutrients, solar radiation, temperature) affecting the growth of these phytoplankton groups as "bottom-up" effects, and compare them with the grazing "top-down" effects affecting their mortality. The work is quite original and goes beyond the simple description of the model results and seeks to dig into the mechanisms, so I really like the approach. The article is well written, the objectives are clear, and the results are well explained.

However I did find several technical and conceptual issues that I feel need to be addressed. Especifically the authors call i) "relative growth rate" what in reality is a "relative growth limitation" term, which is not the same thing; and ii) "relative grazing rate" what in reality is a "relative clearance rate", which is not the same thing either. Further, they make inferences on the relative effect of bottom-up versus top-down effects based on comparing these two concepts (i and ii) while they are not comparable – basically because they have different units (before making the log10 of their ratio for coccos : diatoms). What they need to compare is log10(x/y) where x has units of days-1 for coccos and y has units of days-1; BOTH for growth rate and grazing rate. At the moment for relative growth they are using x and y using non dimensional units (n.d), and for the relative grazing they are using x and y using clearance rate units (m3 * m-3 * d-1). To be consistent the growth limitation terms must be multiplied by maximum growth rate (mupmax; d-1) and the grazing clearance rate must be multiplied by zooplankton biomass (Z; mmol * m-3). This will make those processes (bottom-up vs. top-down) comparable because they will have the same units (d-1) before making the ratio and taking the log10. The two major things the author will notice are: 1) the specific growth rate (d-1) of diatoms is larger than coccos for most of the environmental conditions; only at very narrow window of small nutrient concentration will coccos outcompete diatmos; 2) the specific grazing rate (d-1) on diatoms is ALWAYS smaller than on coccos (for a constant zooplankton biomass).

I have performed myself an extensive analysis using MATLAB/OCTAVE of the niche

properties of these simulated phytoplankton taxa based on the model parameters provided. That is how I discovered these and other conceptual errors, which may affect the discussion of the manuscript. For example, the grazing equation eq(5) described in the text: G1 = gmax * Z * P1 / (K + P1) is incorrect to compute the grazing on P1. The correct equation should be G1 = gmax * Z * P1 / (K + P1 + P2) if we assume that there are two prey (P1 + P2) available for grazing (e.g. diatoms and coccos). This correction (adding P1 + P2 in the denominator of the grazing functional response) alters the results of the "relative clearance rates" (C:D) and the coccos are always grazed faster than diatoms. Therefore, given the parameter values provided by this model there is no surprise that diatoms will dominate coccos by a factor of x10 almost everywhere and at anytime. The idealized analyses of niche properties that I performed I think should be added as Supplementary Material in a revised form of this manuscript. A table with the model parameters is important for replication of this work but a figures showing the actual shapes of the nutrient uptake curves and grazing functional response are important for the reader because it provides a way of fast visual inspection and understanding. For example, just by visual inspection of the grazing functional responses one can predict a-priori that coccos are ALWAYS going to be grazed faster than diatoms.

Also the selection of uptake curves for nutrient uptake is quite puzzling to me. For DIN, NH4 and DIP, they are kind of similar where coccos are always losers to either diatom or small phyto; then for DOP the coccos appear to have small nutrient window where they can dominate. Then for iron coccos and small phyto are very similar in nutrient uptake strategy while diatoms dominates. Finally dizotrophs are clear losers for all nutrients against all other phytoplankton groups – no surprise they are almost zero biomass (less than 1%) in the model. I would suggest the authors to simplify the model parameters: define a single set of four values (one per phyto group) for the half saturation constant on Dissolved Organic Phosphorous (DOP) and then compute the half-sat for all other nutrients (DIN, NH4, DIP, Fe2, Silica) using constant redfield ratios. I did this myself and I feel that uptake curves obtained are nicer and more consistent across nutrient

types, and they also provide a slightly larger nutrient concentration niche window for coccos where they can dominate. There is "key switch" in my MATLAB/OCTAVE code (key_params = 'Original' or 'Redfield') to change from the original model parameters setup to the suggest parameter setup. The results of the simulation should not change qualitatively after this minor changes on parameter values, and if they do then there is an issue of model sensitivity to parameter values. Please see the resulting Figures in the attached PDF documents. I provide the MATLAB/OCTAVE code I wrote at the end of this review report. The authors are free to use it.

Regarding the Sensitivity Analysis (SA) performed by the authors, I am afraid to say that is not useful in the way it has been designed. Basically because 1) changing the selected parameters "from the values of coccos to the values of diamtoms" provides too little change of the parameter values; 2) the percentage of change for the selected parameters is different among them. That means that 1) the response of the model for the "sensitivity runs" will usually be quite small for most of the parameters; and 2) the responses of the model for the "sensitivity runs" will not be comparable accross parameters. For example, changing the Q10_coccos (1.45 n.d.) to Q10_diatoms (1.55 n.d.) is a meager 7% variation in the Q10 value of coccos; or changing the alpha_coccos (0.4) to alpha_diatoms (0.44) means a 10% variation in the alpha PI value of coccos. Then, changing mupmax_coccos (3.8 d-1) to mupmax_diatoms (4.6 d-1) implies a 20% variation in the mupmax of coccos; and changing the half sat kno3_coccos (0.3) to kno3_diatoms (0.5) implies a 66% variation. On the other hand, changing the half sat for grazing zgrz_coccos (1.05) to zgrz_diatoms (1.0) implies a 5% variation. Therefore, some parameters are changed less than 10% while others are changed more than 60%, and others are changed around 20%. This is too messy for a meaningful SA. The correct way to perform a proper (yet simple) SA is by changing a plus/minus 50% the selected parameter values, and then computing the Sensitivity Index (SI) as follows – and the plotted as shown in Figure 12 of Vallina et al. (2008) DMOS model, JGR:

[Figure]

SI = 100 * (X_pmax - X_pmin) / X_pcontrol ;

where:

**X – is the model state variable of interest (e.g. Diatom concentration) # pcontrol – is the selected parameter value for the control run (e.g. mupmax of coccos) # pmax = 1.5 x pcontrol – is the run for the plus(+) 50% increase in the parameter value # pmax = 1.5 x pcontrol – is the run for the minus(-) 50% decrease in the parameter value # SI – is the Sensitivity Index in percentage of change (can be positive or negative %)**

II. DETAILED COMMENTS:

- Page 1, Line 01: Change "controls not only the local biogeo but also" to "controls the local biogeo and" – Avoid unnecessary negatives in affirmative sentences. - Page 3, Line 03: "interaction" is very generic term – Would be better to specify "competitive interaction" (for light and nutrients) or just "competition". - Page 4, Table 1: The parameter values for mupmax at 30C are quite large (3.6, 4.6 d-1) and in my view quite beyond the reality of these values in nature. - Page 4, Table 1: The parameter values for Q10 are almost the same for all phyto – why dont you just use 1.5 for all of them? - Page 4, Table 1: The parameter values for knh4 are a factor of x 10 smaller (i.e. 0.10) than kno3 for all C, D and SP; but 0.15 for diazotrophs – why dont you just use time 0.10 for all of them? - Page 4, Table 1: The parameter values for kpo4 are the same value than for kno3 for C, D, and SP (even if the Redfiled ratio N:P is 16:1 – why?) but for diazos is ten times smaller – why? - Page 4, Table 1: The parameter values for alpha PI curve are between 0.38 and 0.44 for all phyto groups (very narrow variability) – why dont you just use 0.4 for all of them? - Page 4, Table 1: The parameter values for zgrz half sat grazing are between 1.0 and 1.2 for all phyto groups (very narrow variability) – why dont you just use 1.0 for all of them? - Page 4, Table 1: - Page 4, Table 1: Basically I find that the selection of parameter values can be simplified a lot given the similarity among many of the values. I would suggest selecting one nutrient (e.g. DOP) and define the half-sat values for it, then compute the half-sat for all other

nutrients (DIN, NH4, DIP) using Redfield ratios and constant factors. This will make the list of parameter values much smaller without compromising the simulations. Also this will simplify the Sensitivity Analysis because it reduces the degrees of freemdom in the parameter space. - Page 4, Line 5: "coccos have a higher nutrient affinity (smaller half-sat) and a smaller max growth rate than diatoms" – the half sat parameter is in fact a composite of two primary parameters (ksat = mupmax / affinity) therefor lower half sat does not necessarily mean higher affinity, it also depends on the mupmax value. Also, the smaller size of coccos cannot be the reason of their smaller mupmax because mupmax usually decreases with increasing cell size. - Page 6, Table 2: Run name 10 and 11; Holling Type III and Active Switching – I dont understand... Very confusing. Please double check and make it clearer. - Page 6: Table 2: Run name 12; TEMP; Reduce temperature by 1C – This seems a pretty low decrease in temperature for a SA. What percentage is this? - Page 6, Line 04: "We then add an active prey switching term to the original Holling Type II" – The Holling Type III has already "active prey switching" dynamics so I dont really know what this addition to Holling Type II means. - Page 7, Line 15: "Ideal env conditions ... at every location" – The nutrient concentration window where coccos reach a higher specific growth rate than diatoms is very narrow and happens at very low nutrient concentrations; for all other conditions the diatom are superior competitors regardless of light and temperature levels. Therefore what dominates the competition between coccos and diatoms is basically nutrients only (in your model setup). - Page 7, Line 19: The equation for grazing rate on Pj is wrong: it should be – $gamma\_j = gammax\_j * f(T) * Z * (P\_j / (zgrz\_j + sum[P\_i]))$ NOTE: The sum[P\_i] - Page 7, Line 23: Why do you use the notation "P\_j prime" (P') instead of just P (without the prime)? - Page 7, Line 20: "loss threshold below which no losses occur (eq B11)" – This is called imposed "prey refuge" and helps prevent competitive exclusion of the less dominant prey types (in a similar way as active prey switching). - Page 8, Line 03: The ratio ($gamma\_j$ / $P\_j$) will only have units of clearance rate (m3 * mmol-1 * d-1) if divided by zooplankton biomass. Otherwise it will have units of specific grazing rate (d-1) – please double check units and concepts. - Page 8, Line 03: Nevertheless, if (gamma_j / P_j) is called a "clearance rate", you should call "log10((gamma_C / P_D)/(gamma_C / P_D))" as "Relative clearance rate" (instead of "Relative grazing rate"). - Page 8, Line 04: "the specific grazing (clearance) rate on diatoms is larger" – as far as I can tell, using the correct gamma_j functional response (with the zgrz + sum[P_i]; see above) the specific grazing (clearance) rate on diatoms will ALWAYS be smaller than on coccos. Plase double check. See my MATLAB/OCTAVE analyses and figures. - Page 8, Line 25: "caused either by too much uptake by phytoplankton" – Given the too high mupmax selected for this model I am not surprised about this; why dont you tune down mupmax accordingly to prevent this bias? - Page 9, Table 3: "(global)" – what that this means? confusing. - Page 9, Table 3: The contribution of coccos is: 15%, 10% and 1% for the three regions – isnt it this too low? - Page 9, Table 3: I dont really see how coccos can even survive based on the uptake curves (nutrient niches) defined in the model set up. The coccos uptake curves seem to be always below those of either diatoms (at high nutrients) and small phyto (at small nutrients). The temperature dependence is almost the same for all phyto groups so it does not play a role. The light dependence is also quite similar for the three dominant groups (C,D,SP). Thus it is mostly down to nutrients competition and if the coccos uptake curve is never above those of their competitors, with a Type II grazing functional response they should be competitively excluded. Only the imposed prey refuge will keep them persisting in the model. - Page 9, Line 17: "within the globally estimated" – what do you mean? how can you compare a regional estimate to a whole ocean estimate? - Page 9, Figure 1: The model simulates chla fairly well spatially despite some biases – How well does the model reproduce the seasonal dynamics of chla? Please provide a figure in Supplementary Material. - Page 9, Figure 1, f) Personally I find that the model does a poor job with coccos in terms of spatial distribution, only the absolute values seem to be the correct but clearly not the patterns. The deviation from the data values especially between 60S and 75S is too large, up to an order of magnitude even. Can you say that the model simulations for coccos have been validated really? - Page 10, Line 7: The model NPP estimates are between [0.23 - 0.69] (PgC * y-1) and the real data NPP

estimates are between [0.64 - 0.94] (PgC * y-1). This implies that the model's NPP is basically half of the real data NPP estimates – why do you say then that if "falls with the range estimated by satellite"? Just say that the models underestimates NPP by 50%. - Page 12, Figure 3: Dizatrophs are basically extinct in this model simulations. This is because the selected uptake curves for them (see my MATLAB/OCTAVE figures) that makes them very poor nutrient competitors. Why do you even bother in having them as a phytoplankton group? I dont understand this. - Page 12, Lines 15-20: Basically this means that you dont know why? Is not it any way to explore the reasons beyond verbal speculation? - Page 13, Line 1: "this PFT (Phaeocystis) is not included in our simulations" – Honestly I dont think this is a valid excuse. Given that diazotrophs are irrelevant in this model simulations, why dont you use that tracer to model Phaeocystis instead? - Page 13, Line 4: "simulated gradient ... coccos contribution" – This is a misleading statement: diatoms clearly dominate everywhere in the model by about a factor of x10, so talking about mixed community, south-north increase in coccos contribution, etc. is verbally misleading. - Page 13, Line 13: "whereas diatom biomass peaks south of 60S where they dominate the community" – again, this is misleading; diatoms dominate the community *everywhere* in your domain, not just south of 60S. - Page 15, Line 10: "the specific growth rate of coccos is on average 10% larger than of diatoms" – IMPORTANT: This statement is wrong due to a conceptual misunderstanding. Doing log(x/y) where x = DIN / (kdin + DIN) for diatoms and y is the equivalent thing for coccos does *not* measure the "relative growth rate" of diatoms versus coccos but the "relative growth limitation", which is not the same thing. If you want to evaluate "relative growth rate" you have to do: Rel growth = log(umax_D/umax_C) + log(x/y) – Please change this in the analysis and the text accordingly. - Page 15, Line 18: "The 21% larger umax of diatoms ... all year round in the whole model domain" – This is a main result of the model simulations. Thus it must be at the beginning of section 4.5 not buried here. - Page 15, Line 21: "coccos being less temperature limited" – Misleading, the differences of Q10 are marginal (7% is basically nothing) - Page 15, Line 24: "are less nutrient limited" – This is the right wording. One thing is being less nutrient limited

and another thing is having a faster nutrient uptake curve, the difference lies on the umax. - Page 15, Line 35: "differences in the sensitivity to increases of PAR at low irradiance (alfa PI) and diffs in photoaclim" – Misleading, the differences of alfa_PI are marginal (9% is basically nothing) - Page 16, Figure 5: The different terms that are being plotted here as "relative growth ratio" for nutrients, temperature, light, and total are not really "growth rate" ratios" but "growth limitation" ratios. This leads to wrong interpretation of the results. The right way should be to plot the following curves:

Qdin = DIN / (Kdin + DIN); Nutrients limitation [0 - 1] n.d. Qpar = f(PAR); Irradiance limitation [0 - 1] n.d. Qsst = f(SST); Temperature limitation [0 - 1] n.d.

Rel_umax = log(umax_D/umax_C); Rel_Qdin = log(Qdin_D/Qdin_C); – Relative nutrient growth "limitation" Rel_Qpar = log(Qpar_D/Qpar_C); – Relative irradiance growth "limitation" Rel_Qsst = log(Qsst_D/Qsst_C); – Relative temperature growth "limitation"

Rel_din = Rel_umax + Rel_Qdin; – Relative nutrient growth "rate" Rel_par = Rel_umax + Rel_Qpar; – Relative irradiance growth "rate" Rel_sst = Rel_umax + Rel_Qsst; – Relative temperature growth "rate"

Rel_Qlim = Rel_Qdin + Rel_Qpar + Rel_Qsst; – Relative total growth "limitation" Rel_growth = Rel_mupmax + Rel_Qlim; – Relative total growth "rate"

- Page 17, Figure 6b) – "Relative grazing ratio higher on diatoms" – This is very weird, I think this computation of grazing ratio is wrong. When I did on my MATLAB/OCTAVE analyses I find that the "grazing ratio" (in fact it should be called "clearance rate" ratio) is ALWAYS higher on coccos. And if we look at the grazing functional response we can se that it is higher than those for coccos at any prey concentration. I think the reason is the lack of the (zgrz + sum[Pi]) at the denominator of the grazing computation. If you are using (zgrz + P1), instead of (zgrz + P1 + P2), that will be wrong. Please double-check. - Page 17, Line 1: "coccos have a lower alfa PI" – Only 9% lower, please explicitly say so. - Page 17, Line 2: "a generally lower chla-to-carbon" ratio (not shown) – How much lower in percentage? Why it is not shown? - Page 17, Line 4: "coccos

are on average 2% - 3% more light limited than diatom" – This is basically nothing from a competitive exclusion point of view at ecological (seasonal) time scales. - Page 17, Line 11: "coccos and diatom together contribute on average 87% and 95%" – Misleading statement because diatoms clearly dominate (90% diatoms vs 10% coccos); dont plug them together. - Page 17, Line 13: "They are thus not only competing for resources between each other but with SP as well" – This statement is obvious; of course all phytoplankton PFT compete among them for nutrients. - Page 18, Line 5: "advantage in specific growth" – LIMITATION not RATE. This must be clear. - Page 18, Line 6: "greater importance" – Misleading, coccos are always poor competitors in this model simulations. - Page 18, Line 10: "higher specific growth rate" – WRONG: this should say "higher specific growth limitation"; it is *not* the same thing. - Page 18, Line 11: "We calculated whether the length of the growing season is long enough" – Good try, but wrong answer. Coccos may grow faster (in days-1) than diatoms over a very narrow band of nutrient concentration which I dont think is even hapening on your models simulations. I suspect that diatoms grow faster (in days-1) than coccos everywhere in your domain and everytime in the year. Plase double-check. - Page 18, Line 25: If grazing pressure were able to explain the mismatch between the expected results for coccos vs diatoms (from the model) and observed ones (from the model), this could be easily confirmed by performing a run where maximum grazing rate (gamma max) and half saturation constant for grazing (zgrz) are the same for both coccos and diatoms. Please do it and report the results in Supp. Material. - Page 18, Line 28: "specific grazing rate on coccos" – it is not clear to me if this is measuring relative "specific grazing rates" (d-1 vs d-1) or relative "specific clearance rates" (m3 * mol-1 * d-1) vs. (m3 * mol-1 * d-1). Ideally it should measure (d-1 vs d-1) to be correct and consistent. - Page 18, Line 34: "differences in specific grazing rates between diatoms and coccos are of similar magnitude as differences in specific growth rates" – WRONG: You cannot compare differences in specific clearance rates (I am not sure about the units yet; maybe [d-1] or maybe [m3 * mol-1 * d-1]) with specific LIMITATION rates (in non dimensional [n.d.] units), even when having them log transformed so that
their both lose their original units. Make sure that "relative grazing rate" and "relative growth rate" is based on process with units of days-1 in both cases. Otherwise they are not comparable. - Page 19, Line 7: "During this times coccos experience a larger per biomass grazing pressure" – in fact coccos experience a larger per biomass grazing pressure at ALL TIMES. Please double check and correct the text accordingly (e.g. line 10) - Page 19, Line 17: "Parameter Sensitivity Simulations" – Change to "Parameter Sensitivity Analysis" and perform the SA suggested in my General Comments. The current SA is not meaningful. - Page 20, Line 18: "coccos are a non-negligible" – Change to "coccos are a minor but non-negligible" - Page 21, Line 20: "The net sign of ... future research" – Why future research? I think this may actually be the most important point to be addressed by this work. Why cannot be done now? - Page 21, Line 26: "succession" – Margalef's Mandala concept of succession implies a temporal dominance. However coccos do never get anywhere close to dominate the biomass since diatoms are always above 80%. Therefore the term "succession" does not apply here. - Page 21, Line 24: "specific growth rate" – Change to "specific growth limitation" or compute the correct "specific growth rate" after multiplying by umax. Currently what is higher is the nutrient limitation "Qdim = DIN / (kdin + DIN)" term (n.d.) but not the growth rate "umax * Qdim" term (d-1) - Page 23, Figure 8: This figure is pretty but to complex – Too many info (colors, shades, arrows, shapes, letters, low high); I honestly dont understand anything. - Page 25, Line 1: "pressure on less abundant" – Change to "pressure on relatively less abundant". - Page 26, Line 9: "coccos biomass is high when diatoms" – Change to "coccos biomass is higher"; their biomass is never high. - Page 26, Line 11: "never exceeds that of" – Change to "never gets even close to" that of diatoms.

— # MATLAB / OCTAVE:

more off close all clear all format short g

%SWITCHING KEYS: key_params = 'Original'; %%key_params = 'Redfield';

cmap_redblue(01,:)    = [0.00000 0.15959 0.35490]; cmap_redblue(02,:)    = [0.13721 0.37159 0.56690]; cmap_redblue(03,:)  = [0.34921 0.58359 0.77890]; cmap_redblue(04,:)    = [0.56121 0.79559 0.99090]; cmap_redblue(05,:)    = [0.77321 1.00000 1.00000]; cmap_redblue(06,:)  = [0.98521 1.00000 1.00000]; cmap_redblue(07,:)    = [1.00000 1.00000 0.97950]; cmap_redblue(08,:)    = [1.00000 0.95243 0.68290]; cmap_redblue(09,:)  = [1.00000 0.65583 0.38630]; cmap_redblue(10,:)  = [1.00000 0.35923 0.08970]; cmap_redblue(11,:)  = [0.79310 0.06263 0.00000]; cmap_redblue(12,:)  = [0.49650 0.00000 0.00000]; cmap = [cmap_redblue(01:05,:);cmap_redblue(08:12,:)];

%REDFIELD RATIOS: R_FeP = (0.10/1.00); %[molFe* molP-1] R_NP = (16.00/1.00); %[molN * molP-1] R_CN = (106.00/16.00); %[molC * molN-1]

%MODEL PARAMS: Q10(1) = 1.45; Q10(2) = 1.55; Q10(3) = 1.50; Q10(4) = 1.50;

alfa(1) = 0.40; alfa(2) = 0.44; alfa(3) = 0.44; alfa(4) = 0.38;

teta(1) = 0.01; %[mgChla * mgC-1] teta(2) = 0.01; teta(3) = 0.01; teta(4) = 0.01;

kpar = (1./alfa) .* (1./teta)*(1/4); %[W * m-2] gammax(1) = 4.40; %Coccos gammax(2) = 3.80; %Diatom gammax(3) = 4.40; %SamllP gammax(4) = 2.00; %Diazos

zgrz(1) = 1.05; %Coccos zgrz(2) = 1.00; %Diatom zgrz(3) = 1.05; %SamllP zgrz(4) = 1.20; %Diazos

mupmax(1) = 3.8; %Coccos mupmax(2) = 4.6; %Diatom mupmax(3) = 3.6; %SamllP mupmax(4) = 0.9; %Diazos

kdin = ones(1,4)*nan; %Nitrate (NO3) kdip = ones(1,4)*nan; %Phosphate (PO4) kdop = ones(1,4)*nan; %Diss. Org. Phos. (DOP) knh4 = ones(1,4)*nan; %Ammonium (NH4) kfe2 = ones(1,4)*nan; %Iron (Fe2) ksio = ones(1,4)*inf; %Silica (SiO3)

kdop(1) = 0.30; %Coccos kdop(2) = 0.90; %Diatom kdop(3) = 0.26; %SamllP kdop(4) = 0.09; %Diazos

```
if strcmp(key_params,'Original')

%ORIGINAL VALUES: kdin(1) = 0.3; kdin(2) = 0.5; kdin(3) = 0.1; kdin(4) = 1.0; kfe2(1)
= 0.10; kfe2(2) = 0.12; kfe2(3) = 0.08; kfe2(4) = 0.08; ksio(2) = 1.00;

kdip = kdin/10; knh4 = kdin/10; kdip(4) = kdin(4)/50; knh4(4) = kdin(4)*(3/20);

elseif strcmp(key_params,'Redfield')

%REDFIELD VALUES:

mupmax(3) = mupmax(3)*0.90; %SmallP (my modification) kdop(3) = kdop(3)*(1/4);
%SmallP (my modification)

kfe2 = kdop*(R_FeP)*2.00; kdin = kdop*(R_NP )*1/20; kdip = kdin*(1/R_NP); knh4 =
kdin/10; ksio(2) = 1.0; %Diatom

end %endif

figure(5) subplot(2,2,1) plot(kdin,knh4,'b-',kdin,knh4,'r*') set(gca,'Xlim',[0 1.20 + eps])
set(gca,'Ylim',[0 0.20 + eps]) xlabel('ksat (DIN)') ylabel('ksat (NH4)') title('ksat (NH4
vs. DIN)') grid on subplot(2,2,2) plot(kdin,kdip,'b-',kdin,kdip,'r*') set(gca,'Xlim',[0 1.20 +
eps]) set(gca,'Ylim',[0 0.12 + eps]) xlabel('ksat (DIN)') ylabel('ksat (DIP)') title('ksat (DIP
vs. DIN)') grid on subplot(2,2,3) plot(kdin,kdop,'b-',kdin,kdop,'r*') set(gca,'Xlim',[0 1.20
+ eps]) set(gca,'Ylim',[0 1.40 + eps]) xlabel('ksat (DIN)') ylabel('ksat (DOP)') title('ksat
(DOP vs. DIN)') grid on subplot(2,2,4) plot(kdin,kfe2,'b-',kdin,kfe2,'r*') set(gca,'Xlim',[0
1.20 + eps]) set(gca,'Ylim',[0 0.30 + eps]) xlabel('ksat (DIN)') ylabel('ksat (Fe2)') ti-
tle('ksat (Fe2 vs. DIN)') grid on print('-dpng ','-r100','nissen_bg_KSAT.png') print('-
depsc','-r100','nissen_bg_KSAT.eps')

%ENVIRONMENTAL LIMITATIONS (DIN, PAR, SST): dinmin = 0.05; dinmax = 6.4;
parmin = 4.0; %[W * m-2] parmax = 256; %[W * m-2] sstmin = 0; sstmax = 30;
sstref = 30; sst = [sstmin:0.50:sstmax]; par = [parmin:4.00:parmax]; %[W * m-2] din
= [dinmin:0.05:dinmax]; dip = din*(1.00/16.00); nh4 = din/10; dop = dip*20; fe2 =
```

dop*R_FeP*2.0; sio = din; par(1) = par(2)/1d2; %to avoid +inf when divided by zero. sst(1) = sst(2)/1d2; %to avoid +inf when divided by zero.

%GRAZING LIMITATION (PHY): phy_chla = [0.02:0.02:2.56]*2; %(mgChla * m-3) cf1 = 1/teta(1); %(mgC * mgChla-1) cf2 = 1/(12.00*R_CN); %(mmolN * mgC-1) phy = phy_chla * cf1 * cf2; %(mmolN * m-3) pc = phy; %(mmolN * m-3) %%pd = phy; %(mmolN * m-3) pd = phy([1:2:end])*2; %(mmolN * m-3)

for i = 1:4 Qsst(:,i) = Q10(i) * exp( (sst - sstref) / 10.0); Qphy(:,i) = phy ./ (zgrz(i) + phy); Qdin(:,i) = din ./ (kdin(i) + din); Qdip(:,i) = dip ./ (kdip(i) + dip); Qnh4(:,i) = nh4 ./ (knh4(i) + nh4); Qdop(:,i) = dop ./ (kdop(i) + dop); Qfe2(:,i) = fe2 ./ (kfe2(i) + fe2); Qsio(:,i) = sio ./ (ksio(i) + sio); %%Qpar(:,i) = par ./ (kpar(i) + par); %Simpler function.

dinAve = median(kdin); sstAve = median(sst); QdinAve = dinAve ./ (kdin(i) + dinAve); QsstAve = Q10(i) * exp( (sstAve - sstref) / 10.0); % $$$ QdinAve = mean(Qdin(:,i)); % $$$ QsstAve = mean(Qsst(:,i)); Qpar(:,i) = 1.0 - exp(-1.0 * (alfa(i)*teta(i)*par)./(mupmax(i)*QdinAve*QsstAve)); %Original function.

grz_phy(:,i) = Qphy(:,i)*gammax(i); mup_din(:,i) = Qdin(:,i)*mupmax(i); mup_dip(:,i) = Qdip(:,i)*mupmax(i); mup_nh4(:,i) = Qnh4(:,i)*mupmax(i); mup_dop(:,i) = Qdop(:,i)*mupmax(i); mup_fe2(:,i) = Qfe2(:,i)*mupmax(i); mup_sio(:,i) = Qsio(:,i)*mupmax(i); end

figure(10) subplot(2,3,1) hp = plot(din,mup_din,'-'); set(hp,'linewidth',[2]); set(gca,'Xlim',[0 max(din) + eps]) set(gca,'Ylim',[0 5.00 + eps]) axis square xlabel('DIN') ylabel('DIN growth [d-1]') title('DIN') grid on subplot(2,3,1+3) hp = plot(dip,mup_dip,'-'); set(gca,'Xlim',[0 max(dip) + eps]) set(gca,'Ylim',[0 5.00 + eps]) set(hp,'linewidth',[2]); axis square xlabel('DIP') ylabel('DIP growth [d-1]') title('DIP') grid on subplot(2,3,2) hp = plot(nh4,mup_nh4,'-'); set(gca,'Xlim',[0 max(nh4) + eps]) set(gca,'Ylim',[0 5.00 + eps]) set(hp,'linewidth',[2]); axis square xlabel('NH4') ylabel('NH4 growth [d-1]') title('NH4') grid on subplot(2,3,2+3) hp = plot(dop,mup_dop,'-'); set(gca,'Xlim',[0 max(dop) + eps]) set(gca,'Ylim',[0 5.00 + eps]) set(hp,'linewidth',[2]); axis square

```
xlabel('DOP') ylabel('DOP growth [d-1]') title('DOP') grid on subplot(2,3,3) hp =
plot(sio,mup_sio,'-'); set(gca,'Xlim',[0 max(sio) + eps]) set(gca,'Ylim',[0 5.00 + eps])
set(hp,'linewidth',[2]); axis square xlabel('SiO') ylabel('SiO growth [d-1]') title('SiO')
legend('coccos','diatom','smallP','diazos','location','east') grid on subplot(2,3,3+3)
hp = plot(fe2,mup_fe2,'-'); set(gca,'Xlim',[0 max(fe2) + eps]) set(gca,'Ylim',[0
5.00 + eps]) set(hp,'linewidth',[2]); axis square xlabel('Fe2') ylabel('Fe2 growth
[d-1]') title('Fe2') grid on print('-dpng ','-r100','nissen_bg_DIN.png') print('-depsc','-
r100','nissen_bg_DIN.eps')

figure(15) subplot(2,2,1) hp = plot(par,Qpar,'-'); set(gca,'Xlim',[0 max(par)]);
set(gca,'Ylim',[0 1]) set(hp,'linewidth',[2]); axis square xlabel('PAR') ylabel('PAR
limit [n.d.]') title('PAR limit') grid on subplot(2,2,2) hp = plot(sst,Qsst,'-');
set(hp,'linewidth',[2]); axis square xlabel('SST') ylabel('SST limit [n.d.]') ti-
tle('SST limit') legend('coccos','diatom','smallP','diazos','location','north') grid on sub-
plot(2,2,3) hp = plot(phy,Qphy,'-'); set(gca,'Xlim',[0 max(phy)]); set(gca,'Ylim',[0 1.0]);
set(hp,'linewidth',[2]); axis square xlabel('PHY') ylabel('Graze limit [n.d.]') title('Grazing
limit') grid on subplot(2,2,4) hp = plot(phy,grz_phy,'-'); set(gca,'Xlim',[0 max(phy)]);
set(gca,'Ylim',[0 floor(max(gammax))]); set(hp(1),'linewidth',[2.0],'linestyle','-
'); set(hp(2),'linewidth',[2.0],'linestyle','-'); set(hp(3),'linewidth',[2.0],'linestyle',':');
set(hp(4),'linewidth',[2.0],'linestyle','-'); axis square xlabel('PHY') ylabel('Graze
rate [d-1]') title('Grazing rate') grid on print('-dpng ','-r100','nissen_bg_PAR.png')
print('-depsc','-r100','nissen_bg_PAR.eps')
```

%RELATIVE GROWTH RATIO (Diatoms / Coccos) 1D:

```
Rel_mupmax = log(mupmax(2)./mupmax(1)); Rel_Qdin = log(Qdin(:,2)./Qdin(:,1));
Rel_Qpar = log(Qpar(:,2)./Qpar(:,1)); Rel_Qsst = log(Qsst(:,2)./Qsst(:,1));

Rel_din = Rel_mupmax + Rel_Qdin; Rel_par = Rel_mupmax + Rel_Qpar; Rel_sst =
Rel_mupmax + Rel_Qsst;

figure(20) subplot(3,3,1) hp = plot(din,Rel_Qdin,'-'); set(gca,'Xlim',[0 max(din) +
```

```
eps]) set(gca,'Ylim',[-1.0 +1.0]) set(hp,'linewidth',[2]); xlabel('DIN') ylabel('Rel limit –
DIN [n.d.]')   title('DIN limitation') grid on subplot(3,3,2) hp = plot(par,Rel_Qpar,'-');
set(gca,'Xlim',[0 max(par) + eps]) set(gca,'Ylim',[-1.0 +1.0]) set(hp,'linewidth',[2]); xla-
bel('PAR') ylabel('Rel limit – PAR [n.d.]')   title('PAR limitation') grid on subplot(3,3,3)
hp = plot(sst,Rel_Qsst,'-'); set(gca,'Xlim',[0 max(sst) + eps]) set(gca,'Ylim',[-1.0 +1.0])
set(hp,'linewidth',[2]); xlabel('SST') ylabel('Rel limit – SST [n.d.]')   title('SST limita-
tion') grid on subplot(3,3,4) hp = plot(din,Rel_din,'-'); set(gca,'Xlim',[0 max(din) +
eps]) set(gca,'Ylim',[-1.0 +1.0]) set(hp,'linewidth',[2]); xlabel('DIN') ylabel('Rel growth
– DIN [n.d.]')   title('DIN growth') grid on subplot(3,3,5) hp = plot(par,Rel_par,'-
'); set(gca,'Xlim',[0 max(par) + eps]) set(gca,'Ylim',[-1.0 +1.0]) set(hp,'linewidth',[2]);
xlabel('PAR') ylabel('Rel growth – PAR [n.d.]')   title('PAR growth') grid on sub-
plot(3,3,6) hp = plot(sst,Rel_sst,'-'); set(gca,'Xlim',[0 max(sst) + eps]) set(gca,'Ylim',[-
1.0 +1.0]) set(hp,'linewidth',[2]); xlabel('SST') ylabel('Rel growth – SST [n.d.]')   ti-
tle('SST growth') grid on print('-dpng ','-r100','nissen_bg_RelMup_1D.png') print('-
depsc','-r100','nissen_bg_RelMup_1D.eps')
```

%ENVIRONMENTAL NICHE 3D:

```
[DIN,PAR,SST] = ndgrid(din,par,sst); [DIP,PAR,SST] = ndgrid(dip,par,sst);

for i = 1:4

[QDIN_i,QPAR_i,QSST_i]              =              ndgrid(Qdin(:,i),Qpar(:,i),Qsst(:,i));
[QDIP_i,QPAR_i,QSST_i] = ndgrid(Qdip(:,i),Qpar(:,i),Qsst(:,i));

QDIN(:,:,:,i) = QDIN_i; QDIP(:,:,:,i) = QDIP_i; QPAR(:,:,:,i) = QPAR_i; QSST(:,:,:,i) =
QSST_i;

end

for i = 1:4 QDIN_bis(:,:,:,i) = DIN ./ (kdin(i) + DIN); QDIP_bis(:,:,:,i) = DIP ./ (kdip(i) +
DIP); QPAR_bis(:,:,:,i) = PAR ./ (kpar(i) + PAR); QSST_bis(:,:,:,i) = Q10(i) * exp( (SST -
sstref) / 10.0); end
```

```
QLIM = QDIN .* QPAR .* QSST; for i = 1:4
```

```
QLIM_dinpar(:,:,i)  =  squeeze(QDIN(:,:,1,i)  .*  QPAR(:,:,1,i));  %QDIN-QPAR  ONLY
QLIM_dinsst(:,:,i)  =  squeeze(QDIN(:,1,:,i)  .*  QSST(:,1,:,i));  %QDIN-QSST  ONLY
QLIM_parsst(:,:,i) = squeeze(QPAR(1,:,:,i) .* QSST(1,:,:,i)); %QPAR-QSST ONLY
```

```
MUP(:,:,:,i) = QLIM(:,:,:,i) * mupmax(i); end
```

%RELATIVE GROWTH RATIO 3D:

%—

```
Rel_QDIN        =        log(QDIN(:,:,:,2)./QDIN(:,:,:,1));        Rel_QDIP        =
log(QDIP(:,:,:,2)./QDIP(:,:,:,1));      Rel_QPAR      =      log(QPAR(:,:,:,2)./QPAR(:,:,:,1));
Rel_QSST = log(QSST(:,:,:,2)./QSST(:,:,:,1)); Rel_QLIM = Rel_QDIN + Rel_QPAR +
Rel_QSST;
```

```
Rel_QLIM_dinpar = squeeze(Rel_QDIN(:,:,1) + Rel_QPAR(:,:,1)); %QDIN-QPAR
ONLY Rel_QLIM_dinsst = squeeze(Rel_QDIN(:,1,:)  + Rel_QSST(:,1,:)); %QDIN-
QSST ONLY Rel_QLIM_parsst = squeeze(Rel_QPAR(1,:,:)  + Rel_QSST(1,:,:));
%QPAR-QSST ONLY
```

%—

```
Rel_MUP_din = Rel_mupmax + Rel_QDIN; Rel_MUP_par = Rel_mupmax +
Rel_QPAR; Rel_MUP_sst = Rel_mupmax + Rel_QSST; Rel_MUP_tot = Rel_mupmax
+ Rel_QLIM;
```

```
Rel_MUP_dinpar    =    Rel_mupmax    +    Rel_QLIM_dinpar;       Rel_MUP_dinsst
=  Rel_mupmax  +  Rel_QLIM_dinsst;    Rel_MUP_parsst  =  Rel_mupmax  +
Rel_QLIM_parsst;
```

%—

%RELATIVE GRAZING RATIO ON COCCOS AND DIATOMS (ONLY) – 2D: CLRmax =

max(gammax) / min(zgrz); %Maximum clearance rate (m3 * mmolN-1 * d-1) gammaxC = gammax(1); %Coccos gammaxD = gammax(2); %Diatom zgrzC = zgrz(1); %Coccos zgrzD = zgrz(2); %Diatom

[PC,PD] = ndgrid(pc,pd);

GRZ_COCO = gammaxC * (PC ./ (zgrzC + PC + PD)); %OK GRZ_DIAT = gammaxD * (PD ./ (zgrzD + PC + PD)); %OK

GRZ_COCO_WRONG = gammaxC * (PC ./ (zgrzC + PC)); %WRONG! GRZ_DIAT_WRONG = gammaxD * (PD ./ (zgrzD + PD)); %WRONG!

CLR_COCO = GRZ_COCO./PC; %clearance rate on coccos (m3 * mmolN-1 * d-1) CLR_DIAT = GRZ_DIAT./PD; %clearance rate on diatom (m3 * mmolN-1 * d-1) Rel_CLR = log(CLR_COCO./CLR_DIAT);

CLR_COCO_WRONG = GRZ_COCO_WRONG./PC; %clearance rate on coccos (m3 * mmolN-1 * d-1) CLR_DIAT_WRONG = GRZ_DIAT_WRONG./PD; %clearance rate on diatom (m3 * mmolN-1 * d-1) Rel_CLR_WRONG = log(CLR_COCO_WRONG./CLR_DIAT_WRONG);

%»> DY = diff(GRZ_COCO_WRONG,[],1); DX = diff(PC,[],1); SLOPE = DY./DX; CLR_COCO_ALONE = [SLOPE;SLOPE(end,:)];

DY = diff(GRZ_DIAT_WRONG,[],2); DX = diff(PD,[],2); SLOPE = DY./DX; CLR_DIAT_ALONE = [SLOPE,SLOPE(:,end)];

figure(1) %just checking things. subplot(2,2,1) plot(CLR_COCO_WRONG(:),CLR_COCO_ALONE(:),'*') set(gca,'Xlim',[0    4]) set(gca,'Ylim',[0    4]) grid on subplot(2,2,2) plot(CLR_DIAT_WRONG(:),CLR_DIAT_ALONE(:),'*') set(gca,'Xlim',[0    4]) set(gca,'Ylim',[0 4]) grid on %«<

%ALTERNATIVE METHOD: PHY(:,:,1) = PC; PHY(:,:,2) = PD; PHYTOT = sum(PHY,3); for i = 1:2 %Coccos and Diatoms (only) QPHY(:,:,i) = PHY(:,:,i) ./ (zgrz(i) + PHYTOT);

```
GRZ(:,:,i) = QPHY(:,:,i)*gammax(i); end
```

Rel_gammax = log(gammax(1)./gammax(2)); Rel_QPHY = log(QPHY(:,:,1)./QPHY(:,:,2)); Rel_GRZ = Rel_gammax + Rel_QPHY;

CLR = GRZ ./ PHY; %clearance rate (m3 * mmolN-1 * d-1) GRZ_COCO_bis = GRZ(:,:,1); %specific grazing rate on coccos (d-1) GRZ_DIAT_bis = GRZ(:,:,2); %specific grazing rate on diatom (d-1) CLR_COCO_bis = CLR(:,:,1); %clearance rate on coccos (m3 * mmolN-1 * d-1) CLR_DIAT_bis = CLR(:,:,2); %clearance rate on diatom (m3 * mmolN-1 * d-1) Rel_GRZ_bis = log(GRZ_COCO_bis./GRZ_DIAT_bis); Rel_CLR_bis = log(CLR_COCO_bis./CLR_DIAT_bis);

QPHY_TOT = sum(QPHY,3); GRZ_TOT = sum(GRZ,3); CLR_TOT = sum(CLR,3);

figure(100) subplot(2,2,1) imagesc(din,par,QLIM_dinpar(:,:,1)') set(gca,'YDir','normal') caxis([0.0 1.0]) xlabel('DIN') ylabel('PAR') title('Coccos Limit (DIN + PAR)') colormap(cmap) colorbar grid on subplot(2,2,2) imagesc(din,par,QLIM_dinpar(:,:,2)') set(gca,'YDir','normal') caxis([0.0 1.0]) xlabel('DIN') ylabel('PAR') title('Diatom Limit (DIN + PAR)') colormap(cmap) colorbar grid on subplot(2,2,3) imagesc(din,par,Rel_QLIM_dinpar') set(gca,'YDir','normal') caxis([-0.4 +0.4]) xlabel('DIN') ylabel('PAR') title('Rel Limit (DIN + PAR)') colormap(cmap) colorbar grid on subplot(2,2,4) imagesc(din,par,Rel_MUP_dinpar') set(gca,'YDir','normal') caxis([-0.4 +0.4]) xlabel('DIN') ylabel('PAR') title('Rel Growth (DIN + PAR)') colormap(cmap) colorbar grid on print('-dpng ','-r100','nissen_bg_RelMup_Limit-vs-Rate_2D.png') print('-depsc','-r100','nissen_bg_RelMup_Limit-vs-Rate_2D.eps')

figure(110) subplot(3,3,1) imagesc(pc,pd,QPHY(:,:,1)') set(gca,'YDir','normal') caxis([0.0 1.0]) xlabel('PC') ylabel('PD') title('Grazing Limit (PC)') colormap(cmap) colorbar grid on subplot(3,3,2) imagesc(pc,pd,QPHY(:,:,2)') set(gca,'YDir','normal') caxis([0.0 1.0]) xlabel('PC') ylabel('PD') title('Grazing Limit (PD)') colormap(cmap) colorbar grid on subplot(3,3,3) imagesc(pc,pd,QPHY_TOT') set(gca,'YDir','normal') caxis([0.0 1.0]) xlabel('PC') ylabel('PD') title('Grazing Limit (PC + PD)') col-
ormap(cmap) colorbar grid on subplot(3,3,1+3) imagesc(pc,pd,GRZ(:,:,1)')
set(gca,'YDir','normal') caxis([0.0 floor(max(gammax))]) xlabel('PC') ylabel('PD')
title('Grazing Rate (PC)') colormap(cmap) colorbar grid on subplot(3,3,2+3) im-
agesc(pc,pd,GRZ(:,:,2)') set(gca,'YDir','normal') caxis([0.0 floor(max(gammax))])
xlabel('PC') ylabel('PD') title('Grazing Rate (PD)') colormap(cmap) colorbar grid
on subplot(3,3,3+3) imagesc(pc,pd,GRZ_TOT') set(gca,'YDir','normal') caxis([0.0
floor(max(gammax))]) xlabel('PC') ylabel('PD') title('Grazing Rate (PC + PD)')
colormap(cmap) colorbar grid on subplot(3,3,1+6) imagesc(pc,pd,CLR(:,:,1)')
set(gca,'YDir','normal') caxis([0.0 CLRmax/2]) xlabel('PC') ylabel('PD') ti-
tle('Clearance Rate (PC)') colormap(cmap) colorbar grid on subplot(3,3,2+6) im-
agesc(pc,pd,CLR(:,:,2)') set(gca,'YDir','normal') caxis([0.0 CLRmax/2]) xlabel('PC')
ylabel('PD') title('Clearance Rate (PD)') colormap(cmap) colorbar grid on sub-
plot(3,3,3+6) imagesc(pc,pd,CLR_TOT') set(gca,'YDir','normal') caxis([0.0 CLRmax/2])
xlabel('PC') ylabel('PD') title('Clearance Rate (PC + PD)') colormap(cmap) colorbar
grid on print('-dpng ','-r100','nissen_bg_RelGrz_Limit-vs-Rate_2D.png') print('-depsc','-
r100','nissen_bg_RelGrz_Limit-vs-Rate_2D.eps')

figure(40) subplot(3,3,1) himg = imagesc(din,par,Rel_QLIM_dinpar); %QDIN-QPAR
ONLY. set(gca,'YDir','normal') %%axis square caxis([-0.4 +0.4]) xlabel('DIN') yla-
bel('PAR') title('Rel Limit (DIN + PAR)') %%colormap(redbluecmap) colormap(cmap)
colorbar grid on subplot(3,3,2) himg = imagesc(din,sst,Rel_QLIM_dinsst); %QDIN-
QSST ONLY. set(gca,'YDir','normal') %%axis square caxis([-0.4 +0.4]) xlabel('DIN')
ylabel('SST') title('Rel Limit (DIN + SST)') colormap(cmap) colorbar grid on
subplot(3,3,3) himg = imagesc(par,sst,Rel_QLIM_parsst); %QPAR-QSST ONLY.
set(gca,'YDir','normal') %%axis square caxis([-0.4 +0.4]) xlabel('PAR') ylabel('SST')
title('Rel Limit (PAR + SST)') colormap(cmap) colorbar grid on subplot(3,3,4) himg
= imagesc(din,par,Rel_MUP_dinpar); %QDIN-QPAR ONLY. set(gca,'YDir','normal')
%%axis square caxis([-0.4 +0.4]) xlabel('DIN') ylabel('PAR') title('Rel Growth
(DIN + PAR)') %%colormap(redbluecmap) colormap(cmap) colorbar grid on
subplot(3,3,5) himg = imagesc(din,sst,Rel_MUP_dinsst); %QDIN-QSST ONLY.

set(gca,'YDir','normal') %%axis square caxis([-0.4 +0.4]) xlabel('DIN') ylabel('SST') title('Rel Growth (DIN + SST)') colormap(cmap) colorbar grid on subplot(3,3,6) himg = imagesc(par,sst,Rel_MUP_parsst); %QPAR-QSST ONLY. set(gca,'YDir','normal') %%axis square caxis([-0.4 +0.4]) xlabel('PAR') ylabel('SST') title('Rel Growth (PAR + SST)') colormap(cmap) colorbar grid on %»> % $$$ subplot(2,2,4) % $$$ caxis([-0.4 +0.4]) % $$$ text(0.4,0.8,'Diatom win (red)') % $$$ text(0.4,0.2,'Coccos win (blue)') % $$$ set(gca,'Xtick',[],'Xticklabel',[]) % $$$ set(gca,'Ytick',[],'Yticklabel',[]) % $$$ colormap(cmap) % $$$ colorbar % $$$ axis off %«< print('-dpng ','-r100','nissen_bg_RelMup_2D.png') print('-depsc','-r100','nissen_bg_RelMup_2D.eps')

figure(70) subplot(3,3,1) himg = imagesc(pc,pd,CLR_COCO_WRONG'); set(gca,'YDir','normal') caxis([0.0 CLRmax/2]) %%axis square xlabel('PC – WRONG!') ylabel('PD') title('Clearance rate – PC') colormap(cmap) colorbar grid on subplot(3,3,2) himg = imagesc(pc,pd,CLR_DIAT_WRONG'); set(gca,'YDir','normal') caxis([0.0 CLRmax/2]) %%axis square xlabel('PC – WRONG!') ylabel('PD') title('Clearance rate – PD') colormap(cmap) colorbar grid on subplot(3,3,3) himg = imagesc(pc,pd,Rel_CLR_WRONG'); %if Rel_CLR is positive, then PC is being grazed faster than PD. set(gca,'YDir','normal') caxis([-0.4 +0.4]) %%axis square xlabel('PC – WRONG!') ylabel('PD') title('Rel Clearance (C:D)') colormap(cmap) colorbar grid on subplot(3,3,1+3) himg = imagesc(pc,pd,CLR_COCO'); set(gca,'YDir','normal') caxis([0.0 CLRmax/2]) %%axis square xlabel('PC') ylabel('PD') title('Clearance rate – PC') colormap(cmap) colorbar grid on subplot(3,3,2+3) himg = imagesc(pc,pd,CLR_DIAT'); set(gca,'YDir','normal') caxis([0.0 CLRmax/2]) %%axis square xlabel('PC') ylabel('PD') title('Clearance rate – PD') colormap(cmap) colorbar grid on subplot(3,3,3+3) himg = imagesc(pc,pd,Rel_CLR'); %if Rel_CLR is positive, then PC is being grazed faster than PD. set(gca,'YDir','normal') caxis([-0.4 +0.4]) %%axis square xlabel('PC') ylabel('PD') title('Rel Clearance (C:D)') colormap(cmap) colorbar grid on subplot(3,3,1+6) imagesc(pc,pd,Rel_QPHY') set(gca,'YDir','normal') caxis([-0.4 +0.4]) axis square xlabel('PC') ylabel('PD') title('Rel Limit (PC:PD)') colormap(cmap) colorbar grid on subplot(3,3,2+6) im-

```
agesc(pc,pd,Rel_GRZ') set(gca,'YDir','normal') caxis([-0.4 +0.4]) axis square xla-
bel('PC') ylabel('PD') title('Rel Grazing (PC:PD)') colormap(cmap) colorbar grid
on subplot(3,3,3+6) imagesc(pc,pd,Rel_CLR') set(gca,'YDir','normal') caxis([-0.4
+0.4]) axis square xlabel('PC') ylabel('PD') title('Rel Clearance (PC:PD)') col-
ormap(cmap) colorbar grid on print('-dpng ','-r100','nissen_bg_RelGrz.png') print('-
depsc','-r100','nissen_bg_RelGrz.eps')

return

publish('nissen_bg_myreview','pdf')
```

—

Please also note the supplement to this comment:
https://www.biogeosciences-discuss.net/bg-2018-157/bg-2018-157-RC2-
supplement.pdf

---

## Author Comment (AC1) · 1 Oct 2018

**Answer to referee #1:**

We thank referee #1 for taking the time to review our manuscript and for his/her valuable comments and suggestions that have significantly improved the quality of our manuscript.

Below, we include our detailed answers to all comments and questions.

**Answers to general comments (GC):**

**General Comment #1:**
*Non-grazing mortality – It is not explicitly discussed in the paper as to what the authors consider this to be. Viral lysis is seen as a major mortality pathway for coccolithophore (bloom) communities and so is this what the authors mean by this terminology? How is it parameterised and does it fairly represent viral mortality or (e.g.) programmed cell death? Not representing (or discussing) such a major mortality pathway seems like a limitation of the study, but a necessary limitation due to the uncertainties around viral mortality dynamics and its role in the Southern Ocean. The authors should include viral mortality in their discussion over model limitations, as well as directions for future field observations.*

**Answer to GC1:**
We thank reviewer for this important comment. While not explicitly stated in the original version of the manuscript, BEC implicitly accounts for the effect of viral lysis in the non-grazing mortality term for its phytoplankton functional types (see Eq. B14 in the original manuscript). According to Moore et al. (2002) "this term [non-grazing mortality] would include losses due to viral lysis […], as well as internal respiration/degradation, and excretion." In BEC, the non-grazing mortality rate of phytoplankton scales linearly with their production, i.e. a constant fraction of photosynthetic production is immediately lost due to this term at every time step of integration. We acknowledge that BEC does not represent non-linear increases in losses due to viral lysis towards the end of coccolithophore blooms as suggested in the literature based on observational evidence **(see e.g. Lehahn et al., 2014, Evans et al., 2007, Brussard et al., 2004)**. To better constrain model simulations, future observational studies should investigate whether viral lysis is as important for the termination of coccolithophore blooms in the Southern Ocean, as it has been shown to be e.g. in the North Atlantic **(Lehahn et al., 2014)**. To the best of our knowledge, there are only two studies from the Southern Ocean assessing the relative importance of viral lysis and grazing by zooplankton as sinks for phytoplankton biomass, and both point to a minor importance of viral lysis in this ocean region (**Evans et al., 2012, Brussard et al., 2008**), but unfortunately, none of these studies explicitly assessed their importance for coccolithophore biomass dynamics.

We agree with the reviewer that this process should be included in the discussion of limitations of our study, and we have changed section 5.4 to include the following sentences:

"While the importance of viral lysis has been shown for the termination of coccolithophore blooms in the North Atlantic (e.g. Lehahn et al., 2014, Evans et al., 2007, Brussaard et al., 2004), to the best of our knowledge, there are only two studies from the SO assessing the relative importance of viral lysis and grazing by zooplankton as sinks for phytoplankton biomass, and both point to a minor importance of viral lysis in this ocean region (Evans and Brussaard, 2012; Brussaard et al., 2008). However, none of these studies explicitly assessed the importance for coccolithophore biomass dynamics, which should be investigated in future observational studies."

**General Comment #2:**

*Importance of bottom-up and top-down controls – The conclusion that both types of controls need to be considered when examining phytoplankton (and coccolithophore) population dynamics and biogeography is very important point to be made. However, the statement is not limited to the Southern Ocean and is relevant across the full bio- geographical range of coccolithophores.*

**Answer to GC2:**

We agree with the reviewer that our conclusion that both bottom-up and top-down factors are important when assessing phytoplankton dynamics in general (or diatom and coccolithophore dynamics in particular) is not per-se restricted to the Southern Ocean, but the relative importance of both controls may be regionally varying, an effect which we cannot assess with our Southern Ocean model setup. We thank the reviewer for this comment and include a statement along these lines in the conclusion section:

"Top-down factors are important regulators of phytoplankton biomass dynamics not only in the SO, but globally (Behrenfeld, 2014). Being restricted to the SO by the regional model setup used here, future work with global models should better quantify regional variability in the relative importance of bottom-up and top-down factors in controlling phytoplankton biogeography."

**General Comment #3:**

*Coccolithophores/Emiliania huxleyi – Do the authors consider they have parameterised their model to describe the whole coccolithophore community, or rather that they are limited to E. huxleyi dynamics in the Southern Ocean? For this region it is relatively simple as E. huxleyi dominates (to almost monospecific levels depending on latitude). Within the authors recognised limitations, discussion of this point should be considered, especially if there are aspirations to expand such modelling efforts to low- latitude highly-diverse coccolithophore communities.*
*Related to this point, the 400% overestimation of coccolithophore biomass (Pg 19, Lns 25-26) applies to the whole coccolithophore diversity, and in diverse communities would indeed lead to significant issues, however in the E. huxleyi dominated Southern Ocean such issues are far less extreme. There are also numerous estimates of E. huxleyi cell biomass (and even B/C biomass), which are in agreement (and don't vary by 400%).*

**Answer to GC3:**

Thank you for this comment. We focused our literature research on E. huxleyi literature - the by far most dominant coccolithophore species in our model domain (e.g. **Saavedra-Pellitero et al., 2014**). Parameter values require adjustment when expanding any regional modelling effort to the global scale, as different coccolithophores show a wide range of growth and calcification rates and environmental dependencies (see recent review by **Krumhardt et al, 2017**). With regard to the biomass validation, we have clarified a few points in the manuscript to address the reviewer's comment: We are not, as suggested by the reviewer, overestimating coccolithophore biomass by 400%. This is the uncertainty range (conversion error) obtained when converting cell count observations to carbon biomass, given reported size ranges for E. huxleyi (**O'Brien et al., 2013**). Our model estimates of coccolithophore biomass are within this uncertainty of the biomass observations (see updated Fig. 1d in the manuscript, **Fig. R1** below). As stated correctly by the reviewer, the uncertainty is indeed smaller for almost mono-specific coccolithophore communities as present in the Southern Ocean as compared to more diverse communities including a larger size range than E. huxleyi alone (see Fig 8b in **O'Brien et al., 2013**).

We have clarified the description of the conversion from cell counts observations to biomass estimates, as well as the calculation of the uncertainty range in the supplementary material in section S1 as follows:

"Based on available information in the literature, each species is first assigned an idealized shape (e.g. sphere for *E. huxleyi*), as well as a mean size (e.g. mean coccosphere diameter for *E. huxleyi*).

Assuming the cytoplasm diameter to be 60% of the coccosphere diameter, we then calculate the mean biovolume of each cell. To get estimates of carbon biomass for each cell, the biovolume is ultimately multiplied with the specific carbon conversion factors from Menden-Deuer and Lessard (2000). The uncertainty range of this conversion is obtained by repeating the conversion using the minimum and maximum reported diameter for each species, respectively, and reporting the uncertainty range in percent of the mean biomass estimate. "

[Figure]

**Figure R1:** Modified version of panel 1d &f in the manuscript with standard deviation of observations added as the grey bars to illustrate variability. We have included this version of the panels in the revised manuscript.

**Answers to specific comments (SC):**

**SC1**: *Pg 1, Ln 16: Please specify 'Ocean Acidification' rather than just 'acidification'.*
Thank you, corrected as suggested.

**SC2**: *Pg 1, Ln 22: It is not just the ratio of calcifying to silicifying phytoplankton that is crucial to consider, it is the ratio of calcifying to non-calcifying (organic only) phytoplankton.*
Thank you, corrected as suggested.

**SC3**: *Pg 2, Lns 4-5: It should be recognised that all these references are model based estimates rather than field estimates, and also take varying ways to parameterise coccolithophore production. See also pg 19, ln 21 – here it should also be recognised that these low estimates of coccolithophore NPP are derived from model studies with diverse parameterisations of coccolithophore calcification.*
Thank you, we have clarified this part of the introduction. It now reads:

"In comparison, coccolithophores contribute less to biomass (≈0.04-6%, Buitenhuis et al., 2013b) and to global NPP (0.4-17%, model-derived estimates using a variety of coccolithophore parametrizations, see O'Brien, 2015; Jin et al., 2006; Moore et al., 2004; Gregg and Casey, 2007a). "

**SC4**: *Pg 2, Lns 10-11: Cell densities of 2.4 x 103 cells mL-1 have to be for the Patagonian Shelf bloom and are really (really) high whilst cell densities elsewhere in the Atlantic sector of the SO are much (much) lower. The authors should make it clear that these high numbers are from bloom waters.*
Thank you for pointing this out. We have modified this part of the manuscript accordingly in the introduction:
"In-situ observations confirmed coccolithophore abundances of up to $2.4 \cdot 10^3$ cells ml$^{-1}$ in the Atlantic sector (blooms on the Patagonian Shelf), up to $3.8 \cdot 10^2$ cells ml$^{-1}$ in the Indian sector (Balch

et al., 2016) and up to $5.4 \cdot 10^2$ cells ml$^{-1}$ in the Pacific sector of the SO (Cubillos et al., 2007) with *Emiliania huxleyi* being the dominant species (Balch et al., 2016; Saavedra-Pellitero et al., 2014)."

**SC5**: *Pg 3, Ln 14: Please make clear that zooplankton grazing includes both micro- and macro-zooplankton (rather than just the latter).*
Thank you, we have clarified this as suggested.

**SC6**: *Pg 4, Lns 6-7: 'Coccolithophores grow well at high light intensities and at a range of different temperatures, but have been shown to be light-inhibited at low light levels' – does this statement fit coccolithophores as a group or just E. huxleyi?*
To the best of our knowledge, the only studies assessing this effect were conducted with E.huxleyi (see **Zondervan et al., 2007**, and references therein). We note, however, that the inhibition threshold of 1 W m$^{-2}$ is likely of very minor importance for the simulated coccolithophore biogeography, as it represents a very low light level that is not attained in the surface layers during the growing months.

**SC7**: *Pg 5, Ln 19: What is the justification (reference) for using such extremely low carbon to chlorophyll ratios (3 to 5)? These lead to extremely chlorophyll-rich phytoplankton cells whereas ratios are typically 10 to 20 times higher. Are these based on Southern Ocean studies?*
We thank the reviewer for pointing out that we use rather uncommon units in the original manuscript, which may have led to the confusion with regard to the reported carbon-to-chlorphyll values. In fact, we initialize with carbon-to-chlorophyll ratios of 3 and 5 mmol C / mg chl for diatoms and all other phytoplankton types, respectively, admittedly a unit rather uncommon when it comes to reporting carbon-to-chlorophyll ratios in phytoplankton (see e.g. **Thomalla et al., 2017**). These numbers correspond to 36 and 60 mg C / mg chl, respectively, which is in close agreement with the range of ratios suggested by the reviewer and in the literature **(Sathyendranath et al., 2009, Thomalla et al., 2017)**. We have changed the respective line in section 2.2 of the manuscript to report the ratios in mg C/mg chl, have specified units more clearly, and have added a reference to justify the higher carbon-to-chlorophyll ratio of diatoms compared to the other phytoplankton types **(Sathyendranath et al., 2009)**. It now reads:

"Phytoplankton carbon biomass fields are then derived using a constant carbon-to-chlorophyll ratio of 36 mg C (mg chl)$^{-1}$ for diatoms and 60 mg C (mg chl)$^{-1}$ for all other PFTs (Sathyendranath et al., 2009)."

**Figure 2:** *Colours seem to have changed on panel (a) – blue looks olive green and grey looks to be light green?*
Unfortunately, we cannot identify issues with the color scale in Figure 2 in the published version of the manuscript, and are thus unable to track the origin of this comment.

**SC8**: *Pg 14, Ln 4: extra 'a' in this sentence.*
Thank you for pointing out this typo. We deleted the extra "a".

**SC9**: *Pg 20, Ln 30: A key statement – 'coccolithophores appear to be of minor importance for global oceanic organic carbon fixation'. Many in situ studies agree with such small contributions to phytoplankton biomass or primary production in the Southern Ocean (including those already cited in the paper: Smith et al., 2017; Charalampopoulou et al., 2016; Poulton et al., 2013; Hinz et al., 2012).*
Thanks for this comment. We have modified this sentence accordingly, and it now states:

"Contributing only a few percent to global NPP, coccolithophores appear to be of minor importance for global oceanic organic carbon fixation, in agreement with previous observational studies form the SO (Smith et al., 2017; Charalampopoulou et al., 2016; Poulton et al., 2013; Hinz et al., 2012). "

**SC10**: *Pg 24, Lns 22: 'Based on our findings, future SO in-situ studies should consider both bottom-up and top-down factors when assessing coccolithophore biogeography in space and time'. This statement should not be limited to just the Southern Ocean.*

Please see answer to GC2 above.

**SC11**: *Pg 25, Lns 19 and 22-23: As well as multiple trophic levels (and trophic cascades), what about non-grazing mortality (i.e. viral mortality?). This is not discussed any- where in the paper and the omission of viral driven population dynamics needs to be addressed in the limitations.*
Please see answer to GC1 above.

**Cited literature:**

Brussaard, C. P. D. (2004). Viral control of phytoplankton populations - a review. Journal of Eukaryotic Microbiology, 52(6), 549–551. https://doi.org/10.1111/j.1550-7408.2005.000vol-cont.x

Brussaard, C. P. D., Timmermans, K. R., Uitz, J., & Veldhuis, M. J. W. (2008). Virioplankton dynamics and virally induced phytoplankton lysis versus microzooplankton grazing southeast of the Kerguelen (Southern Ocean). Deep Sea Research Part II: Topical Studies in Oceanography, 55(5–7), 752–765. https://doi.org/10.1016/j.dsr2.2007.12.034

Evans, C., & Brussaard, C. P. D. (2012). Viral lysis and microzooplankton grazing of phytoplankton throughout the Southern Ocean. Limnology and Oceanography, 57(6), 1826–1837. https://doi.org/10.4319/lo.2012.57.6.1826

Evans, C., Kadner, S. V, Darroch, L. J., Wilson, W. H., Liss, P. S., & Malin, G. (2007). The relative significance of viral lysis and microzooplankton grazing as pathways of dimethylsulfoniopropionate (DMSP) cleavage: An Emiliania huxleyi culture study. *Limnology and Oceanography*, *52*(3), 1036–1045. https://doi.org/10.4319/lo.2007.52.3.1036

Krumhardt, K. M., Lovenduski, N. S., Iglesias-Rodriguez, M. D., & Kleypas, J. A. (2017). Coccolithophore growth and calcification in a changing ocean. *Progress in Oceanography*, *159*(June), 276–295. https://doi.org/10.1016/j.pocean.2017.10.007

Lehahn, Y., Koren, I., Schatz, D., Frada, M., Sheyn, U., Boss, E., Efrati, S., Rudich, Y., Trainic, M., Sharoni, S., Laber, C., DiTullio, G.R., Coolen, M.J.L., Martins, A.M., Van Mooy, B.A.S., Bidle, K.D., Vardi, A. (2014). Decoupling Physical from Biological Processes to Assess the Impact of Viruses on a Mesoscale Algal Bloom. Current Biology, 24(17), 2041–2046. https://doi.org/10.1016/j.cub.2014.07.046

O'Brien, C. J., Peloquin, J. A., Vogt, M., Heinle, M., Gruber, N., Ajani, P., Andruleit, H., Arístegui, J., Beaufort, L., Estrada, M., Karentz, D., Kopczyńska, E., Lee, R., Poulton, A. J., Pritchard, T., Widdicombe, C. (2013). Global marine plankton functional type biomass distributions: coccolithophores. *Earth System Science Data*, *5*(2), 259–276. https://doi.org/10.5194/essd-5-259-2013

Saavedra-Pellitero, M., Baumann, K.-H., Flores, J.-A., & Gersonde, R. (2014). Biogeographic distribution of living coccolithophores in the Pacific sector of the Southern Ocean. *Marine Micropaleontology*, *109*, 1–20. https://doi.org/10.1016/j.marmicro.2014.03.003

Sathyendranath, S., Stuart, V., Nair, A., Oka, K., Nakane, T., Bouman, H., Forget, M.H., Maass, H., Platt, T. (2009). Carbon-to-chlorophyll ratio and growth rate of phytoplankton in the sea. Marine Ecology Progress Series, 383, 73–84. https://doi.org/10.3354/meps07998

Thomalla, S. J., Ogunkoya, A. G., Vichi, M., & Swart, S. (2017). Using Optical Sensors on Gliders to Estimate Phytoplankton Carbon Concentrations and Chlorophyll-to-Carbon Ratios in the

Southern Ocean. Frontiers in Marine Science, 4(February), 1–19.
https://doi.org/10.3389/fmars.2017.00034

Zondervan, I. (2007). The effects of light, macronutrients, trace metals and CO2 on the production of calcium carbonate and organic carbon in coccolithophores—A review. *Deep Sea Research Part II: Topical Studies in Oceanography*, *54*(5–7), 521–537. https://doi.org/10.1016/j.dsr2.2006.12.004

---

## Author Comment (AC2) · 1 Oct 2018

**Answer to referee #2:**

We thank Dr. Sergio Vallina for taking the time to review our manuscript in extraordinary depth and detail and for his comments and suggestions. We have conducted major revisions of the originally submitted manuscript in response to the majority of the comments: We have carefully revised all text in order to clarify the structure of our model, the rationale behind chosen parametrizations and parameters, the choice and setup of the sensitivity simulations, the presentation of our analysis framework, as well as the interpretation of results. While we agree with many of the issues raised by the reviewer, we beg to differ in a few areas. In particular, we came to the conclusion that misunderstandings of our results occurred. Furthermore, we added three figures to the supplementary material to visualize chosen parametrizations with respect to phytoplankton growth and grazing, as well as simulated total chlorophyll seasonality and phytoplankton carbon-to-chlorophyll ratios.

All comments have helped to enormously improve the quality of our manuscript. Below, we include our detailed answers to all comments/questions.

**Answers to general comments (GC):**

**General Comment #1:**
*Specifically the authors call i) "relative growth rate" what in reality is a "relative growth limitation" term, which is not the same thing; and ii) "relative grazing rate" what in reality is a "relative clearance rate", which is not the same thing either. Further, they make inferences on the relative effect of bottom-up versus top-down effects based on comparing these two concepts (i and ii) while they are not comparable -- basically because they have different units (before making the log10 of their ratio for coccos : diatoms). What they need to compare is log10(x/y) where x has units of days-1 for coccos and y has units of days-1; BOTH for growth rate and grazing rate. At the moment for relative growth they are using x and y using non dimensional units (n.d), and for the relative grazing they are using x and y using clearance rate units (m3 \* m-3 \* d-1). To be consistent the growth limitation terms must be multiplied by maximum growth rate (mupmax; d-1) and the grazing clearance rate must be multiplied by zooplankton biomass (Z; mmol \* m-3). This will make those processes (bottom-up vs. top-down) comparable because they will have the same units (d-1) before making the ratio and taking the log10. The two major things the author will notice are: 1) the specific growth rate (d-1) of diatoms is larger than coccos for most of the environmental conditions; only at very narrow window of small nutrient concentration will coccos outcompete diatoms; 2) the specific grazing rate (d-1) on diatoms is ALWAYS smaller than on coccos (for a constant zooplankton biomass).*

**Answer to GC1:**
We thank the reviewer for pointing out that the units were not specified explicitly enough in the methods section of our original manuscript, which must have caused certain misunderstandings with regard to the equations used, and thus the interpretation of the results specified above.

We fully agree with the reviewer that for both the relative growth ratio and for the relative grazing ratio, both numerator and denominator should be in [$d^{-1}$]. We double-checked the units of the equations as reported in the manuscript and find that the specific growth rates of coccolithophores and diatoms (whose ratio we define as the relative growth ratio; **Hashioka et al., 2013**), as well as the biomass-normalized specific grazing rates (ratio defined as the relative grazing ratio; **Hashioka et al., 2013**) are indeed in $d^{-1}$ and *not*, as suggested by the reviewer, dimensionless for the relative growth ratio and mol m$^{-3}$ d$^{-1}$ for the relative grazing rate:

The specific growth rates of coccolithophores and diatoms in Eq. 3 in the manuscript are in units of d$^{-1}$ because $\mu_{max}$ is in d$^{-1}$, so that Eq. 4 takes the log ratio of d$^{-1}$/d$^{-1}$.

The specific grazing rates in Eq. 5 are in units of mmol C m$^{-3}$ d$^{-1}$ because $Z$ is in mmol C m$^{-3}$ and $\gamma_{max}$ is in d$^{-1}$. Normalizing this term by phytoplankton biomass $P$ [mmol C m$^{-3}$] in Eq. 6 results in the

calculation of the log ratio of a unitless quantity ($d^{-1}/d^{-1}$). Hence, we do show the relative growth ratio (not the relative growth limitation), as well as the relative grazing ratio and apologize for the confusion.

To clarify these units, we have changed **section 3** in the manuscript by adding the units of each term (highlighted in bold):

"In BEC, phytoplankton biomass $P^i$ (**[mmol C m$^{-3}$]**, $i \in \{C,D,SP,N\}$) is the balance of growth ($\mu^i$) and loss terms (grazing by zooplankton $\gamma g^i$, non-grazing mortality $\gamma m^i$ and aggregation $\gamma a^i$, see Appendix B for a full description of the model equations regarding phytoplankton growth and loss terms): […] with the specific phytoplankton growth $\mu^i$ **[d$^{-1}$]** being dependent on the maximum growth rate $\mu^i$max (**[d$^{-1}$]**, Table 1), temperature ($f^i(T)$, Eq. B5), nutrient availability ($g^i(N)$, Eq. B8; nitrate, ammonium, phosphorus and iron for all PFTs, silicate for diatoms only) and light levels ($h^i(I)$, Eq. B9; following the growth model by Geider et al. (1998)) : […] The specific grazing rate $\gamma g^i$ **[mmol C m$^{-3}$ d$^{-1}$]** of the generic zooplankton on the respective phytoplankton i is described by […] with Z being zooplankton biomass **[mmol C m$^{-3}$]**, $f^Z(T)$ the temperature scaling unction (Eq.B13), $\gamma^i$max the maximum growth rate of zooplankton when feeding on phytoplankton i (**[d$^{-1}$]**,Table1), $z^i$grz the respective half-saturation coefficient for ingestion (**[mmol C m$^{-3}$]**, Table 1) and $P^{\prime i}$ the phytoplankton biomass **[mmol C m$^{-3}$]**, which was corrected for a loss threshold below which no losses occur (Eq. B11)."

Additionally, we made the definition of the relative growth ratio clearer in **section 3** by defining the individual growth limitation ratios as $\beta_X$, which in sum give the relative growth ratio (see also answer to SC39):

"The non-dimensional relative growth ratio $\mu_{rel}^{ij}$ between two phytoplankton types i and j, e.g. diatoms and coccolithophores, can then be defined as the ratio of their specific growth rates (Hashioka et al., 2013):

$$\mu_{rel}^{DC} = \log \frac{\mu^D}{\mu^C}$$
$$= \underbrace{\log \frac{\mu_{max}^D}{\mu_{max}^C}}_{\beta_{\mu_{max}}} + \underbrace{\log \frac{f^D(T)}{f^C(T)}}_{\beta_T} + \underbrace{\log \frac{g^D(N)}{g^C(N)}}_{\beta_N} + \underbrace{\log \frac{h^D(I)}{h^C(I)}}_{\beta_I} \tag{4}$$

In this equation, the terms $\beta_{\mu_{max}}$, $\beta_T$, $\beta_N$, and $\beta_I$ describe the log-transformed differences in the maximum growth rate $\mu_{max}$, temperature limitation $f(T)$, nutrient limitation $g(N)$, and light limitation $h(I)$ between diatoms and coccolithophores, which in sum give the difference in the relative growth ratio $\mu^{DC}_{rel}$."

Furthermore, to avoid confusion, we no longer use the term "clearance rate" in the revised version of the manuscript, but refer to the term gamma_j/P_j in Eq. 6 & 7 of the original manuscript as the "biomass-normalized specific grazing rate" instead (see answer to SC18).

We have changed the respective part in **section 3** to:

"To assess differences in biomass accumulation rates between different PFTs, we compute biomass-normalized specific grazing rates $c^i$ **[d$^{-1}$]** of phytoplankton i as the ratio of the specific grazing rate and the respective phytoplankton's biomass $P^i$:

$$c^i = \frac{\gamma_g^i}{P^i} \qquad (6)$$

The higher this rate, the more difficult it is for a phytoplankton i to accumulate biomass.

Consequently, the non-dimensional relative grazing ratio $\gamma^{ij}$ of phytoplankton i and j, e.g. diatoms and coccolithophores, is defined as (Hashioka et al., 2013):

$$\gamma_{g,rel}^{DC} = \log \frac{c^C}{c^D} \qquad (7)$$

"

**General Comment #2:**
*(1) I have performed myself an extensive analysis using MATLAB/OCTAVE of the niche properties of these simulated phytoplankton taxa based on the model parameters provided. That is how I discovered these and other conceptual errors, which may affect the discussion of the manuscript. For example, the grazing equation eq(5) described in the text: G1 = gmax \* Z \* P1 / (K + P1) is incorrect to compute the grazing on P1. The correct equation should be G1 = gmax \* Z \* P1 / (K + P1 + P2) if we assume that there are two prey (P1 + P2) available for grazing (e.g. diatoms and coccos). This correction (adding P1 + P2 in the denominator of the grazing functional response) alters the results of the "relative clearance rates" (C:D) and the coccos are always grazed faster than diatoms. Therefore, given the parameter values provided by this model there is no surprise that diatoms will dominate coccos by a factor of x10 almost everywhere and at anytime.*

**Answer to GC2:**
**(1)** We sincerely thank the reviewer for investing a tremendous amount of time to perform the detailed niche analysis and providing code and plots. These analyses have triggered much additional discussion among the authors and developers of the current BEC version, which has substantially enhanced our understanding of model behavior and model limitations.

We confirm that Eq. 5 is correctly reported in the originally submitted manuscript. From its early stages, BEC has treated food sources independently in the grazing response, i.e. it computes the zooplankton grazing rate on each phytoplankton PFT separately, without constraining total grazing by total phytoplankton biomass (see e.g. **Moore et al., 2002; Moore et al., 2004; Moore et al., 2013**, see also **Sailley et al. 2013**; Table 2 and **Hashioka et al. 2013**, Appendix B3).
In contrast to early applications with BEC (**Moore et al., 2002; Moore et al., 2004**), sometime prior to 2013, the grazing equation in BEC was changed from the previously published Holling type III function (**Moore et al. 2002, 2004**) to a Holling Type II functional response for the ingestion term (Matthew Long, personal communication), which is, however, not documented in the published literature (e.g. **Moore et al. 2013**). The independent treatment of food sources in the grazing response has however not been affected by the move from a Holling Type III to a Holling Type II function.

We agree that the grazing function currently used in global or regional applications of BEC is likely to exert a large influence on the phytoplankton biogeography (e.g. **Prowe et al. 2012, Vallina et al. 2014**, etc.). The grazing response in models has been heavily debated in the literature (e.g. **Gentleman et al. 2003**). Since grazing formulations vary strongly between models (**Sailley et al. 2013**), and are prone to different advantages and limitations that this reviewer has pointed out multiple times in the literature (**Vallina and LeQuéré 2011; Vallina et al. 2014**), we chose to remain consistent with the current global and regional versions of BEC, and to interpret the functional response of our model behavior in terms of phytoplankton biogeography and the relative importance of top-down and bottom-up controls within this specific context.

Since the grazing formulation is of substantial importance for the behavior of any lower trophic level ecosystem model (**Prowe et al. 2012**, **Vallina et al. 2014**, **Le Quéré et al. 2016**, but also earlier papers by **Sailley et al. (2013)** and **Hashioka et al. (2013)** within the context of the MAREMIP project), and since we share the concern of the reviewer that the currently used functional response has

some disadvantages in terms of its biological realism, we included the grazing tests to assess the sensitivity of our results to the chosen grazing function as part of the sensitivity experiments in our original manuscript (HOLLINGIII as well as ACTIVE SWITCHING in Table 2). In the revised version of the manuscript, we have included a third grazing sensitivity experiment (HOLLINGII_SUM_P), in which we use a Holling type II ingestion function with the total phytoplankton biomass in the denominator, thereby constraining total grazing by the total food available.

We understand that we have not been sufficiently clear in the original version of the manuscript with regard to this important aspect, and have addressed the comment in the following way in the revised version of the paper:

In the **introduction**, we now state that the impact of different grazing formulations is ongoing research amongst ecosystem modelers, and that Earth System models use a range of grazing formulations:

"In the SO, previous studies have shown zooplankton grazing to control total phytoplankton biomass (Le Quéré et al., 2016), phytoplankton community composition (Scotia Weddell Sea, Granéli et al., 1993) and ecosystem structure (Smetacek et al., 2004; DeBaar, 2005), suggesting that top-down control might also be an important driver for the relative abundance of coccolithophores and diatoms. But the role of zooplankton grazing in current Earth System models is not well considered (Sailley et al., 2013; Hashioka et al., 2013), and the impact of different grazing formulations on phytoplankton biogeography and diversity is subject to ongoing research (e.g. Prowe et al., 2012; Vallina et al., 2014)."

In the **methods section 2.1**, we now highlight the switch from Holling Type III to Holling Type II in the currently used BEC version:

"In BEC, phytoplankton are grazed by a single zooplankton PFT, comprising characteristics of both micro- and macrozooplankton (Moore et al., 2002; Sailley et al., 2013). The single zooplankton PFT grazes on all phytoplankton PFTs using a Holling type II ingestion function (Holling, 1959). This is in contrast to earlier versions of BEC, which used a Holling type III ingestion function (see e.g. Moore et al., 2002). While not explicitly stated in the published literature, the formulation was already changed to a Holling type II ingestion function in previous, more recent applications of BEC (Moore et al., 2013, Matthew Long, pers. comm.)."

In addition, as pointed out above, we have added a third grazing sensitivity experiment to test the impact of constraining total grazing by the total phytoplankton biomass in the denominator of the Holling type II ingestion function. We have added this run to **Table 2** and have adjusted the **method section 2.3** accordingly:

| Grazing | Run Name | Description |
| --- | --- | --- |
| 12 | HOLLING_III | Instead of Eq. 5, use $\gamma_g^i = \gamma_{max}^i \cdot f^Z(T) \cdot Z \cdot \frac{P'^i \cdot P'^i}{z_{grz}^i \cdot z_{grz}^i + P'^i \cdot P'^i}$ |
| 13 | ACTIVE_SWITCHING | Instead of Eq. 5, use $\gamma_g^i = \gamma_{max}^i \cdot f^Z(T) \cdot Z \cdot \frac{P'^i}{\sum_{j=1}^{4} P'^j} \cdot \frac{P'^i}{z_{grz}^i + P'^i}$ |
| 14 | HOLLINGII_SUM_P | Instead of Eq. 5, use $\gamma_g^i = \gamma_{max}^i \cdot f^Z(T) \cdot Z \cdot \frac{P'^i}{z_{grz}^i + \sum_{j=1}^{4} P'^j}$ |

"Third, we assess the sensitivity of the results to the chosen grazing formulation by performing three additional simulations: We first replace the Holling type II ingestion term (Eq. 5) by a Holling type III term (run 12, Holling, 1959). Thereby, the grazing pressure is decreased on prey in low concentrations. We then assess the impact of constraining grazing on each phytoplankton PFT by total phytoplankton biomass in the original Holling type II formulation (Eq. 5). To do so, we first scale the grazing rate on phytoplankton i linearly with the PFT's relative contribution to total phytoplankton biomass (run 13), and ultimately constrain the grazing rate on phytoplankton i by total phytoplankton biomass in the Holling type II ingestion function (run 14). […]"

Furthermore, we now discuss the sensitivity of the model results to the choice of grazing function in the **discussion section 5.3**:

"[…] The tight coupling between phytoplankton and the single zooplankton in BEC suggests a possible overestimation of the importance of top-down control in controlling the relative importance of coccolithophores in the SO, as compared to models with more zooplankton complexity.

Besides missing complexity by only including a single zooplankton PFT, the simulated biogeography and controls of the diatom-coccolithophore competition are also sensitive to the chosen zooplankton ingestion function. In ROMS-BEC, we found the effect of both a Holling type III and constraining zooplankton grazing by the total phytoplankton biomass on our results to be similar (run 12-14 in Table 2): The use of a Holling type III (HOLLINGIII) or an active prey switching (ACTIVE_SWITCHING) grazing formulation, as well as a Holling type II formulation constrained by total phytoplankton biomass (HOLLINGII_SUM_P), instead of our standard Holling type II grazing formulation with fixed prey preferences leads to increased coexistence in the phytoplankton community. This is because either of these changes reduces the grazing pressure on the less abundant PFTs. As a result, coccolithophores and SP increase in relative biomass importance compared to diatoms in all three sensitivity simulations (Fig. S9). At the same time, coccolithophore biomass is pushed outside of the observed range for both sensitivity cases (Fig. S9), indicating a parameter retuning to be necessary for a true comparison of drivers of coccolithophore biogeography across simulations. Regardless, this highlights again the strong impact of top-down controls on phytoplankton biogeography in ROMS-BEC.

The key role of zooplankton grazing for determining SO phytoplankton biomass (Le Quéré et al., 2016; Painter et al., 2010; Garcia et al., 2008) and community composition (e.g. Smetacek et al., 2004; Granéli et al., 1993; De Baar, 2005) has been demonstrated before, but its possible role for SO coccolithophore biogeography has not yet been addressed. […]"

We highlight in the **caveats discussion section** (section 5.4) that the grazing formulation remains one of the largest uncertainties in BEC (and other global models), which rewards further research:

"Ecosystem models do not only vary in the number of zooplankton PFTs, but also in the chosen grazing formulation (Sailley et al., 2013), e.g. in their functional response regarding the ingestion of prey (e.g. Holling Type II vs. Holling Type III, Holling, 1959) or in the prey preferences of each predator (variable or fixed). It has been shown previously in global models that the choice of the grazing formulation impacts phytoplankton biogeography and diversity (e.g. Prowe et al., 2012; Vallina et al., 2014). For ROMS-BEC, the chosen grazing formulation quantitatively impacts our results, but does not qualitatively change the importance of top-down factors. This finding agrees with previous modeling studies, which despite using different ecosystem complexity and grazing formulations, came to the conclusion that top-down control is of vital importance for phytoplankton biogeography and diversity (Sailley et al., 2013; Vallina et al., 2014; Prowe et al., 2012). However, we acknowledge the simplicity of the current grazing formulation in BEC, and future research should assess the impact of increased zooplankton complexity on the simulated controls of SO phytoplankton biogeography."

*(2) The idealized analyses of niche properties that I performed I think should be added as supplementary Material in a revised form of this manuscript. A table with the model parameters is important for replication of this work but figures showing the actual shapes of the nutrient uptake curves and grazing functional response are important for the reader because it provides a way of fast visual inspection and understanding. For example, just by visual inspection of the grazing functional responses one can predict a-priori that coccos are ALWAYS going to be grazed faster than diatoms.*

**(2)** We agree with the reviewer that a visualization of the different ecological niches of the PFTs is helpful, and now include idealized analyses of the niche properties in the supplementary information to this paper (see **Fig. R1** below). We expanded on the characteristics of the chosen parametrizations and added references to Fig. S10 in **appendix B**, which now reads:

"The temperature function f(T) is an exponential function (see Fig. S10a), being <1 for temperatures below Tref=30°C, modified by the constant Q10 specific to every phytoplankton i (see Table 1) describing the growth rate increase for every temperature increase of 10°C: […] Generally, the smaller Q10, the weaker is the temperature limitation of the respective phytoplankton."

"The limitation by surrounding nutrients $L^i(N)$ is first calculated separately for each nutrient (nitrogen, phosphorus, iron for all phytoplankton, silicate for diatoms only) following a Michaelis-Menten function (see Table 1 for half-saturation constants $k^i_N$ for the respective nutrient and phytoplankton i). For iron (Fe) and silicate (SiO3), the limitation factor is calculated following (see Fig. S10c): [...]"

"The light limitation function $h^i(I)$ accounts for photoacclimation effects by including the chlorophyll-to-carbon ratio $\theta^i$, as well as the nutrient and temperature limitation of the respective phytoplankton i (see Fig. S10b): […] Generally, the higher the αPI, temperature and nutrient stress, and the chlorophyll-to-carbon ratio of the respective phytoplankton, the weaker is the light limitation."

"The grazing rate $\gamma g^i$ [mmol C m$^{-3}$ day$^{-1}$] of the generic zooplankton Z [mmol C m$^{-3}$] on the respective phytoplankton i [mmol C m$^{-3}$] is described by (see Fig. S10d) [...]"

However, we would like to emphasize that there are certain misconceptions of the reviewer resulting from conclusions drawn based on his analysis of the potential niche behavior in an idealized single resource set-up, rather than those niches realized in a non-steady-state multi-resource ocean in our model. In our model, the niche characteristics are not influenced by the growth of each PFT on each single resource, but rather by the limitation of growth by the nutrient with minimal availability of all nutrients. Furthermore, additional limitation terms based on temperature and light availability influence growth behavior, which the reviewer has not taken into account. Hence, our realized ecological niches differ from those estimated by the reviewer (see **Figures R1 & R2** below). We discuss the realized niches in our response to General Comment 3 below.

[Figure]

**Figure R1:** Functional responses used in ROMS-BEC:
a) Temperature limitation (Eq. B5 + temperature correction for coccolithophores and diazotrophs, see appendix B of manuscript)

b) Light limitation (Eq. B9 in manuscript, using domain & annual mean surface chlorophyll-to-carbon ratio of each PFT and max. growth rate (dashed) or N-Temp-limited growth rate (0.1*max. growth rate, solid)), note that SP is not shown to enhance visibility as SP light limitation is very similar to that of diatoms (red)

c) Nutrient limitation (Eq. B6, example for iron shown here)

d) Grazing rate on phytoplankton (note that the rate shown here will be further scaled with zooplankton biomass and the zooplankton temperature limitation, see Eq. B12).

[Figure]

**Figure R2:** Realized niches of diatoms (a-b) and coccolithophores (c-d) south of 40°S in ROMS-BEC, example for iron: a) & c): Annual mean surface nutrient limited growth rate (g(N) *$\mu_{max}$, see Eq. 3 in the manuscript) vs. surrounding iron concentrations. b) & d): Annual mean surface realized specific growth rate (f(T) * g(N) * h(I) *$\mu_{max}$, see Eq. 3 in the manuscript) vs. surrounding iron concentrations. Panel c) resembles the shape of the nutrient limitation curve in Fig. R1c, as coccolithophore growth is limited by iron almost everywhere south of 40°S (see Fig. 2 in manuscript). However, panel a) deviates from theoretical curve because diatoms are not limited by iron everywhere south of 40°S, resulting in a nutrient-limited growth that does not follow the Michaelis-Menten curve shown in Fig.

R1c. The realized growth rates in panel b) & d) show the combined effect of all factors controlling phytoplankton growth, and the distribution shows significant discrepancies from the theoretical niche expected from single-resource limitation only.

**General Comment #3:**

*(1) Also the selection of uptake curves for nutrient uptake is quite puzzling to me. For DIN, NH4 and DIP, they are kind of similar where coccos are always losers to either diatom or small phyto; then for DOP the coccos appear to have small nutrient window where they can dominate. Then for iron coccos and small phyto are very similar in nutrient uptake strategy while diatoms dominates.*

**Answer to GC3:**

**(1)** We would like to thank Dr. Vallina for this comment and the suggestions regarding the nutrient uptake parameters and the attached code and plots. As known to the reviewer, parameter choice in marine ecosystem models is subject to uncertainties due to the large range of species-specific responses, and the fact that data is only acquired under laboratory conditions, and for the limited number of species currently kept in culture (**Anderson 2005**). While the actual parameters chosen can be debated, and are, we would like to point out that conclusions drawn based on the ecological niche behavior of the represented PFTs based on their responses to a variation in single nutrients are misleading, since the total nutrient limitation of each PFT is calculated as a minimum function of the most limiting nutrient in BEC and other models (see Eq. B6-B8 of original manuscript and e.g. **Hashioka et al. 2013** for a comparison of the phytoplankton parameterization of multiple global marine ecosystem models).

As documented in Table 1 of the original manuscript, the half-saturation constants for nutrient uptake by coccolithophores lie between those of diatoms (higher values) and small phytoplankton (smaller values) for all nutrients. i.e. nutrient concentrations *always* favor coccolithophores relative to diatoms (and small phytoplankton relative to coccolithophores, see blue areas in Fig. 5 & S6 of the manuscript). These values have been chosen to reflect differences in the size of the different PFTs, with diatoms being largest, coccolithophores the size of nanophytoplankton (i.e. *Emiliania huxleyi*), and small phytoplankton have been parameterized to reflect nano- and picophytoplankton.

We agree with the reviewer that when calculating the nutrient-limited growth rate separately for each nutrient and each phytoplankton, coccolithophores are only favored for a small window of DOP concentrations. However, this is not how the model simulates nutrient limited growth. Assuming phytoplankton growth is proportional to the most-limiting resource, modelled growth is thus scaled by the limitation function of the *most-limiting* nutrient only, which is used to compute the nutrient-limited growth rate (see equations B6-B8 in appendix of the original manuscript). The nutrient limitation term is first computed separately for each nutrient and each phytoplankton, and the limitation factor of the most limiting resource (these can be different nutrients for different phytoplankton types at any point in space and time, see Fig. 2 in the original manuscript) is then multiplied with $\mu_{max}$ to give the nutrient limited growth rate. This means that the window for which nutrient-limited growth of coccolithophores is larger than that of diatoms is wider than what the reviewer suggests (e.g. in instances where diatom growth becomes severely Si-limited, as coccolithophore growth is unaffected by Si availability, see **Fig. R3** below for the annual mean difference in nutrient limited growth rates of diatoms and coccolithophores).

Nutrient limited phytoplankton growth is further reduced by temperature and light limitation, to give the final specific growth rate (see Eq. 3 of the manuscript). Consequently, nutrient concentrations are not the only factor controlling coccolithophore/phytoplankton growth rates (see **Fig. R2** above). Our goal is not to create, as stated by the reviewer, "a larger nutrient concentration growth window for coccos where they can dominate" by setting the half-saturation constants in a specific way. Instead, we

chose all coccolithophore parameters in accordance with published laboratory data (see e.g. **Daniels et al., 2014, Heinle 2013, Buitenhuis et al., 2008, Zondervan 2007, Nielsen 1997, Le Quéré et al. 2016** and references therein), with minor tuning within the observational constraints to give the best fit with available biomass observations (**O'Brien et al., 2013, Balch et al., 2016; Saavedra-Pellitero et al., 2014; Tyrrell and Charalampopoulou, 2009; Gravalosa et al., 2008; Cubillos et al., 2007**).

[Figure]

**Figure R3:** Difference in annual mean surface nutrient limited growth rate ($g(N) * \mu_{max}$, see also Eq. 3 in the manuscript) between a) diatoms and coccolithophores and b) small phytoplankton and coccolithophores. Note that coccolithophores have a higher nutrient limited growth rate than diatoms for large areas north of approximately 45°S, corresponding to the onset of Si limitation of diatoms north of 45°S. Compare also to nutrient limitation patterns in Fig. 2 of the manuscript.

*(2) I would suggest the authors to simplify the model parameters: define a single set of four values (one per phyto group) for the half saturation constant on Dissolved Organic Phosphorous (DOP) and then compute the half-sat for all other nutrients (DIN, NH4, DIP, Fe2, Silica) using constant redfield ratios. I did this myself and I feel that uptake curves obtained are nicer and more consistent across nutrient types, and they also provide a slightly larger nutrient concentration niche window for coccos where they can dominate. There is "key switch" in my MATLAB/OCTAVE code (key_params = 'Original' or 'Redfield') to change from the original model parameters setup to the suggest parameter setup. The results of the simulation should not change qualitatively after this minor changes on parameter values, and if they do then there is an issue of model sensitivity to parameter values. Please see the resulting Figures in the attached PDF documents. I provide the MATLAB/OCTAVE code I wrote at the end of this review report. The authors are free to use it.*

**(2)** We thank the reviewer for the suggestion to simplify model parameters. However, as stated above, we chose all coccolithophore parameters in accordance with published laboratory data (see e.g. **Daniels et al., 2014, Heinle 2013, Buitenhuis et al., 2008, Zondervan 2007, Nielsen 1997, Le Quéré et al. 2016** and references therein), with minor tuning within the observational constraints to give the best fit with available biomass observations (**O'Brien et al., 2013, Balch et al., 2016; Saavedra-Pellitero et al., 2014; Tyrrell and Charalampopoulou, 2009; Gravalosa et al., 2008; Cubillos et al., 2007**).

We have performed a test to assess the impact of using Redfield-stoichiometry for the half-saturation constants of $NO_3$ and $PO_4$ on phytoplankton biogeography in our model: For the test, we kept the original $kPO_4$ of all phytoplankton groups (see Table 1 in original manuscript for parameters used in reference simulation) and scaled the $kNO_3$ of each phytoplankton group following the Redfield ratio for all except diazotrophs, which have been shown to have significantly higher N:P ratios (around 45,

e.g. **Letelier & Karl, 1998**, **White et al., 2006**). Furthermore, we scaled all kNH4 to be 10% of kNO3 and all kDOP to be a factor 20 higher than kPO4 for all PFTs. We left all kSi and kFe unchanged as compared to our reference run. The resulting simulated phytoplankton biogeography is very close to the one in our reference simulation (see **Fig. R4**, <1% change in %contribution of each PFT to total NPP between 30-90°S). This result is not surprising, as phytoplankton growth south of 30°S is largely limited by Fe and/or Si availability in ROMS-BEC (see Fig. 2 in original manuscript). Changes in phytoplankton biomass only occur in the small regions of N-/P-limitation towards the northern end of the domain, but are minor overall. In conclusion, whether phytoplankton half-saturation constants with respect to kNO3 and kPO4 follow Redfield ratios or not does not affect the outcome of our study.

In the revised version of the manuscript, we specified the respective sentence in **section 2.1** to highlight that nutrient uptake by diazotrophs follows non-Redfield stoichiometry:

"Phytoplankton C/N/P stoichiometry in photosynthesis is fixed close to Redfield ratios (117:16:1 for diatoms, coccolithophores, and SP, 117:45:1 for diazotrophs, Anderson and Sarmiento, 1994; Letelier and Karl, 1998), but the ratios of Fe/C, Si/C and Chl/C vary according to surrounding nutrient levels."

[Figure]

**Figure R4:** Difference in DJFM mean a) total surface chlorophyll, b) top 50 m mean coccolithophore biomass, and c) top 50 m mean diatom biomass between a test with half-saturation constants scaled according to Redfield ratios (see answer to GC 3) and the reference simulation (for parameters see Table 1 in original manuscript).

*(3) Finally diazotrophs are clear losers for all nutrients against all other phytoplankton groups -- no surprise they are almost zero biomass (less than 1%) in the model.*

**(3)** Diazotroph biomass/$N_2$ fixation is thought to be of very minor importance in the Southern Ocean (see e.g. **Luo et al., 2012**), as *Trichodesmium spp.* thrives in warm (sub)tropical waters (see discussion of this PFT in BEC in **Moore et al., 2002**). We note that we left parameters untouched as compared to previous global applications of BEC, and that diazotrophs are parameterized not to grow in waters with temperatures below 14°C. Hence, diazotroph growth in summer is inhibited by surrounding temperatures in >63% of our domain (south of 42°S), where DJFM mean SST is <14°C.

**General Comment #4:**
*Regarding the Sensitivity Analysis (SA) performed by the authors, I am afraid to say that is not useful in the way it has been designed. Basically because 1) changing the selected parameters "from the values of coccos to the values of diamtoms" provides too little change of the parameter values; 2) the percentage of change for the selected parameters is different among them. That means that 1) the response of the model for the "sensitivity runs" will usually be quite small for most of the parameters; and 2) the responses of the model for the "sensitivity runs" will not be comparable accross parameters. For example, changing the Q10_coccos (1.45 n.d.) to Q10_diatoms (1.55 n.d.) is a meager 7% variation in the Q10 value of coccos; or changing the alpha_coccos (0.4) to alpha_diatoms (0.44) means a 10% variation in the alpha PI value of coccos. Then, changing mupmax_coccos (3.8 d-1) to mupmax_diatoms (4.6 d-1) implies a 20% variation in the mupmax of coccos; and changing the half sat kno3_coccos (0.3) to kno3_diatoms (0.5) implies a 66% variation. On the other hand, changing the half sat for grazing zgrz_coccos (1.05) to zgrz_diatoms (1.0) implies a 5% variation. Therefore, some parameters are changed less than 10% while others are changed more than 60%, and others are changed around 20%. This is too messy for a meaningful SA. The correct way to perform a proper (yet simple) SA is by changing a plus/minus 50% the selected parameter values, and then computing the Sensitivity Index (SI) as follows – and the plotted as shown in Figure 12 of Vallina et al. (2008) DMOS model, JGR:*

*SI = 100 * (X_pmax - X_pmin) / X_pcontrol ;*

*where:*
*# X -- is the model state variable of interest (e.g. Diatom concentration)*
*# pcontrol -- is the selected parameter value for the control run (e.g. mupmax of coccos)*
*# pmax = 1.5 x pcontrol -- is the run for the plus(+) 50% increase in the parameter value*
*# pmax = 1.5 x pcontrol -- is the run for the minus(-) 50% decrease in the parameter value*
*# SI -- is the Sensitivity Index in percentage of change (can be positive or negative %)*

**Answer to GC4:**
We thank the reviewer for this comment, and recognize that the purpose of our sensitivity experiments, as well as the wording in the section title of section 4.7 "Parameter sensitivity simulations" may have been confusing. We need to distinguish between classical sensitivity analysis usually performed for one single model, and sensitivity experiments that allow for the testing of multiple model structures and/or the identification of the importance of several processes known to affect a target variable.

In a "classical" sensitivity analysis, modelers assess the sensitivity of one single model to a variation in its parameter values using a defined degree of variation, such as e.g. the 50% changes in all parameters suggested by the reviewer. This is usually used to rank the importance of the model parameters for one specific target or outcome variable (e.g. NPP), which is a very useful exercise when the goal is e.g. to compare internal sensitivities of one or different models to individual processes across models (like e.g. within MAREMIP).

From our point of view, a "classical" parameter sensitivity study as suggested by the reviewer does not add additional understanding to the manuscript with regard to the simulated coccolithophore biogeography for two reasons: *First*, changing e.g. the half-saturation constant of $NO_3$ by $\pm50\%$ will lead to non-linear changes (nutrient limitation is a non-linear function of the half-saturation constant, see Eq. B6/B7), making these runs not directly comparable to the control in a strict sense – especially when comparing to runs assessing e.g. the sensitivity to changes in the Q10 value, which enters the growth function in an exponential way, thus artificially amplifying the sensitivity of phytoplankton biomass to modifications in this parameter (see Eq. B5). *Second* and more importantly, while a 50% variation might be justifiable for the half-saturation constants to remain within their experimental constraints, other parameters (such as the Q10 value), do not show such a large variability within one single PFT in nature (see e.g. review by **Le Quéré et al., 2016**), thus making it difficult to extract any meaningful information on coccolithophore biogeography when varying parameters way beyond their observed variability.

Last but not least, we use the discussed set of sensitivity experiments using multiple similar models is used to assess the dependence of model responses to differences in model structure, model formulations, and parameter sets, i.e. to compare multiple models with different characteristics. This can be used, e.g. to identify those processes most important for the representation of a target tracer, or other outcome variables, or the impact of specific biases on the model results. Together with the reviewer, we have employed both strategies, but for different purposes in multiple publications (e.g. **Le Quéré et al. 2016**; **Vogt et al. 2010**; etc.). Hence, in the current manuscript we do not perform a sensitivity analysis *senso stricto* to assess the model sensitivity to the set of chosen parameters, since these dependences have been explored in previous publications of BEC, and are fairly consistent across known marine ecosystem models. For instance, the parameters used to describe the top predator are usually ranked high in terms of the results in biomass and biogeochemical target variables such as NPP, C and N cycling, or certain gas exchange processes. Here, we intend to test the dependence of the model behavior not only on parameter choices, but also on selected model equations in our grazing tests (Table 2 in the original manuscript), as well as using sensitivity experiments to assess which coccolithophore parameter is the one most crucial for phytoplankton biogeography and phenology, but also for diatom-coccolithophore competition. We understand the confusion of the reviewer, and acknowledge that the purpose of our sensitivity experiments was unclear.

In our diatom-coccolithophore competition experiments (see Table 2 in original and revised manuscript), we specifically chose to change each parameter by a different percentage, namely setting the coccolithophore parameters to those of diatoms. Our goal with these experiments was to directly assess the impact of the difference in parameters between coccolithophores and diatoms on the relative abundance of these two PFTs in our model domain. We asked ourselves: What would coccolithophore biogeography look like if e.g. their maximum growth rate was that of diatoms? By how much does their biomass (integrated over e.g. 40-50°S) change when coccolithophores have the same maximum growth rate as diatoms? Here, we are not evaluating the sensitivity of the baseline setup in this classical sense to get parameter sensitivities of BEC, but rather compare different realizations of the baseline with respect to the parametrization of coccolithophores to quantify the impact on the relative importance of coccolithophores and diatoms in the SO. Our setup of the simulations is therefore rather comparable to sensitivity simulations in the literature in which ecosystem structure or phytoplankton biogeography was altered by "making two PFTs equal/different" in a single or in multiple parameter values (**Wang & Moore 2011**) or by removing/adding more zooplankton complexity (**Le Quéré et al., 2016**).

We agree with the reviewer that calling the simulations 1-9 in Table 2 of the manuscript "parameter sensitivity simulations" (section 4.7) might be unsuitable in this context as the reader might misinterpret the title. We therefore change the name of **section 4.7** to "Sensitivity of coccolithophore biogeography to chosen parameter values" to more adequately represent the purpose of these runs. Additionally, we expanded the description of sensitivity simulations 1-9 in **section 2.3** by the following sentence to make its purpose clearer:

"Thereby, we can directly assess the impact of differences between coccolithophores and diatoms in each of the model parameters on the relative biomass of coccolithophores. For all simulations, we quantify the sensitivity as a change of each PFT's annual mean surface biomass, focusing particularly on coccolithophores in section 4.7."

Furthermore, to make the purpose of the different simulations clearer, we split the list of all sensitivity simulations in **Table 2** of the manuscript into three subgroups "Competition" (run 1-9), "Biases" (run 10-11), and "Grazing" (run 12-14):

**Table 2.** Overview of sensitivity simulations. 1-9: Sensitivity of simulated coccolithophore-diatom competition to chosen parameter values of coccolithophores. See Table 1 for parameter values of coccolithophores in reference run. 10-11: Sensitivity of simulated biogeography to biases in temperature and mixed layer depth. 12-14: Sensitivity of simulated biogeography to the chosen grazing formulation. C=coccolithophores, D=diatoms.

| Competition | Run Name | Description |
|---|---|---|
| 1 | GROWTH | Set $\mu_{max}^C$ to $\mu_{max}^D$ |
| 2 | ALPHA$_{PI}$ | Set $\alpha_{PI}^C$ to $\alpha_{PI}^D$ |
| 3 | Q10 | Set $Q_{10}^C$ to $Q_{10}^D$ |
| 4 | GRAZING | Set $\gamma_{max}^C$ and $z_{grz}^C$ to $\gamma_{max}^D$ and $z_{grz}^D$ |
| 5 | IRON | Set $k_{Fe}^C$ to $k_{Fe}^D$ |
| 6 | SILICATE | Limit coccolithophore growth by silicic acid by using $k_{SiO3}^D$ |
| 7 | NITRATE | Set $k_{NO3}^C$ and $k_{NH4}^C$ to $k_{NO3}^D$ and $k_{NH4}^D$ |
| 8 | PHOSPHATE | Set $k_{PO4}^C$ and $k_{DOP}^C$ to $k_{PO4}^D$ and $k_{DOP}^D$ |
| 9 | NUTRIENTS | Set all $k_{Nutrient}^C$ to $k_{Nutrient}^D$ |

| Biases | Run Name | Description |
|---|---|---|
| 10 | TEMP | Reduce temperature in BEC subroutine by $1^{\circ}$C everywhere |
| 11 | MLD | Reduce incoming PAR in BEC subroutine by -20% everywhere |

| Grazing | Run Name | Description |
|---|---|---|
| 12 | HOLLING_III | Instead of Eq. 5, use $\gamma_g^i = \gamma_{max}^i \cdot f^Z(T) \cdot Z \cdot \frac{P'^i \cdot P'^i}{z_{grz}^i \cdot z_{grz}^i + P'^i \cdot P'^i}$ |
| 13 | ACTIVE_SWITCHING | Instead of Eq. 5, use $\gamma_g^i = \gamma_{max}^i \cdot f^Z(T) \cdot Z \cdot \frac{P'^i}{\sum_{j=1}^4 P'^j} \cdot \frac{P'^i}{z_{grz}^i + P'^i}$ |
| 14 | HOLLINGII_SUM_P | Instead of Eq. 5, use $\gamma_g^i = \gamma_{max}^i \cdot f^Z(T) \cdot Z \cdot \frac{P'^i}{z_{grz}^i + \sum_{j=1}^4 P'^j}$ |

**Answers to specific comments (SC):**

**SC1**: *Page 1, Line 01: Change "controls not only the local biogeo but also" to "controls the local biogeo and" -- Avoid unnecessary negatives in affirmative sentences.*
Agreed, we have changed the text accordingly.

**SC2**: *Page 3, Line 03: "interaction" is very generic term -- Would be better to specify "competitive interaction" (for light and nutrients) or just "competition".*
Thanks for making this comment, we have changed the text accordingly.

**SC3**: *Page 4, Table 1: The parameter values for mupmax at 30C are quite large (3.6, 4.6 d-1) and in my view quite beyond the reality of these values in nature.*

We thank the reviewer for this comment. We acknowledge that the chosen $\mu_{max}$ values of 3.8 d$^{-1}$ at 30°C for coccolithophores, and of 4.6 d$^{-1}$ at 30°C for diatoms seem high when looking at available growth rate data from the lab (see Fig. 2 in **Le Quéré et al. (2016)**, and **Fig. R5** below). However, for subantarctic latitudes, choosing a Q10-formulation with the reported $\mu_{max}$ results in growth rates for diatoms and coccolithophores within the range suggested by lab studies for the temperature range simulated in these latitudes of the SO (see **Fig. R5).** We acknowledge that our overestimation of coccolithophore biomass towards the northern boundary of our domain, as well as the overestimation of diatom biomass at high southern latitudes are likely partly caused by the chosen temperature limitation function and the relatively high $\mu_{max}$. Therefore, we note that this should be reconsidered when implementing coccolithophores into global models (as done in **Le Quéré et al., 2016**, see also **Krumhardt et al., 2017**).

[Figure]

[Figure]

**Figure R5:** Temperature limited growth rates of coccolithophores and diatoms. Blue dots show phytoplankton growth rates as reported in the literature (see references in **Le Quéré et al. (2016)**), black line represents the temperature limited maximum growth rate as simulated in ROMS-BEC (Eq. B5 and B10 of original manuscript).

**SC4**: *Page 4, Table 1: The parameter values for Q10 are almost the same for all phyto -- why dont you just use 1.5 for all of them?*
Thanks for pointing this out. While the differences in Q10 between the phytoplankton do appear quite small at first (1.45 for coccolithophores, 1.55 for diatoms, 1.5 for others), the temperature limitation function of coccolithophores and diatoms differs by roughly 10% in our focus area between 40-60°S (see Fig. 6 in the manuscript), thereby contributing to differences in the specific growth rates of the two phytoplankton groups. The literature review reported by **Le Quéré et al. (2016)** (see references therein) suggests that the Q10 values of coccolithophores and diatoms differ, and we acknowledge that we were rather conservative by not using a difference as large as suggested by Le Quéré et al. (2016, 1.14 +/- 0.17 for coccolithophores vs 1.45 in ROMS-BEC, 1.97 +/-0.07 for diatoms vs 1.55 in ROMS-BEC). We therefore rather underestimate the importance of differences in temperature limitation for the relative importance of diatoms and coccolithophores in the Southern Ocean.

Along these lines, we have added a discussion of the importance of temperature in controlling SO coccolithophore biogeography in **section 5.3**:

"Temperature has been suggested to be a major driver of latitudinal gradients in SO coccolithophore abundance (e.g. Saavedra-Pellitero et al., 2014; Hinz et al., 2012). In our study, differences in temperature sensitivity between diatoms and coccolithophores play a minor role in controlling the relative importance of these two phytoplankton groups (see Fig. 5 & 6). However, globally, the Difference in temperature sensitivity (Q10) of diatom and coccolithophore growth appears to be larger (1.93 and 1.14, respectively, see Le Quéré et al., 2016) than what is currently used in ROMS-BEC (1.55 and 1.45, respectively, see Table 1), indicating that we likely underestimate the importance of temperature in controlling the relative importance of diatoms and coccolithophores in our model."

**SC5**: *Page 4, Table 1: The parameter values for knh4 are a factor of x 10 smaller (i.e. 0.10) than kno3 for all C, D and SP; but 0.15 for diazotrophs -- why dont you just use time 0.10 for all of them?*
Please refer to our answer to GC3. Since diazotrophs are known to be unimportant in terms of their contribution to total phytoplankton biomass in the Southern Ocean, the diazotroph parameters have not been tuned in this study. Changing the value from 0.15 to 0.10 would not change the outcome of our study significantly, as diazotrophs are such a minor member of the SO phytoplankton community in our simulation and therefore not the focus of this study (see also answer to SC29).

**SC6**: *Page 4, Table 1: The parameter values for kpo4 are the same value than for kno3 for C, D, and SP (even if the Redfiled ratio N:P is 16:1 -- why?) but for diazos is ten times smaller -- why?*
Thank you for this comment, but we think there is a misunderstanding. As reported in Table 1 in the manuscript, the half-saturation constants for $PO_4$ are an order of magnitude smaller than those for $NO_3$ for all phytoplankton PFTs, not just for diazotrophs as suggested by the reviewer.

**SC7**: *Page 4, Table 1: The parameter values for alpha PI curve are between 0.38 and 0.44 for all phyto groups (very narrow variability) -- why dont you just use 0.4 for all of them?*
We thank the reviewer for this comment. As pointed out by **Le Quéré et al. (2016)**, literature values do not allow for the identification of a clear difference in alphaPI values for different PFTs, which is why a lot of the published ecosystem models do use the same value for alphaPI for all their PFTs (see e.g. **Le Quéré et al. (2016)** for PlankTOM10 and **Aumont et al. (2015)** for PISCES). However, alphaPI is an important parameter for the onset of phytoplankton blooms, as it controls the sensitivity of phytoplankton to changes in light intensity at low irradiance levels. Coccolithophores (including those from the Southern Ocean) are known to thrive especially in high-light environments (in summer, see e.g. **Balch et al. (2004)**) suggesting that diatoms are generally better in coping with lower light conditions than coccolithophores. Therefore, the motivation for setting alphaPI of coccolithophores slightly lower than the value of diatoms was to slightly delay their blooms as compared to those of diatoms. As pointed out in section 4.5 (see Fig. S5 in supplementary material), the variability in light limitation across phytoplankton types in ROMS-BEC is much more controlled by differences in photoacclimation than in alphaPI alone. The variability in alphaPI is indeed small, but as **Fig. R1** shows (see our answer to GC2), variability in light limitation is rather large across phytoplankton types when accounting for differences in chlorophyll-to-carbon ratios and nutrient/temperature stress of all phytoplankton types (**Geider et al., 1998**). In our sensitivity simulation ALPHAPI (in which we set alphaPI of coccolithophores to the value of diatoms, see Table 2 of the manuscript), annual mean surface coccolithophore biomass increases by only 10% between 40-50°S (and less everywhere else) as compared to the baseline simulation (see Fig. 7 in the manuscript), suggesting that the chosen difference in alphaPI in ROMS-BEC does not significantly impact the outcome of this study in terms of simulated phytoplankton biogeography.

**SC8**: *Page 4, Table 1: The parameter values for zgrz half sat grazing are between 1.0 and 1.2 for all phyto groups (very narrow variability) -- why dont you just use 1.0 for all of them?*
We agree with the reviewer that this variability is indeed small and affects phytoplankton biomass only at high biomass concentration (**Fig. R6** below). We admit that we could have set this parameter equal for all PFTs without changing the resulting phytoplankton biogeography much (**see Fig. R6** below), especially when only considering diatoms ($z_{grz}$=1.0), coccolithophores (1.05), and small phytoplankton (1.05). When implementing the new PFT, we set coccolithophore parameters to SP parameters, since we had no other information available to base it on. We point out that for the application of BEC in this manuscript, i.e. the competition between diatoms and coccolithophores in the SO, differences in $v_{max}$ between diatoms and coccolithophores impact grazing rates more than differences in $z_{grz}$ (see **Fig. R6**: compare black to dashed blue (impact of difference in zgrz) vs black to solid blue (impact of difference in $v_{max}$)).

[Figure]

**Figure R6:** Grazing rates on phytoplankton in BEC using different values for maximum growth rate of zooplankton when grazing on phytoplankton ($\gamma_{max}$), as well as the half-saturation constant for ingestion ($z_{grz}$).

**SC9**: *Page 4, Table 1: Basically, I find that the selection of parameter values can be simplified a lot given the similarity among many of the values. I would suggest selecting one nutrient (e.g. DOP) and define the half-sat values for it, then compute the half-sat for all other nutrients (DIN, NH4, DIP) using Redfield ratios and constant factors. This will make the list of parameter values much smaller without compromising the simulations. Also this will simplify the Sensitivity Analysis because it reduces the degrees of freedom in the parameter space.*
Please see our answer to GC3.

**SC10**: *Page 4, Line 5: "coccos have a higher nutrient affinity (smaller half-sat) and a smaller max growth rate than diatoms" -- the half sat parameter is in fact a composite of two primary parameters (ksat = mupmax / affinity) therefor lower half sat does not necessarily mean higher affinity, it also depends on the mupmax value. Also, the smaller size of coccos cannot be the reason of their smaller mupmax because mupmax usually decreases with increasing cell size.*
We thank the reviewer for this comment. We have rephrased the sentence to "coccolithophores are less nutrient limited at low nutrient concentrations (smaller half-saturation constants) [...]".

**SC11**: *Page 6, Table 2: Run name 10 and 11; Holling Type III and Active Switching – I dont understand... Very confusing. Please double check and make it clearer.*
Thanks for pointing out that the manuscript appears to be written not clearly enough here. Our baseline simulation uses a Holling Type II functional response for ingestion, not accounting for the total phytoplankton biomass in the denominator (see Eq. 5 in the manuscript and our answer to GC 2). The two sensitivity simulations assessing the sensitivity of coccolithophore biogeography and coccolithophore-diatom competition to the choice of the grazing parametrization are the two simulations the reviewer refers to here: for the Holling Type III simulation, we replace the Holling Type II part of Eq. 5 by a Holling Type III functional response, for the ACTIVE_SWITCHING simulation in the original manuscript, we add the term $P^i/sum(P^i)$ to Eq. 5 (keeping a Holling Type II function for the ingestion term), thereby distributing the total grazing linearly across the phytoplankton PFTs in accordance with each PFT's contribution to total phytoplankton biomass. We have modified the description of this run, which now reads:

"We then assess the impact of constraining grazing on each phytoplankton PFT by total phytoplankton biomass in the original Holling type II formulation (Eq. 6). To do so, we […] scale the grazing rate on phytoplankton i linearly with the PFT's relative contribution to total phytoplankton biomass (run 13), […]."

To avoid confusion, we made the description of all grazing sensitivity simulations clearer in **Table 2** by explicitly stating the equation used for zooplankton grazing instead of Eq. 5:

| Grazing | Run Name | Description |
|---|---|---|
| 12 | HOLLING_III | Instead of Eq. 6, use $\gamma_g^i = \gamma_{max}^i \cdot f^Z(T) \cdot Z \cdot \frac{P'^i \cdot P'^i}{z_{grz}^i \cdot z_{grz}^i + P'^i \cdot P'^i}$ |
| 13 | ACTIVE_SWITCHING | Instead of Eq. 6, use $\gamma_g^i = \gamma_{max}^i \cdot f^Z(T) \cdot Z \cdot \frac{P'^i}{\sum_{j=1}^4 P'^j} \cdot \frac{P'^i}{z_{grz}^i + P'^i}$ |
| 14 | HOLLINGII_SUM_P | Instead of Eq. 6, use $\gamma_g^i = \gamma_{max}^i \cdot f^Z(T) \cdot Z \cdot \frac{P'^i}{z_{grz}^i + \sum_{j=1}^4 P'^j}$ |

**SC12**: - *Page 6: Table 2: Run name 12; TEMP; Reduce temperature by 1C -- This seems a pretty low decrease in temperature for a SA. What percentage is this?*
Thanks for this comment. This temperature change is motivated by the mean bias in the domain which is largest with ~1°C between 60-90°S (see Fig. S1). We have added this information in the main text in **section 2.3**., the respective sentence now reads:

"To do this, we reduce temperatures by $1°C$ (corresponding to the mean bias between 60-90°S, see Fig. S1, run 12) and the incoming PAR field by 20% (to counteract bias in MLD, run 13) everywhere for the biological subroutine only. "

**SC13**: *Page 6, Line 04: "We then add an active prey switching term to the original Holling Type II" -- The Holling Type III has already "active prey switching" dynamics so I dont really know what this addition to Holling Type II means.*
We have made the description of the different types of sensitivity experiments clearer in **section 2.3** of the manuscript, please see our answer to SC11 and GC2.

**SC14**: *Page 7, Line 15: "Ideal env conditions ... at every location" -- The nutrient concentration window where coccos reach a higher specific growth rate than diatoms is very narrow and happens at very low nutrient concentrations; for all other conditions the diatom are superior competitors regardless of light and temperature levels. Therefore what dominates the competition between coccos and diatoms is basically nutrients only (in your model setup).*
We agree with the reviewer and find in our simulation that the advantage in specific growth rate of coccolithophores in summer is indeed to a large part driven by advantages in nutrient uptake (especially between 40-50°S, see Fig. 6A, as well as **Fig. R3** in this document), but advantages in temperature sensitivity and the disadvantage due to the predefined smaller $\mu_{max}$ also contribute to the overall difference in specific growth rates, with light being relevant mainly in other seasons (see Fig. 5 & 6). We have changed the respective sentence in the manuscript as follows:

"Since the coccolithophores' maximum growth rate is lower than that of diatoms (Table 1), ideal environmental conditions, i.e., low nutrient concentrations and temperature, as well as high light levels, are required for coccolithophores to overcome this disadvantage and to develop a higher specific growth rate than diatoms."

**SC15**: *Page 7, Line 19: The equation for grazing rate on Pj is wrong: it should be -- gamma_j = gammax_j * f(T) * Z * (P_j / (zgrz_j + sum[P_i])) NOTE: The sum[P_i]*
Please see our answer to GC 2.

**SC16**: *Page 7, Line 23: Why do you use the notation "P_j prime" (P') instead of just P (without the prime)?*
In the manuscript, the difference between P'j and Pj denotes the consideration of a fixed loss threshold, below which no losses occur (prey refuge, see SC17). In ROMS-BEC, the prey refuge is accounted for to calculate the grazing on Pj .

**SC17**: *Page 7, Line 20: "loss threshold below which no losses occur (eq B11)" -- This is called imposed "prey refuge" and helps prevent competitive exclusion of the less dominant prey types (in a similar way as active prey switching).*
Thanks for pointing this out, we have added the term "prey refuge" to the manuscript.

**SC18**: *Page 8, Line 03: The ratio (gamma_j / P_j) will only have units of clearance rate (m3 * mmol-1 * d-1) if divided by zooplankton biomass. Otherwise it will have units of specific grazing rate (d-1) -- please double check units and concepts.*
We thank the reviewer for pointing this out. We will not use the term "clearance rate" any more in the revised version of the manuscript, but will refer to the term gamma_j/P_j as the "biomass-normalized specific grazing rate" instead. This way, Fig 5 g&h in the manuscript are exactly analogous to Fig. 5 e&f , showing the biomass-normalized specific growth (e&f) and grazing rates (g&h) respectively.

We have changed the respective part in **section 3** which now reads:

"To assess differences in biomass accumulation rates between different PFTs, we compute biomass-normalized specific grazing rates $c^i$ $[d^{-1}]$ of phytoplankton i as the ratio of the specific grazing rate and the respective phytoplankton's biomass $P^i$:

$$c^i = \frac{\gamma_g^i}{P^i}$$
(6)

The higher this rate, the more difficult it is for a phytoplankton i to accumulate biomass. […]

$$\gamma_{g,rel}^{DC} = \log \frac{c^C}{c^D}$$
(7)
"

We have also changed **section 4.6** accordingly:

"Due to the higher γmax associated with grazing on coccolithophores as compared to diatoms (Table 1), biomass-normalized specific grazing rates for coccolithophores are higher than those for diatoms for both 40-50$^\circ$S and 50-60$^\circ$S in summer (Fig. 5g & h), resulting in slower biomass accumulation rates for coccolithophores. In summary, in ROMS-BEC, lower biomass-normalized grazing rates make diatoms more successful than coccolithophores in accumulating and sustaining higher biomass concentrations, resulting from a higher per biomass grazing pressure on coccolithophores as compared to that on diatoms between 40-60$^\circ$S."

**SC19**: *Page 8, Line 03: Nevertheless, if (gamma_j / P_j) is called a "clearance rate", you should call "log10((gamma_C / P_D)/(gamma_C / P_D))" as "Relative clearance rate" (instead of "Relative grazing rate").*
Please see our answer to SC18.

**SC20**: *Page 8, Line 04: "the specific grazing (clearance) rate on diatoms is larger" -- as far as I can tell, using the correct gamma_j functional response (with the zgrz + sum[P_i]; see above) the specific grazing (clearance) rate on diatoms will ALWAYS be smaller than on coccos. Plase double check. See my MATLAB/OCTAVE analyses and figures.*
As pointed out in the answer to GC 2, the original grazing parametrization in ROMS-BEC does not include the total phytoplankton biomass in the denominator (see also **Moore et al., 2002, 2004**), and the clearance rate (=biomass-normalized specific grazing rate, see SC18) on coccolithophores is larger than that of diatoms in summer, while the opposite is true for the winter months (see Fig. 5 a&b and g&h in the manuscript). We agree with the reviewer that if total biomass was accounted for in the denominator, differences in the constants gamma_max and zgrz (see Table 1 of manuscript) are

controlling differences in the biomass-normalized specific grazing rate on each phytoplankton, with coccolithophores experiencing the larger grazing pressure due to their higher gamma_max.

**SC21**: *Page 8, Line 25: "caused either by too much uptake by phytoplankton" -- Given the too high mupmax selected for this model I am not surprised about this; why dont you tune down mupmax accordingly to prevent this bias?*
While phytoplankton biomass and NPP are too high at high SO latitudes in ROMS-BEC, they are too low at subantarctic latitudes (see Fig. S2 in supplementary material), the focus area of this study. Since tuning down $\mu_{max}$ will affect growth rates everywhere in the domain, we decided not to increase the negative bias even more by tuning down $\mu_{max}$. We have changed the sentences in the revised manuscript, and it now reads:

"Macronutrients in ROMS- BEC are generally too low at the surface compared to WOA data (especially south of $60^\circ$S, Fig. S1 & S2), caused either by too much nutrient uptake by phytoplankton, too little nutrient supply from below, or both."

**SC22**: *Page 9, Table 3: "(global)" -- what that this means? confusing.*
The reported estimates from the MAREDAT data base in Buitenhuis et al. (2013) are global estimates. Here, we want to put the SO estimate for coccolithophore and diatom biomass from ROMS-BEC into the global context and therefore report the estimates as published in Buitenhuis et al. (2013). We have added a footnote in **Table 3** to explain "global":

"The reported estimates from the MAREDAT data base in Buitenhuis et al. (2013) are global estimates of phytoplankton biomass."

**SC23**: *Page 9, Table 3: The contribution of coccos is: 15%, 10% and 1% for the three regions -- isnt it this too low?*
We thank the reviewer for this comment, but we are not quite sure why the reviewer thinks this is too low. Our estimate is the first estimate for the contribution of coccolithophores to total phytoplankton biomass in the SO as a whole. And as we have pointed out in the discussion section, when comparing our estimates to global NPP, we get a contribution of SO coccolithophores to total global NPP of 5%, which is larger than the published global estimate for the contribution of coccolithophores to total NPP (<2% in **O'Brien, 2015** and **Jin et al, 2006**). In agreement, coccolithophores have also been suggested to be a minor contributor to total global phytoplankton biomass (0.04-6%, **Buitenhuis et al., 2013**). Therefore, if assuming that previous global estimates are accurate, we would rather conclude that our estimate is too high than too low.

**SC24**: *Page 9, Table 3: I dont really see how coccos can even survive based on the uptake curves (nutrient niches) defined in the model set up. The coccos uptake curves seem to be always below those of either diatoms (at high nutrients) and small phyto (at small nutrients). The temperature dependence is almost the same for all phyto groups so it does not play a role. The light dependence is also quite similar for the three dominant groups (C,D,SP). Thus it is mostly down to nutrients competition and if the coccos uptake curve is never above those of their competitors, with a Type II grazing functional response they should be competitively excluded. Only the imposed prey refuge will keep them persisting in the model.*
We thank the reviewer for this comment, but after careful consideration of this point, we do not agree. In BEC, the competitive advantage of group X over group Y (for example diatoms over coccolithophores), defined as the difference in specific growth rate, results from differences in light, temperature and the most-limiting nutrient, as detailed in Equations B4-B9 of the manuscript. Hence, a competitive advantage of group X over group Y can result from multiple factors:
- Small differences in the surrounding temperature enter the temperature limitation equation exponentially, and can therefore lead to substantial differences in the specific growth rate due

to the differences in temperature sensitivity of phytoplankton X and Y (see Eq. B5 in the manuscript, as well as **Fig. R1** and Fig. **R10** in this review).
- For the competition for nutrients, differences in nutrient limitation can be due to differences in the most limiting nutrient, in addition to differences in the half-saturation constants of the same nutrient (see Eq. B6-B8 in manuscript and our answer to GC3).
- Differences in alphaPI across phytoplankton types are small in BEC (see Table 1 of the original manuscript), but differences in light limitation also arise due to differences in nutrient limitation and differences in carbon-to-chlorophyll ratios (see Eq. B9 in the manuscript and **Fig. R1 & R11** in this review).

In our answer to GC 3 and **Fig. R3**, we show that the annual mean nutrient limited growth rate of coccolithophores is indeed larger than that of diatoms in large parts of the subantarctic Southern Ocean. In addition, while the Q10 value is indeed very similar for coccolithophores and diatoms (1.55 for diatoms and 1.45 for coccolithophores), the resulting difference in temperature limitation depends on the surrounding temperature, and can be higher than the difference in Q10 alone (especially at low temperature, see **Fig. R10** and also our answer to comment SC 36).

Hence, coccolithophores can build up biomass relative to diatoms at low nutrient levels (or when/where diatoms become limited by Si), high light levels and low temperatures.

**SC25**: *Page 9, Line 17: "within the globally estimated" -- what do you mean? how can you compare a regional estimate to a whole ocean estimate?*
Thanks for this comment. It is true that a strict comparison is not possible. However, if the regional model estimate was above of the maximum estimate suggested for the globe, we would be able to say that the regional estimate likely severely overestimates the annual mean diatom of coccolithophore biomass in the top 200 m. Since the modeled values are within the range given for global annual mean biomass of both coccolithophores and diatoms, we cannot conclude that. We changed the respective sentence in the manuscript as follows:

"In ROMS-BEC, the annual mean SO coccolithophore carbon biomass within the top 200 m is 0.013 Pg C (Table 3), which is within the globally estimated range based on in-situ observations (0.001-0.03 Pg C, see O'Brien et al., 2013) and suggests that SO coccolithophores contribute substantially to global coccolithophore biomass."

Please see also our answer to SC22.

**SC26**: *Page 9, Figure 1: The model simulates chla fairly well spatially despite some biases -- How well does the model reproduce the seasonal dynamics of chla? Please provide a figure in Supplementary Material.*
We thank the reviewer for this helpful comment. We have added **Fig. R7** (see below) showing the seasonal dynamics of total chlorophyll to the supplementary material.

[Figure]

**Figure R7:** Surface total chlorophyll in ROMS-BEC (black) as compared to satellite chlorophyll (red, MODIS-Aqua climatology) over the course of the year for different latitudinal bands. $r_S$ in top right corner of each panel denotes the Spearman correlation coefficient.

**SC27**: *Page 9, Figure 1, f) Personally I find that the model does a poor job with coccos in terms of spatial distribution, only the absolute values seem to be the correct but clearly not the patterns. The deviation from the data values especially between 60S and 75S is too large, up to an order of magnitude even. Can you say that the model simulations for coccos have been validated really?*

We are not sure whether the reviewer is referring to the diatom evaluation in Fig. 1f) or the coccolithophore evaluation in Fig. 1d) and therefore comment on both.
When considering the match between the absolute biomass concentration without taking into account the uncertainty in the observations, we admit that it looks as if there is substantial model-observation disagreement for both coccolithophores and diatoms (see Fig. 1 in the manuscript, as well as Fig. S4 in the supplementary material), and have pointed this out on page 12 (lines 6-7) in the original manuscript. However, all cell-count-derived phytoplankton carbon biomass estimates used here to validate our model are subject to a large uncertainty (up to a few hundred percent due to the large size range of the cells, see **Le Blanc et al., 2012**, and **O'Brien et al., 2013**, p.12 lines 7-9 of the original manuscript).
In addition to this uncertainty, data coverage is low, especially between 60-75°S (136 observations for coccolithophores, 55 for diatoms, see Fig. 1d & f in manuscript). Available in-situ observations are mainly one-time observations in this area (and for most of the Southern Ocean), and it is thus totally unclear to what extent the data (especially at high latitudes) represent climatological monthly mean conditions. Therefore, we assume the observational estimates south of 60°S to be temporal snapshots rather than true monthly means, especially for diatoms which follow the boom-and-bust strategy, and which can change their cell abundances exponentially by much more than 1 order of magnitude over the course of one week (see also large variability in in-situ diatom biomass estimates in revised version of Fig. 1f below).
The focus area of this study is between 40-60°S, where we have more observations to evaluate the model (726 observations for coccolithophores, 529 for diatoms). Here, the model-data fit is much better, and the simulated biomass of diatoms and coccolithophores is within one standard deviation of the mean observed biomass between 40-65°S and 45-55°S for coccolithophores and diatoms, respectively (see revised version of panel d&f of Fig. 1 below). We note that the diatom biomass estimates from newly compiled observations for this study in this area (Balch et al., 2016, mainly

between 40-65°S) are possibly lower-bound estimates, due to the assumption of all of these cell counts being *F. pseudonana* (nanophytoplankton), as pointed out in section S1 of the supplementary material.

Overall, we think that the model simulated biomasses have been evaluated in the best way possible, given the sparse data coverage in the Southern Ocean. We consider the biomass match-up to be within the range of uncertainty of the observations. We note that we are more confident in the simulated patterns and the relative importance of coccolithophores and diatoms than in the absolute numbers of coccolithophore and diatom biomass – in this respect, the model tuning/evaluation would surely hugely benefit from additional observational PFT biomass estimates.

In the revised version of the manuscript, we have added the standard deviation of the observations to panel d &f of Fig. 1 to illustrate the observed variability:

[Figure]

**Figure R8:** Modified version of panel 1d &f in the manuscript with standard deviation of observations added as the grey bars to illustrate variability. We have included this version of the panels in the revised manuscript.

**SC28**: *Page 10, Line 7: The model NPP estimates are between [0.23 - 0.69] (PgC * y-1) and the real data NPP estimates are between [0.64 - 0.94] (PgC * y-1). This implies that the model's NPP is basically half of the real data NPP estimates -- why do you say then that if "falls with the range estimated by satellite"? Just say that the models underestimates NPP by 50%.*
Agreed, but there is a misunderstanding here. In this sentence of the manuscript, we refer to calcification and not NPP. We agree with the reviewer that we could just say that calcification is underestimated by 50%. However, both the calcification estimate derived from satellites (18.75%) and that from the model (due to uncertainty in CaCO3:C production ratio of coccolithophores) are subject to substantial uncertainty. If we account for these uncertainties, we find that the model-derived estimate (0.23-0.69 PgC yr-1 in original version of the manuscript, 0.28-0.84 PgC yr-1 in revised version) falls within the range suggested by satellite estimates (0.64-0.94 PgC yr-1).

**SC29**: *Page 12, Figure 3: Dizatrophs are basically extinct in this model simulations. This is because the selected uptake curves for them (see my MATLAB/OCTAVE figures) that makes them very poor nutrient competitors. Why do you even bother in having them as a phytoplankton group? I dont understand this.*
We agree with the reviewer on this point. The inclusion of diazotrophs is a result of our group having the desire to use the same ecosystem/biogeochemical model across all regional configurations. This includes regions, such as the tropical North Atlantic, where diazotrophs are very important. In a further sensitivity study, we have set the diazotroph growth rate to zero, and we find that there are no changes in the major findings of our study, and also to the conclusions (see simulated coccolithophore and diatom biogeography in **Fig. R9** below). We changed the section B1 in the appendix as follows:

"Diazotroph growth is zero at temperatures <14°C. For consistency within the user community of BEC, we decided to keep diazotrophs as a phytoplankton PFT, even though the imposed temperature threshold makes them a very minor player in the SO phytoplankton community. A sensitivity study in which $\mu^N$max = 0 showed that the results presented in this study are unaffected by the presence of diazotrophs in BEC (not shown)."

[Figure]

**Figure R9:** Comparison of simulated top 50 m DJFM average coccolithophore (a+c) and diatom (b+d) carbon biomass in Baseline simulation (top row) and a sensitivity simulation in which the max. growth rate of diazotrophs is set to zero. These simulations are shorter (5 years) and done with a coarser resolution setup (0.5° horizontal resolution) than what is presented in the manuscript, but patterns and magnitudes of simulated phytoplankton biomass are directly comparable (see Fig. 1 in the manuscript). The relative contributions to total phytoplankton NPP between 30-90°S are 60.0/60.3 and 17.8/18.1 for diatoms and coccolithophores in the Baseline simulation and the simulation with zero diazotroph growth, respectively.

**SC30**: *Page 12, Lines 15-20: Basically, this means that you don't know why? Is not it any way to explore the reasons beyond verbal speculation?*
We agree with the reviewer that it would be nice to fully understand the source of the high chlorophyll (and NPP) bias at high SO latitudes in our model in more detail, which can generally be caused by biases in bottom-up or top-down factors (or both). We have done sensitivity simulations to assess the impact of biases in the underlying physics quantitatively (TEMP and MLD in Table 2). Both the bias in temperature (generally too warm) and in MLD (too shallow, see Fig. S1 in the supplementary material) stimulate phytoplankton growth. However, while correcting for these biases in the ecosystem subroutine of ROMS-BEC reduces the maximum diatom biomass south of 60°S by 1.5% (TEMP) and 11.3% (MLD), respectively, diatom biomass is still overestimated in the model in these two simulations when comparing to maximum in-situ diatom biomass.

By increasing zooplankton grazing rates on phytoplankton, we can decrease the simulated diatom biomass at high southern latitudes (not shown). However, thereby, we also increase the top-down pressure at subantarctic latitudes, where phytoplankton biomass is already underestimated in the *Baseline* simulation with ROMS-BEC (see Fig. S2 in the supplementary material). Here, further reducing phytoplankton biomass by increasing the grazing pressure will lead to a larger disagreement between model and observations. We therefore conclude that the high biomass bias in our model is due to a mixture of physical and biological shortcomings. We rephrase the respective paragraph of the manuscript as follows:

"[…] Acknowledging the substantial uncertainty of the observational estimates (165% for the carbon biomass in Fig. 1f, on average at least 20% for satellite derived chlorophyll estimates in Soppa et al. (2014)), both in-situ observations (Fig. 1f) and satellite derived diatom chlorophyll (Soppa et al., 2014, comparison not shown) suggest an overestimation of surface diatom biomass in ROMS-BEC south of $60°$S during austral summer. However, this overestimation in the model can partly be explained by biases in the underlying physics (see section 4.1, with maximum diatom biomass south of $60°$S being 1.5% and 11.3% lower in the simulations TEMP and MLD, respectively). Additionally, missing ecosystem complexity within the zooplankton compartment of ROMS-EBC probably adds to the overestimation of high latitude phytoplankton biomass, as suggested by Le Quéré et al. (2016). In their model, Le Quéré et al. (2016) only simulate total chlorophyll levels comparable to those suggested by satellite observations when including slow-growing macro-zooplankton as well as trophic cascades within the zooplankton compartment of their model, while overestimating satellite-derived chlorophyll levels otherwise."

**SC31**: *Page 13, Line 1: "this PFT (Phaeocystis) is not included in our simulations" -- Honestly I don't think this is a valid excuse. Given that diazotrophs are irrelevant in this model simulations, why don't you use that tracer to model Phaeocystis instead?*
As the reviewer knows (e.g. **LeQuéré et al. 2016**), including further plankton functional types is a laborious task that requires a careful retuning and testing of the model. As a matter of fact, we are currently working on a version of the Southern Ocean set-up of the model that does include Phaeocystis (Nissen et al. in prep.), but its description within the context of the current paper is unnecessary, since the area south of 60°S is not the focus region of this study, and since we are interested in diatom-coccolithophore interactions in the Great Calcite Belt.

**SC32**: *Page 13, Line 4: "simulated gradient ... coccos contribution" -- This is a misleading statement: diatoms clearly dominate everywhere in the model by about a factor of x10, so talking about mixed community, south-north increase in coccos contribution, etc. is verbally misleading.*
We thank the reviewer for this comment and agree with him that it is rather subjective to talk about a "more mixed community" if diatoms dominate everywhere south of 40°S in our model domain. We rephrase this sentence, and it now states:

"CHEMTAX data (based on HPLC data) support the simulated gradient from a clearly diatom dominated community south of $60°$S to a more mixed community north thereof with a south-north increase of the coccolithophore contribution (maximum contribution of >20% of total NPP north of $45°$S, see Fig. 2a) [...]"

**SC33**: *Page 13, Line 13: "whereas diatom biomass peaks south of 60S where they dominate the community" -- again, this is misleading; diatoms dominate the community \*everywhere\* in your domain, not just south of 60S.*
Agreed, it should be clearer that diatoms also dominate between 40-60°S. We have changed the sentence to "[…] diatom biomass peaks south of $60°$S where they dominate the community by far (>80% of total NPP, see Fig. 2a)" to make it clearer. Please see also answer to SC32.

**SC34**: *Page 15, Line 10: "the specific growth rate of coccos is on average 10% larger than of diatoms" -- IMPORTANT: This statement is wrong due to a conceptual misunderstanding. Doing*

*log(x/y) where x = DIN / (kdin + DIN) for diatoms and y is the equivalent thing for coccos does \*not\* measure the "relative growth rate" of diatoms versus coccos but the "relative growth limitation", which is not the same thing. If you want to evaluate "relative growth rate" you have to do: Rel growth = log(umax_D/umax_C) + log(x/y) -- Please change this in the analysis and the text accordingly.*
Thanks for this comment, but in the black solid line in Fig 5a&b as well as in the dark grey bars in Fig. 6 we indeed refer to the ratio of the specific growth rates of coccolithophores and diatoms (=relative growth ratio as defined in Eq. 4), calculated as:

$$
\begin{aligned}
\mu_{\text{rel}}^{\text{DC}} \quad &= \log \frac{\mu^{\text{D}}}{\mu^{\text{C}}} \\
&= \underbrace{\log \frac{\mu_{\text{max}}^{\text{D}}}{\mu_{\text{max}}^{\text{C}}}}_{\beta_{\mu_{\text{max}}}} + \underbrace{\log \frac{f^{\text{D}}(T)}{f^{\text{C}}(T)}}_{\beta_{\text{T}}} + \underbrace{\log \frac{g^{\text{D}}(N)}{g^{\text{C}}(N)}}_{\beta_{\text{N}}} + \underbrace{\log \frac{h^{\text{D}}(I)}{h^{\text{C}}(I)}}_{\beta_{\text{I}}}
\end{aligned}
\tag{4}
$$

In the revised version of the manuscript, we explicitly refer to the respective equation in this part of the manuscript:

"In ROMS-BEC, the latitudinal band between 40-50°S is the area with the highest coccolithophore biomass in austral summer (see Fig. 1d and Fig. 4). The relative growth ratio of diatoms vs. coccolithophores between 40-50°S (solid black line in Fig. 5a) is negative from the end of September until the end of April ($\mu^{\text{Cocco}} > \mu^{\text{Diatoms}}$, see Eq. 4). For the summer months (December-March, DJFM), the specific growth rate of coccolithophores is on average 15% larger than that of diatoms (Fig. 6a, shaded dark grey bar, calculated from non-log transformed ratios), […]"

Please see section 3 of the manuscript for further details and note also our answer to GC1 and SC39.

**SC35**: *Page 15, Line 18: "The 21% larger umax of diatoms ... all year round in the whole model domain" -- This is a main result of the model simulations. Thus it must be at the beginning of section 4.5 not buried here.*
We thank the reviewer for this comment, but the 21% larger $\mu_{\text{max}}$ (maximum growth rate) is not a result of the model simulation, but a result of the predefined parameters as reported in Table 1 of the manuscript. We now refer to Table 1 earlier in the respective sentence to make this clearer:

"The 21% larger µmax of diatoms compared to that of coccolithophores (Table 1) favors diatom relative to coccolithophore growth all year round in the whole model domain (term βµmax in Eq. 4, green area in both Fig. 5a & b is positive)."

**SC36**: *Page 15, Line 21: "coccos being less temperature limited" -- Misleading, the differences of Q10 are marginal (7% is basically nothing)*
We thank the reviewer for this comment. There was a typo in the reported temperature limitation function (Eq. B5), which we have corrected in the revised manuscript. However, red areas in Fig. 5 & 6 were correctly calculated and thus, all relevant figures in the original manuscript were unaffected by this issue: Instead of

$$ f^{i}(T) = Q_{10}^{i} \cdot \exp\left(\frac{T - T_{\text{ref}}}{10°C}\right) $$

Eq. B5 should read

$$ f^{i}(T) = Q_{10}^{i}{}^{\frac{T - T_{\text{ref}}}{10°C}} $$

Hence, since the temperature limitation function is non-linear, a 7% difference between the Q10 of coccolithophores vs. the Q10 of diatoms does not result in a temperature limitation difference of 7% everywhere in the domain. In fact, at temperatures <20°C (as observed/simulated for 40-60°S and

especially south of 60°S, data not shown in the manuscript), the difference in temperature limitation is >7% (7% at 20°C, 10% at 15°C, 13% at 10°C, 15% at 5°C), see also red bars in Fig. 6 and **Fig. R10** below.

[Figure]

**Figure R10:** Difference in corrected temperature limitation function ($\dot{f}(T) = Q10^i \wedge (T-T_{ref}/10°C)$) with $T_{ref}$=30°C between when using a Q10 of 1.45 (coccolithophores, black curve in left panel) vs Q10 of 1.55 (diatoms, blue curve)). The right panel shows the ratio of the temperature limitation between diatoms and coccolithophores as a function of temperature. The difference is >7% at temperatures <20°C. See also SC 36.

**SC37**: *Page 15, Line 24: "are less nutrient limited" -- This is the right wording. One thing is being less nutrient limited and another thing is having a faster nutrient uptake curve, the difference lies on the umax.*
Agreed.

**SC38**: *Page 15, Line 35: "differences in the sensitivity to increases of PAR at low irradiance (alfa PI) and diffs in photoaclim" -- Misleading, the differences of alfa_PI are marginal (9% is basically nothing)*
We agree with the reviewer that the differences in alphaPI are small. We modified the statement, which now reads:

"In ROMS-BEC, differences in light limitation between coccolithophores and diatoms are controlled by the minor difference in the sensitivity to increases of PAR at low irradiances (αPI) and largely by differences in photoacclimation […]"

**SC39**: *Page 16, Figure 5: The different terms that are being plotted here as "relative growth ratio" for nutrients, temperature, light, and total are not really "growth rate" ratios" but "growth limitation" ratios. This leads to wrong interpretation of the results. The right way should be to plot the following curves:*
*Qdin = DIN / (Kdin + DIN); Nutrients limitation [0 - 1] n.d.*
*Qpar = f(PAR); Irradiance limitation [0 - 1] n.d.*
*Qsst = f(SST); Temperature limitation [0 - 1] n.d.*
**Rel_umax = log(umax_D/umax_C);**
**Rel_Qdin = log(Qdin_D/Qdin_C); -- Relative nutrient growth "limitation"**

***Rel_Qpar = log(Qpar_D/Qpar_C); -- Relative irradiance growth "limitation"***
***Rel_Qsst = log(Qsst_D/Qsst_C); -- Relative temperature growth "limitation"***
*Rel_din = Rel_umax + Rel_Qdin; -- Relative nutrient growth "rate"*
*Rel_par = Rel_umax + Rel_Qpar; -- Relative irradiance growth "rate"*
*Rel_sst = Rel_umax + Rel_Qsst; -- Relative temperature growth "rate"*
*Rel_Qlim = Rel_Qdin + Rel_Qpar + Rel_Qsst; -- Relative total growth "limitation"*
*Rel_growth = Rel_mupmax + Rel_Qlim; -- Relative total growth "rate"*

We thank the reviewer for this comment and apologize for the misunderstanding. We are very aware of the difference between the relative growth ratio and ratios of the growth limitation terms. We want to point out that everywhere in the manuscript, we refer to the colored areas/bars in Fig 5a&b/Fig 6 as the *contributions to the relative growth ratio*, in accordance with Eq. 4 in section 3. What is plotted in colors in Fig.5 and Fig.6 corresponds exactly to the equations highlighted in bold above that were suggested by the reviewer. To clarify and avoid confusion, we have added the following statement in the methods **section 3**, where we define the relative growth ratio:

$$
\begin{aligned}
\mu_{\text{rel}}^{\text{DC}} \quad &= \log \frac{\mu^{\text{D}}}{\mu^{\text{C}}} \\
&= \underbrace{\log \frac{\mu_{\text{max}}^{\text{D}}}{\mu_{\text{max}}^{\text{C}}}}_{\beta_{\mu_{\text{max}}}} + \underbrace{\log \frac{f^{\text{D}}(\text{T})}{f^{\text{C}}(\text{T})}}_{\beta_{\text{T}}} + \underbrace{\log \frac{g^{\text{D}}(\text{N})}{g^{\text{C}}(\text{N})}}_{\beta_{\text{N}}} + \underbrace{\log \frac{h^{\text{D}}(\text{I})}{h^{\text{C}}(\text{I})}}_{\beta_{\text{I}}}
\end{aligned}
\qquad (4)
$$

«In this equation, the terms $\beta_{\mu max}$ , $\beta_T$, $\beta_N$, and $\beta_I$ describe the differences in the maximum growth rate $\mu_{max}$, temperature limitation $f(T)$, nutrient limitation $g(N)$, and light limitation $h(I)$ between diatoms and coccolithophores, which in sum give the difference in the relative growth ratio $\mu^{\text{DC}}$. «

Furthermore, we have changed the captions of Fig. 5 and Fig. 6 as follows:

**Fig. 5**: "Colored areas are contributions of the maximum growth rate $\mu_{max}$ (green), nutrient limitation (blue), light limitation (yellow) and temperature sensitivity (red) to the relative growth ratio, i.e. the red area e.g. represents the term $\beta_T$ of Eq. 4 (see section 3)."

**Fig. 6**: "Percent difference in growth rate (dark grey), growth-limiting factors (maximum growth rate $M_{max}$ in green, nutrient limitation in blue, light limitation in yellow and temperature sensitivity in red) and grazing rate (light grey) of diatoms and coccolithophores for a) 40-50°S and b) 50-60°S. Respective left bar shows the December-March average (DJFM) calculated from the non-log transformed ratios (i.e. the red bar e.g. represents $10^{\beta_T}$, see Eq. 4), the shaded right bars show the average for all other months (non-DJFM). Full seasonal cycle is shown in Fig. 5a & b"

In section 4.5, we changed the text to refer back to the definition of the individual terms of Eq. 4:

"The relative growth ratio can be separated into the contribution of the maximum growth rate $\mu_{max}$ ($\beta_{\mu max}$ ), temperature 25 ($\beta_T$), nutrients ($\beta_N$), and light ($\beta_I$), which all affect phytoplankton growth (see Eq. 4, colored areas in Fig. 5a & b and Fig. 6). "

Furthermore, we refer back to the individual terms of Eq. 4 in the respective paragraphs addressing the limitation terms:

"[…] Table 1, term $\beta_{\mu max}$ in Eq. 4, green area in both Fig. 5a & b […]"

"[…] Table 1, term $\beta_T$ in Eq. 4, red area in Fig. 5a […]"

"[…] see Fig. 6, shaded blue bars and term $\beta_N$ in Eq. 4 […]"

"[…] see Fig. 6, shaded yellow bars and term βI in Eq. 4[…]"

**SC40**: *Page 17, Figure 6b) -- "Relative grazing ratio higher on diatoms" -- This is very weird, I think this computation of grazing ratio is wrong. When I did on my MATLAB/OCTAVE analyses I find that the "grazing ratio" (in fact it should be called "clearance rate" ratio) is ALWAYS higher on coccos. And if we look at the grazing functional response we can se that it is higher than those for coccos at any prey concentration. I think the reason is the lack of the (zgrz + sum[Pi]) at the denominator of the grazing computation. If you are using (zgrz + P1), instead of (zgrz + P1 + P2), that will be wrong. Please double-check.*
Please see our answer to GC2 and SC20.

**SC41**: *Page 17, Line 1: "coccos have a lower alfa PI" -- Only 9% lower, please explicitly say so.*
Agreed, we have changed the text accordingly.

**SC42**: *Page 17, Line 2: "a generally lower chla-to-carbon" ratio (not shown) -- How much lower in percentage? Why it is not shown?*
As seen in **Fig. R11** below, the annual mean carbon-to-chlorophyll ratio of diatoms between 40-60°S varies between 50-60 mg C mg chl$^{-1}$ (0.017-0.02 mg chl mg C$^{-1}$) in ROMS-BEC, while it varies between 60-90 mg C mg chl$^{-1}$ (0.011-0.017 mg chl mg C$^{-1}$) for coccolithophores. We have added the corresponding figure to the supplementary material, and refer to the figure in the main text, where the modified statement now states:

"Coccolithophores have a 9% lower αPI (Table 1), a generally lower chlorophyll-to-carbon ratio (Fig. S12) [...]"

[Figure]

**Figure R11:** Annual mean surface carbon-to-chlorophyll ratios [mg C mg chl$^{-1}$] of a) diatoms and b) coccolithophores in ROMS-BEC. The black contour corresponds to a ratio of 80 mg C mg chl$^{-1}$.

**SC43**: *Page 17, Line 4: "coccos are on average 2% - 3% more light limited than diatom" -- This is basically nothing from a competitive exclusion point of view at ecological (seasonal) time scales.*
Agreed. We changed the sentence as follows:

"In austral summer, the light limitation of coccolithophores is not significantly different from that of diatoms between 40-50°S and 50-60°S respectively (4% and 1%, see Fig. 6, shaded yellow bars and term βI in Eq. 4)."

**SC44**: *Page 17, Line 11: "coccos and diatom together contribute on average 87% and 95%" -- Misleading statement because diatoms clearly dominate (90% diatoms vs 10% coccos); dont plug them together.*
Thank you for this comment, but here, we add these contributions to introduce the (short) discussion of small phytoplankton, which contribute to total phytoplankton biomass as well (see Fig. 3 in the manuscript), thus competing for nutrients with coccolithophores and diatoms. To avoid misinterpretation, we changed the respective part of the manuscript, which now reads:

"Coccolithophores and diatoms together contribute on average 87% and 96% to total DJFM mean surface phytoplankton biomass between 40-50$^\circ$S and 50-60$^\circ$S, respectively (Fig. 3), with diatoms constituting the majority of this biomass. This leaves 13% and 4% for small phytoplankton, whose contribution to total biomass levels is thus of the same order of magnitude as that of coccolithophores."

**SC45**: *Page 17, Line 13: "They are thus not only competing for resources between each other but with SP as well" -- This statement is obvious; of course all phytoplankton PFT compete among them for nutrients.*
Agreed.

**SC46**: *Page 18, Line 5: "advantage in specific growth" -- LIMITATION not RATE. This must be clear.*
In fact, coccolithophores have an advantage in specific growth RATE (see relative growth ratio, black line in Fig 5a &b, as well as blue compared to red line in panel e&f).

**SC47**: *Page 18, Line 6: "greater importance" -- Misleading, coccos are always poor competitors in this model simulations.*
Thanks for this comment. We agree with the reviewer that coccolithophores are always of minor importance relative to diatoms between 40-60°S. However, in this sentence we only compare the relative importance of coccolithophores between 40-50°S to that between 50-60°S. We have changed the respective line to clarify, which now reads:

"In summary, coccolithophores have an advantage in specific growth relative to diatoms in austral summer both between 40-50$^\circ$S and 50-60$^\circ$S. Comparing the two latitudinal bands, this advantage is higher for 40-50$^\circ$S, explaining the 10% greater importance of coccolithophores for total phytoplankton biomass in this band as compared to 50-60$^\circ$S (annual mean, Fig. 3)."

**SC48**: *Page 18, Line 10: "higher specific growth rate" -- WRONG: this should say "higher specific growth limitation"; it is \*not\* the same thing.*
As pointed out above in our response to GC1, we are aware that these are two different things, but here, we indeed refer to the higher specific growth rate.

**SC49**: *Page 18, Line 11: "We calculated whether the length of the growing season is long enough" -- Good try, but wrong answer. Coccos may grow faster (in days-1) than diatoms over a very narrow band of nutrient concentration which I dont think is even hapening on your models simulations. I suspect that diatoms grow faster (in days-1) than coccos everywhere in your domain and everytime in the year. Plase double-check.*
Thanks for this comment, but coccolithophore growth is indeed faster than diatom growth across 40-60°S of our model domain for parts of the year. The respective temporal domain over which this occurs is shown in Fig. 5a&b (black solid line) and c&d of the original manuscript. The respective regime is associated with low nutrient conditions (see **Fig. R3** for the difference in nutrient-limited growth rates between diatoms and coccolithophores in this review in our answer to GC3), high light levels (so that differences in light limitation between coccolithophores and diatoms are marginal, see yellow area in Fig. 5&b of the original manuscript), and temperatures <20°C (so that coccolithophore growth is limited less by temperature than diatom growth, see **Fig. R10** in this review). Please see also

our answer to GC3 where we show that the nutrient-limited growth rate of coccolithophores is larger than that of diatoms over large areas of the focus area of this study (especially north of 50°S), because the model considers only the most-limiting nutrient for the growth rate calculation.

**SC50**: *Page 18, Line 25: If grazing pressure were able to explain the mismatch between the expected results for coccos vs diatoms (from the model) and observed ones (from the model), this could be easily confirmed by performing a run where maximum grazing rate (gamma max) and half saturation constant for grazing (zgrz) are the same for both coccos and diatoms. Please do it and report the results in Supp. Material.*

Thanks for making this comment. The run the reviewer is suggesting has already been included in the analysis of our original manuscript and was termed our sensitivity run GRAZING (see Table 2 of the manuscript). Our calculation using the simulated growth advantage of coccolithophores over diatoms and the biomass ratio of the two at the beginning of the growth season suggests that coccolithophores should manage to outcompete diatoms in terms of biomass between 40-50°S, but not between 50-60°S (due to the too small advantage in growth and too large differences in biomass, see last paragraph of section 4.5 in the original manuscript) - if there is no differences in loss rates between the two phytoplankton types. In fact, the biomass evolution of diatoms and coccolithophores in the simulation GRAZING confirms what our calculation and the reviewer suggest: if coccolithophores experience the same grazing pressure as diatoms, the growth advantage of coccolithophores relative to diatoms between 40-50°S is large enough for coccolithophores to dominate in terms of biomass towards the end of the growth season (see **Fig. R12a** below). However, the growth advantage is not large enough between 50-60°S and diatom biomass is larger than coccolithophore biomass all year round (see **Fig. R12b**).

We revised this part in section 4.5 of the manuscript which now states:

"For 40-50˚S, however, our calculations show that despite the 10 times higher biomass of diatoms at the end of November (Fig. 5c), coccolithophores should outcompete diatoms at the end of March with a 15% higher specific growth rate if loss rates are the same for both PFTs. This finding is confirmed by the sensitivity simulation GRAZING in which diatoms and coccolithophores experience the same loss rates (see section 4.7), and coccolithophore biomass is indeed larger than that of diatoms between January and March for 40-50˚S (not shown)."

[Figure]

**Figure R12:** Evolution of surface PFT biomass between a) 40-50°S and b) 50-60°S for the sensitivity simulation GRAZING.

**SC51:** *Page 18, Line 28: "specific grazing rate on coccos" -- it is not clear to me if this is measuring relative "specific grazing rates" (d-1 vs d-1) or relative "specific clearance rates" (m3 * mol-1 * d-1) vs. (m3 * mol-1 * d-1). Ideally it should measure (d-1 vs d-1) to be correct and consistent.*
Please see our answer to GC1.

**SC52**: *Page 18, Line 34: "differences in specific grazing rates between diatoms and coccos are of similar magnitude as differences in specific growth rates" -- WRONG: You cannot compare differences in specific clearance rates (I am not sure about the units yet; maybe [d-1] or maybe [m3 * mol-1 * d-1]) with specific LIMITATION rates (in non dimensional [n.d.] units), even when having*

*them log transformed so that their both lose their original units. Make sure that "relative grazing rate" and "relative growth rate" is based on process with units of days-1 in both cases. Otherwise they are not comparable.*
Please see our answer to GC1.

**SC53**: *Page 19, Line 7: "During this times coccos experience a larger per biomass grazing pressure" -- in fact coccos experience a larger per biomass grazing pressure at ALL TIMES. Please double check and correct the text accordingly (e.g. line 10)*
We thank the reviewer for this comment, but assume that confusion arises due to a misunderstanding about the terminology used to define the grazing pressure (see comment GC1 above). We double-checked in our results and confirm what we show in Fig. 5 a&b, as well as g&h of the original manuscript: The per biomass grazing pressure (in $[d^{-1}]$) as defined by equation 7 of our manuscript (Eq. 6 in the revised version) is higher on coccolithophores during much of spring, summer, and fall (depends on the latitudinal band), but is indeed smaller than that on diatoms in winter (as well as parts of spring and fall for e.g. 50-60°S). This can be seen from the negative relative grazing ratio in Fig. 5 a&b, as well as the smaller clearance rate (in fact equivalent to a smaller specific grazing rate, see our answer to SC18) in Fig. 5 g&h.

**SC54**: *Page 19, Line 17: "Parameter Sensitivity Simulations" -- Change to "Parameter Sensitivity Analysis" and perform the SA suggested in my General Comments. The current SA is not meaningful.*
Please see our answer to GC4.

**SC55**: *Page 20, Line 18: "coccos are a non-negligible" -- Change to "coccos are a minor but non-negligible"*
Agreed, we have changed the text accordingly.

**SC56**: *Page 21, Line 20: "The net sign of ... future research" -- Why future research? I think this may actually be the most important point to be addressed by this work. Why cannot be done now?*
We agree with the reviewer that this is an important point to be addressed. This is indeed ongoing work by the first author. In order not to overload this paper, we decided to focus on the factors controlling phytoplankton biogeography in this paper, and to study the biogeochemical implications of the resulting phytoplankton community structure in a follow-up paper. This will allow us to give the latter analysis the space and thoroughness it deserves.

**SC57:** *Page 21, Line 26: "succession" -- Margalef's Mandala concept of succession implies a temporal dominance. However coccos do never get anywhere close to dominate the biomass since diatoms are always above 80%. Therefore the term "succession" does not apply here.*
In the literature, different definitions exist for the concept termed "succession". To our knowledge, succession does not always have to imply dominance in terms of biomass. While **Margalef (1978)** mentions the term "dominance" when talking about phytoplankton succession, he is not explicit as to whether he refers to dominance in terms of specific growth rate or dominance in terms of biomass. He says that "diatoms will become dominant in turbulent water rich in nutrients" – it will thus depend on their initial relative importance for total phytoplankton biomass and on how long these conditions are sustained whether they also become the dominant phytoplankton in terms of biomass at a given location. **Balch (2004)** expanded the original mandala by Margalef describing the succession of phytoplankton types by a day length axis to include coccolithophores. **Balch (2004)**, as well as novel papers such as e.g. **Romagnan et al. (2015)** describe the succession as an increase in abundance of the respective phytoplankton type which does not necessarily results in a dominance of that type.
We changed the beginning of the discussion **section 5.2**, which now reads:

"In ROMS-BEC, coccolithophore blooms start and peak later than those of diatoms between 40-60°S (Fig. 4), in agreement with the updated version of Margalef's mandala by Balch (2004), predicting the succession of these phytoplankton functional types as a result of changing environmental conditions over time (see also Margalef, 1978)."

**SC58**: *Page 21, Line 24: "specific growth rate" -- Change to "specific growth limitation" or compute the correct "specific growth rate" after multiplying by umax. Currently what is higher is the nutrient limitation "Qdim = DIN / (kdin + DIN)" term (n.d.) but not the growth rate "umax * Qdim" term (d-1)*
Please see our answer to GC1.

**SC59**: *Page 23, Figure 8: This figure is pretty but to complex -- Too many info (colors, shades, arrows, shapes, letters, low high); I honestly don't understand anything.*
We thank the reviewer for this comment. In the revised version of the manuscript, we have updated the original Fig. 8 (see **Fig. R13** below). In the revised version of the figure, instead of showing the specific grazing pressure on coccolithophores and diatoms through the thickness of arrows, we omitted the arrows and introduced a second circle for each phytoplankton group, whose thickness represents the specific grazing pressure. We included white arrows only to illustrate the coupling between the specific grazing pressure (single-colored ring) and the relative importance of diatoms and coccolithophores for total phytoplankton biomass (multi-colored ring), respectively.

[Figure]

**Figure R13:** Updated version of Fig. 8 in the manuscript.

**SC60**: *Page 25, Line 1: "pressure on less abundant" -- Change to "pressure on relatively less abundant".*
Agreed, we have changed the text accordingly.

**SC61**: *Page 26, Line 9: "coccos biomass is high when diatoms" -- Change to "coccos biomass is higher"; their biomass is never high.*
Agreed, we have changed the text accordingly.

**SC62**: *Page 26, Line 11: "never exceeds that of" -- Change to "never gets even close to" that of diatoms.*
Agreed, we have changed the text accordingly.

**Cited literature:**

Anderson, T. R. (2005). Plankton functional type modelling: running before we can walk? *Journal of Plankton Research*, *27*(11), 1073–1081. https://doi.org/10.1093/plankt/fbi076

Balch, W. M. (2004). Re-evaluation of the physiological ecology of coccolithophores. In H. R. Thierstein & J. R. Young (Eds.), *Coccolithophores - From Molecular Processes to Global Impact* (pp. 165–190). Berlin: Springer.

Balch, W. M., Bates, N. R., Lam, P. J., Twining, B. S., Rosengard, S. Z., Bowler, B. C., Drapeau, D. T., Garley, R., Lubelczyk, L. C., Mitchell, C., Rauschenberg, S. (2016). Factors regulating the Great Calcite Belt in the Southern Ocean and its biogeochemical significance. *Global Biogeochemical Cycles*, 1199–1214. https://doi.org/10.1002/2016GB005414

Buitenhuis, E. T., Pangerc, T., Franklin, D. J., Le Quéré, C., & Malin, G. (2008). Growth rates of six coccolithophorid strains as a function of temperature. *Limnology and Oceanography*, *53*(3), 1181–1185. https://doi.org/10.4319/lo.2008.53.3.1181

Buitenhuis, E. T., Vogt, M., Moriarty, R., Bednaršek, N., Doney, S. C., Leblanc, K., Le Quéré, C., Luo, Y. W., O'Brien, C., O'Brien, T., Peloquin, J., Schiebel, R., Swan, C. (2013). MAREDAT: Towards a world atlas of MARine Ecosystem DATa. Earth System Science Data, 5, 227–239. https://doi.org/10.5194/essd-5-227-2013

Cubillos, J. C., Wright, S. W., Nash, G., de Salas, M. F., Griffiths, B., Tilbrook, B., Poisson, A., Hallegraeff, G. M. (2007). Calcification morphotypes of the coccolithophorid Emiliania huxleyi in the Southern Ocean: changes in 2001 to 2006 compared to historical data. *Marine Ecology Progress Series*, *348*, 47–54. https://doi.org/10.3354/meps07058

Daniels, C. J., Sheward, R. M., & Poulton, a. J. (2014). Biogeochemical implications of comparative growth rates of *Emiliania huxleyi* and *Coccolithus* species. *Biogeosciences*, *11*(23), 6915–6925. https://doi.org/10.5194/bg-11-6915-2014

Heinle, M. (2013). *The effects of light, temperature and nutrients on coccolithophores and implications for biogeochemical models*. University of East Anglia.

Geider, R. J., MacIntyre, H. L., & Kana, T. M. (1998). A dynamic regulatory model of phytoplanktonic acclimation to light, nutrients, and temperature. *Limnology and Oceanography*, *43*(4), 679–694. https://doi.org/10.4319/lo.1998.43.4.0679

Gentleman, W., Leising, A., Frost, B., Strom, S., & Murray, J. (2003). Functional responses for zooplankton feeding on multiple resources: a review of assumptions and biological dynamics. *Deep Sea Research Part II: Topical Studies in Oceanography*, *50*(22–26), 2847–2875. https://doi.org/10.1016/j.dsr2.2003.07.001

Gravalosa, J. M., Flores, J.-A., Sierro, F. J., & Gersonde, R. (2008). Sea surface distribution of coccolithophores in the eastern Pacific sector of the Southern Ocean (Bellingshausen and Amundsen Seas) during the late austral summer of 2001. *Marine Micropaleontology*, *69*(1), 16–25. https://doi.org/10.1016/j.marmicro.2007.11.006

Hashioka, T., Vogt, M., Yamanaka, Y., Le Quéré, C., Buitenhuis, E. T., Aita, M. N., Alvain, S., Bopp, L., Hirata, T., Lima, I., Sailley, S. Doney, S. C. (2013). Phytoplankton competition during the spring bloom in four plankton functional type models. *Biogeosciences*, *10*(11), 6833–6850. https://doi.org/10.5194/bg-10-6833-2013

Jin, X., Gruber, N., Dunne, J. P., Sarmiento, J. L., & Armstrong, R. A. (2006). Diagnosing the contribution of phytoplankton functional groups to the production and export of particulate organic carbon, CaCO3 , and opal from global nutrient and alkalinity distributions. *Global Biogeochemical Cycles*, *20*(2), GB2015. https://doi.org/10.1029/2005GB002532

Krumhardt, K. M., Lovenduski, N. S., Iglesias-Rodriguez, M. D., & Kleypas, J. A. (2017). Coccolithophore growth and calcification in a changing ocean. *Progress in Oceanography*, *159*(June), 276–295. https://doi.org/10.1016/j.pocean.2017.10.007

Leblanc, K., Arístegui, J., Armand, L., Assmy, P., Beker, B., Bode, A., Breton, E., Cornet, V., Gibson, J., Gosselin, M.-P., Kopczynska, E., Marshall, H., Peloquin, J., Piontkovski, S., Poulton, A. J., Quéguiner, B., Schiebel, R., Shipe, R., Stefels, J., van Leeuwe, M. A., Varela, M., Widdicombe, C., Yallop, M. (2012). A global diatom database – abundance, biovolume and biomass in the world ocean. Earth System Science Data, 4(1), 149–165. https://doi.org/10.5194/essd-4-149-2012

Le Quéré, C., Harrison, S. P., Colin Prentice, I., Buitenhuis, E. T., Aumont, O., Bopp, L., Claustre, H., Cotrim Da Cunha, L., Geider, R., Giraud, X., Klaas, C., Kohfeld, K. E., Legendre, L., Manizza, M., Platt, T., Rivkin, R. B., Sathyendranath, S., Uitz, J., Watson, A. J., Wolf-Gladrow, D. (2005). Ecosystem dynamics based on plankton functional types for global ocean biogeochemistry models. *Global Change Biology*, *11*, 2016–2040. https://doi.org/10.1111/j.1365-2486.2005.1004.x

Le Quéré, C., Buitenhuis, E. T., Moriarty, R., Alvain, S., Aumont, O., Bopp, L., Chollet, S., Enright, C., Franklin, D. J., Geider, R. J., Harrison, S. P., Hirst, A. G., Larsen, S., Legendre, L., Platt, T., Prentice, I. C., Rivkin, R. B., Sailley, S., Sathyendranath, S., Stephens, N., Vogt, M., Vallina, S. M. (2016). Role of zooplankton dynamics for Southern Ocean phytoplankton biomass and global biogeochemical cycles. *Biogeosciences*, *13*(14), 4111–4133. https://doi.org/10.5194/bg-13-4111-2016

Luo, Y.-W., Doney, S. C., Anderson, L. A., Benavides, M., Berman-Frank, I., Bode, A., Bonnet, S., Boström, K. H., Böttjer, D., Capone, D. G., Carpenter, E. J., Chen, Y. L., Church, M. J., Dore, J. E., Falcón, L. I., Fernández, A., Foster, R. A., Furuya, K., Gómez, F., Gundersen, K., Hynes, A. M., Karl, D. M., Kitajima, S., Langlois, R. J., LaRoche, J., Letelier, R. M., Marañón, E., McGillicuddy, D. J., Moisander, P. H., Moore, C. M., Mouriño-Carballido, B., Mulholland, M. R., Needoba, J. A., Orcutt, K. M., Poulton, A. J., Rahav, E., Raimbault, P., Rees, A. P., Riemann, L., Shiozaki, T., Subramaniam, A., Tyrrell, T., Turk-Kubo, K. A., Varela, M., Villareal, T. A., Webb, E. A., White, A. E., Wu, J., Zehr, J. P. (2012). Database of diazotrophs in global ocean: abundance, biomass and nitrogen fixation rates. Earth System Science Data, 4(1), 47–73. https://doi.org/10.5194/essd-4-47-2012

Margalef, R. (1978). Life-forms of phytoplankton as survival alternatives in an unstable environment. *Oceanologica Acta*, *1*(4), 493–509. Retrieved from http://link.springer.com/10.1007/BF00202661

Moore, J. K., Doney, S. C., Kleypas, J. A., Glover, D. M., & Fung, I. Y. (2002). An intermediate complexity marine ecosystem model for the global domain. Deep Sea Research Part II: Topical Studies in Oceanography, 49(1–3), 403–462. https://doi.org/10.1016/S0967-0645(01)00108-4

Moore, J. K., Doney, S. C., & Lindsay, K. (2004). Upper ocean ecosystem dynamics and iron cycling in a global three-dimensional model. *Global Biogeochemical Cycles*, *18*(4), GB4028. https://doi.org/10.1029/2004GB002220

Moore, J. K., Lindsay, K., Doney, S. C., Long, M. C., & Misumi, K. (2013). Marine Ecosystem Dynamics and Biogeochemical Cycling in the Community Earth System Model

[CESM1(BGC)]: Comparison of the 1990s with the 2090s under the RCP4.5 and RCP8.5 Scenarios. *Journal of Climate*, *26*(23), 9291–9312. https://doi.org/10.1175/JCLI-D-12-00566.1

Nielsen, M. V. (1997). Growth, dark respiration and photosynthetic parameters of the coccolithophorid Emiliania Huxleyi (Prymnesiophyceae) acclimated to different day length-irradiance combinations. *Journal of Phycology*, *33*(5), 818–822. https://doi.org/10.1111/j.0022-3646.1997.00818.x

O'Brien, C. J., Peloquin, J. A., Vogt, M., Heinle, M., Gruber, N., Ajani, P., Andruleit, H., Arístegui, J., Beaufort, L., Estrada, M., Karentz, D., Kopczyńska, E., Lee, R., Poulton, A. J., Pritchard, T., Widdicombe, C. (2013). Global marine plankton functional type biomass distributions: coccolithophores. *Earth System Science Data*, *5*(2), 259–276. https://doi.org/10.5194/essd-5-259-2013

Prowe, A. E. F., Pahlow, M., Dutkiewicz, S., Follows, M., & Oschlies, A. (2012). Top-down control of marine phytoplankton diversity in a global ecosystem model. *Progress in Oceanography*, *101*(1), 1–13. https://doi.org/10.1016/j.pocean.2011.11.016

Romagnan, J., Legendre, L., Guidi, L., & Jamet, J. (2015). Comprehensive Model of Annual Plankton Succession Based on the Whole-Plankton Time Series Approach, 1–18. https://doi.org/10.1594/PANGAEA.833549.

Saavedra-Pellitero, M., Baumann, K.-H., Flores, J.-A., & Gersonde, R. (2014). Biogeographic distribution of living coccolithophores in the Pacific sector of the Southern Ocean. *Marine Micropaleontology*, *109*, 1–20. https://doi.org/10.1016/j.marmicro.2014.03.003

Sailley, S. F., Vogt, M., Doney, S. C., Aita, M. N., Bopp, L., Buitenhuis, E. T., Hashioka, T., Lima, I., Le Quéré, C., Yamanaka, Y. (2013). Comparing food web structures and dynamics across a suite of global marine ecosystem models. *Ecological Modelling*, *261–262*, 43–57. https://doi.org/10.1016/j.ecolmodel.2013.04.006

Tyrrell, T. and Charalampopoulou, A: Coccolithophore size, abundance and calcification across Drake Passage (Southern Ocean), 2009, https://doi.org/10.1594/PANGAEA.771715, 2009.

Vallina, S. M., Ward, B. A., Dutkiewicz, S., & Follows, M. J. (2014). Maximal feeding with active prey-switching: A kill-the-winner functional response and its effect on global diversity and biogeography. *Progress in Oceanography*, *120*, 93–109. https://doi.org/10.1016/j.pocean.2013.08.001

Vallina, S. M., & Le Quéré, C. (2011). Stability of complex food webs: Resilience, resistance and the average interaction strength. *Journal of Theoretical Biology*, *272*(1), 160–173. https://doi.org/10.1016/j.jtbi.2010.11.043

Vogt, M., Vallina, S. M., Buitenhuis, E. T., Bopp, L., & Le Quéré, C. (2010). Simulating dimethylsulphide seasonality with the Dynamic Green Ocean Model PlankTOM5. *Journal of Geophysical Research*, *115*(C6), C06021. https://doi.org/10.1029/2009JC005529

Wang, S., & Moore, J. K. (2011). Incorporating Phaeocystis into a Southern Ocean ecosystem model. *Journal of Geophysical Research*, *116*(C1), C01019. https://doi.org/10.1029/2009JC005817

Zondervan, I. (2007). The effects of light, macronutrients, trace metals and CO2 on the production of calcium carbonate and organic carbon in coccolithophores—A review. *Deep Sea Research Part II: Topical Studies in Oceanography*, *54*(5–7), 521–537. https://doi.org/10.1016/j.dsr2.2006.12.004